# Climate policy portfolios that accelerate emission reductions

Theodoros Arvanitopoulos [1,2,10], Simon Bulian [3,4,10], Charlie Wilson [5,6,10] ✉, Andrew J. Jordan [7], Jale Tosun [3,4,8] & Nicholas Vasilakos[9]

The corpus of national climate policies continues to grow - but to what effect? Using data on 3,917 policy instruments across 43 OECD countries and major emerging economies from 2000-2022, we show that national climate policy portfolios specializing in instrument types and sectors are associated with faster reductions in fossil $CO_2$ emission intensity. Supported by exemplar country case studies, we also provide quantitative evidence that the effectiveness of climate policy is amplified by long-term emission reduction targets and the presence of dedicated governmental bodies including ministries and intergovernmental organisations. The cumulative effect of all climate policy portfolios over our study period amounts to 3.1 $GtCO_2$ fewer emissions in 2022 relative to a no-policy counterfactual - substantially less than what's needed to stay on track for the Paris Agreement goals.

We are midway through the critical decade for climate change to keep Paris Agreement targets within reach. Global emissions need to peak and decline rapidly by 2030 to ensure near-term progress towards 1.5-2 °C climate stabilisation[1]. Encouraging trends have been observed nationally in 23 countries with declining absolute emissions over more than a decade, and in a further 67 countries with declining emission intensities per unit of economic output[2,3]. The two major emitters, the USA and China, fall in to the first and second groups respectively.

National climate policies help to explain these trends[4,5]. There are now thousands of climate-related laws and policies in almost two hundred countries[6,7]. Evidence synthesised in the IPCC's Sixth Assessment Report attributed 1.3–5.9 $GtCO_2e/yr$ of avoided emissions to climate policies including national commitments made under the Kyoto Protocol[8–10]. Attribution studies use econometric methods to identify the general effect of climate policies on emissions, controlling for other determinants of emission trends. These studies have demonstrated the effect of larger policy stocks[11,12], of more stringent policies[13], and of specific policy combinations[14]. The difference-in-differences modelling used by ref. 14 allowed stronger causal attribution but was focused on a set of 69 discontinuities in emission trends from recent policy introductions rather than the cumulative effect of policy portfolios over the long-term.

Policy portfolios have many different elements comprising, inter alia, instruments of different types, coverage of different emitting sectors, and different types of long-term objective setting strategic direction[15,16]. Policy portfolios are in turn shaped, implemented and monitored by different types of governmental organisation[17].

The design of each country's portfolio of climate policies varies in size, specialisation, and organisational context. We contribute an attribution analysis of these different elements of climate policy portfolios across a large sample of countries.

To do this, we analyse the instruments used and sectors targeted by 3917 policies introduced since 2000 in OECD and BRIICS economies [Fig. 1]. These 43 countries account for four fifths of global fossil $CO_2$ emissions[18]. We extend our dataset to include variables on the presence of different types of long-term national emission-reduction targets and governmental organisations relevant to climate policy effectiveness.

[1]Cardiff University, Cardiff Business School, Cardiff, UK. [2]London School of Economics, Hellenic Observatory, London, UK. [3]Heidelberg University, Institute of Political Science, Heidelberg, Germany. [4]Heidelberg University, Heidelberg Center for the Environment, Heidelberg, Germany. [5]University of Oxford, Environmental Change Institute, Oxford, UK. [6]International Institute for Applied Systems Analysis (IIASA), Laxenburg, Austria. [7]University of East Anglia (UEA), Tyndall Centre for Climate Change Research, Norwich England, UK. [8]University of Oslo, Department of Political Science, Oslo, Norway. [9]University of East Anglia (UEA), Norwich Business School, Norwich, UK. [10]These authors contributed equally: Theodoros Arvanitopoulos, Simon Bulian, Charlie Wilson. ✉e-mail: charlie.wilson@eci.ox.ac.uk

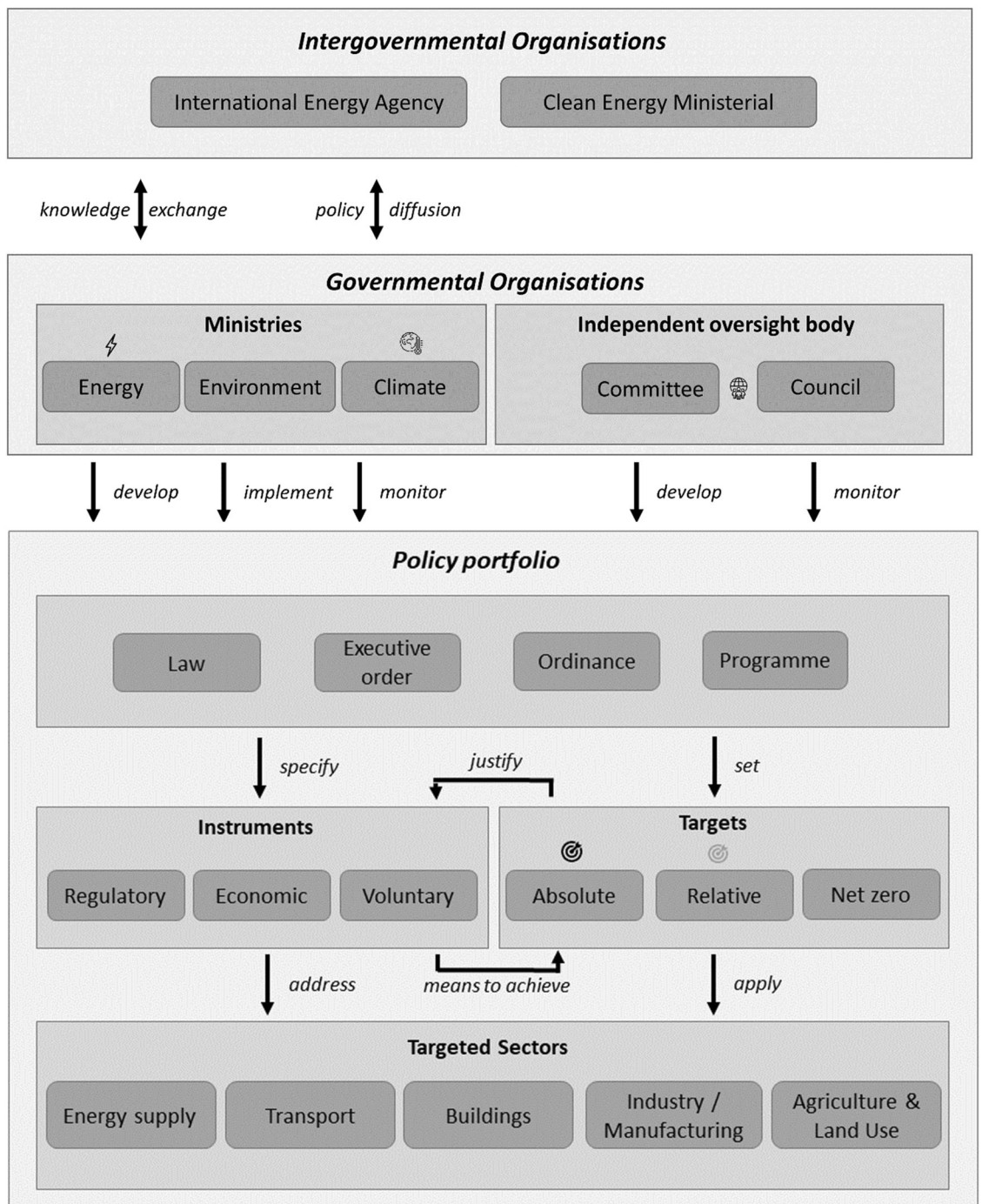

**Fig. 1 | Climate policy portfolios.** Upper part of schematic shows climate-relevant intergovernmental and governmental organisations. Lower part of schematic shows elements of national climate policy portfolios.

We then use panel regression models with country and year fixed effects to test the relationship between different policy portfolio elements and governmental organisations as the independent variables and fossil $CO_2$ emission intensity as the dependent variable over the period 2000–2022. Following[19], we use emission intensity rather than absolute emissions to control for the wide variation in country size and economic output in our sample, but in robustness checks we show our results also hold for absolute emissions. Using fossil $CO_2$ rather than all greenhouse gas (GHG) emissions allows us to focus on energy-related policies for which detailed data are available (see Methods).

Building on prior attribution studies (see SI1), we find that both policy density (i.e., the cumulative number of climate policies)[19] and

policy stringency[13] are associated with faster reductions in fossil $CO_2$ emission intensity. We interpret this result through the lens of policy sequencing which contends that policy portfolios become increasingly stringent as they develop[20,21]. Stringency is the calibration of policy instruments, e.g., the specific level of a carbon tax, and is broadly equivalent to strictness[22] or the extent to which climate policies incentivise or enable emission reductions[23].

We also find that policy portfolios targeting high-emitting sectors and specialising on instrument types are associated with faster reductions in emission intensity than those with more balanced economy-wide coverage or instrument type diversity. These are cumulative long-term effects of policy portfolios, and complement

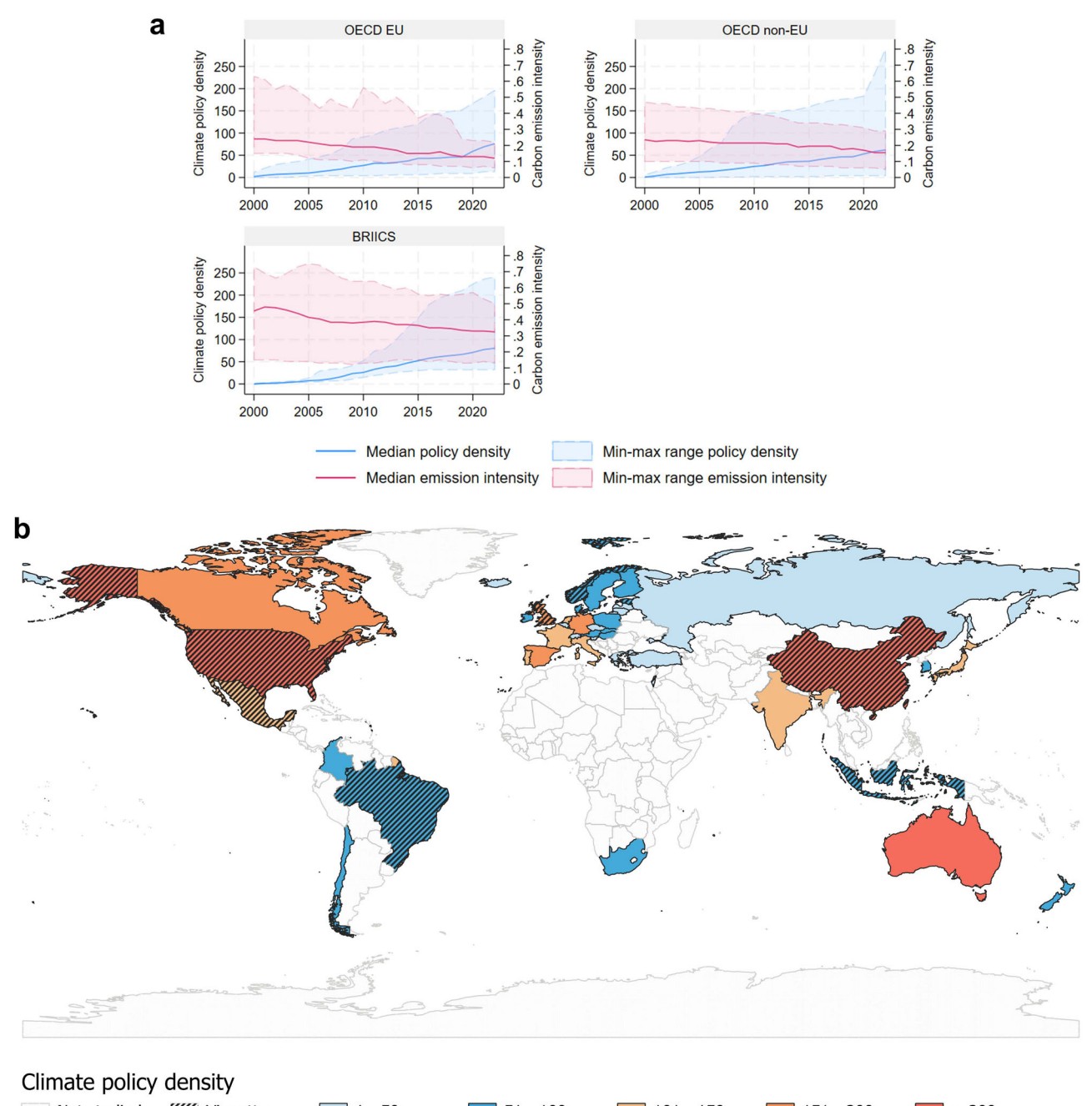

**Fig. 2 | Climate policy density in the 43 OECD + BRIICS countries analysed.** Upper panel [**a**] shows median (blue line) and extreme values (blue band) of climate policy accumulation and median (red line) and extreme values (red band) of fossil $CO_2$ emission intensity over 2000–2022 for three country groups (OECD countries in the EU, non-EU OECD countries, and BRIICS). Lower panel [**b**] maps cumulative numbers of climate policies in 2022, with hatching for countries selected for policy vignettes (see text for details).

insights from ex post evaluation studies that isolate the effect of specific instruments in specific sectors[14,24].

Further, we demonstrate the synergistic effect on climate policy effectiveness of national emission-reduction targets, dedicated energy or climate ministries, and membership of intergovernmental organisations like the International Energy Agency (IEA).

We illustrate and support these findings from our cross-country econometric analysis with policy vignettes for select countries [Figs. 2 + 3]. These vignettes of historical policy development help bring to life the quantitative modelling, illustrating how variation in national context shapes path-dependent policy portfolio design.

## Results

### Density & stringency of policy portfolios

We first apply Eskander and Fankhauser[19]'s model of climate policy density to our more extensive dataset of 3917 policies over the period 2000-2022 (compared with their 1092 policies over the period 2000–2016; see Methods).

The policy density model performs as expected, with faster reductions in $CO_2$ emission intensity observed in countries with larger stocks of climate policies [model 1, Table 1]. This effect controls for the characteristic U-shaped relationship between development stage and emissions, as well as emissions leakage through trade (see Methods and SI1 for extensive discussion on controls and model specification).

**Table 1 | Attribution models for different elements of national climate policy portfolios (43 OECD + BRIICS countries, 2000–2022)**

| Model reference in main text | (1) | (2) | (3a) | (3b) | (3c) |
|---|---|---|---|---|---|
| | Policy density (base model) | Policy instruments, 6 types § | Unweighted sectoral coverage | Weighted sectoral coverage | Weighted sectoral coverage (pre-Covid, to 2019) |
| *Dependent variable* | *emission intensity, log (CO₂/GDP)* | | | | |
| Policy density | -0.000554*** | -0.000688*** | -0.000594*** | -0.000673*** | -0.000562*** |
| | (0.000169) | (0.000166) | (0.000164) | (0.000161) | (0.000182) |
| Policy instrument diversity (6 types) § | | -0.114*** | | | |
| | | (0.0407) | | | |
| Policy sectoral coverage diversity (unweighted) | | | -0.124** | | |
| | | | (0.0513) | | |
| Policy sectoral coverage diversity (weighted) | | | | -0.195 | -0.252** |
| | | | | (0.128) | (0.109) |
| Control variables: | Table 4 (1) | Table SI5 (1) | Table SI5 (3) | Table SI5 (4) | Table SI5 (5) |
| Constant | -17.94*** | -18.08*** | -18.63*** | -18.72*** | -14.50*** |
| | (1.318) | (1.378) | (1.335) | (1.386) | (1.344) |
| Country & Year FE | YES | YES | YES | YES | YES |
| Observations | 941 | 919 | 923 | 923 | 795 |
| *R*-squared | 0.968 | 0.969 | 0.969 | 0.969 | 0.975 |
| within *R*-squared | 0.2728 | 0.2797 | 0.272 | 0.2678 | 0.1915 |
| RMSE | 0.0919 | 0.0895 | 0.0897 | 0.0899 | 0.0788 |

Models use two-way fixed effects panel regression. Dependent variable in all models is emission intensity measured as log(CO₂/GDP). The main independent variable in all models is policy density. Additional independent variables are tested in separate models alongside policy density. A one unit increase in policy density is associated with a change in emission intensity of magnitude given by 100% * the exponential of policy density coefficient minus one. See Methods for reporting of standardised effects. See Table 4 and Table S15 in SI1 for full models including controls.
Notes: Coefficient significance: ***$p < 0.01$, **$p < 0.05$, *$p < 0.1$. Robust standard errors shown in parentheses below coefficients. *FE* fixed effects. *RMSE* Root Mean Square Error.
§ Six policy instrument types are regulatory, market-based, direct provision (e.g. infrastructure investment), information, planning, and voluntary.

To help interpret the policy density model, we replace policy density with policy stringency (which is highly correlated) and show similar results.

Emissions fall faster as climate policy portfolios expand and become more stringent. Policy vignettes for the USA and China illustrate this general insight. The USA has continuously accumulated climate policies since 2000 anchored by milestone federal legislation in 2005 and 2009 on both clean energy production and the efficient use of energy, and more recently in 2021-22 for investment in infrastructure [Fig. 3a][25]. These reinforced a secular transition in the energy supply from coal to gas driven by cost and market dynamics.

China was later to start accumulating policies but caught up rapidly from 2005 including through five-year plans beginning in 2006 and 2011 that included energy efficiency and emission reduction targets respectively [Fig. 3b]. By 2022, China had adopted over 245 policies in total, second only to the USA. However, due to its different industrial development stage, China's emissions in absolute terms are still rising despite its high policy density[26].

Brazil provides a counter example of a country with comparatively few climate policies and slow reductions in emission intensity, exacerbated by a weak emphasis on energy and transport sector emissions in its flagship 2009 climate legislation [Fig. 3c][27].

Beyond these simple associations with policy density and stringency, other determinants of emission intensity trends are discussed in the policy vignettes in SI2. These include changing political regimes and priorities towards domestic resource exploitation including shale gas in the USA and oil in Brazil.

## Types of instrument in policy portfolios

Countries' climate policy portfolios vary not just in size and stringency but also in composition. We augment our policy density model [model 1, Table 1] to explore the additional effect of different design features of climate policy portfolios. We start with types of policy instrument.

A common taxonomy of policy instruments distinguishes three main categories: regulatory (rule-based), economic (market-based), and voluntary (participation-based)[28].

Using a diversity measure of instrument types, we find countries with higher specialisation on instrument types in their policy portfolios are associated with faster reductions in fossil CO₂ emission intensity. This holds for the simple three-category taxonomy (Table S15 in SI1) and for a more granular taxonomy of six instrument types [model 2, Table 1].

In sensitivity testing we find economic instruments have stronger effects than regulatory or voluntary instruments in line with expectations (see Methods and SI1). This is a general finding at the level of policy portfolios, complemented by other studies that pinpoint the effect of specific policy combinations[14] or instrument types[29].

Specialisation on instrument type is country-specific according to policy traditions (see SI1). Three policy vignettes illustrate the range.

Estonia specialises in economic instruments ( > 50% of all instruments since 2000) including carbon taxes (one of the first in the world), renewable energy feed-in tariffs, electric vehicle subsidies, and energy efficiency incentives [Fig. 3f][30]. In contrast, Israel specialises in regulatory instruments ( > 50% of all instruments since 2012) including rules on renewable energy grid connections and standards for energy efficiency in buildings [Fig. 3d]. Both countries show relatively large reductions in emission intensity.

Indonesia provides a counter example with no specialisation on instrument type and only small reductions in emission intensity [Fig. 3g]. Indonesia's policy portfolio contains a more diverse selection of instruments, including feed-in-tariffs for renewable energy and standards to enhance energy efficiency in buildings and transport[31].

## Sectoral coverage of policy portfolios

Climate policy portfolios target GHG emitting groups in different sectors. In most countries the highest emitting sector is either transport or the energy supply. Some countries focus policies on these or

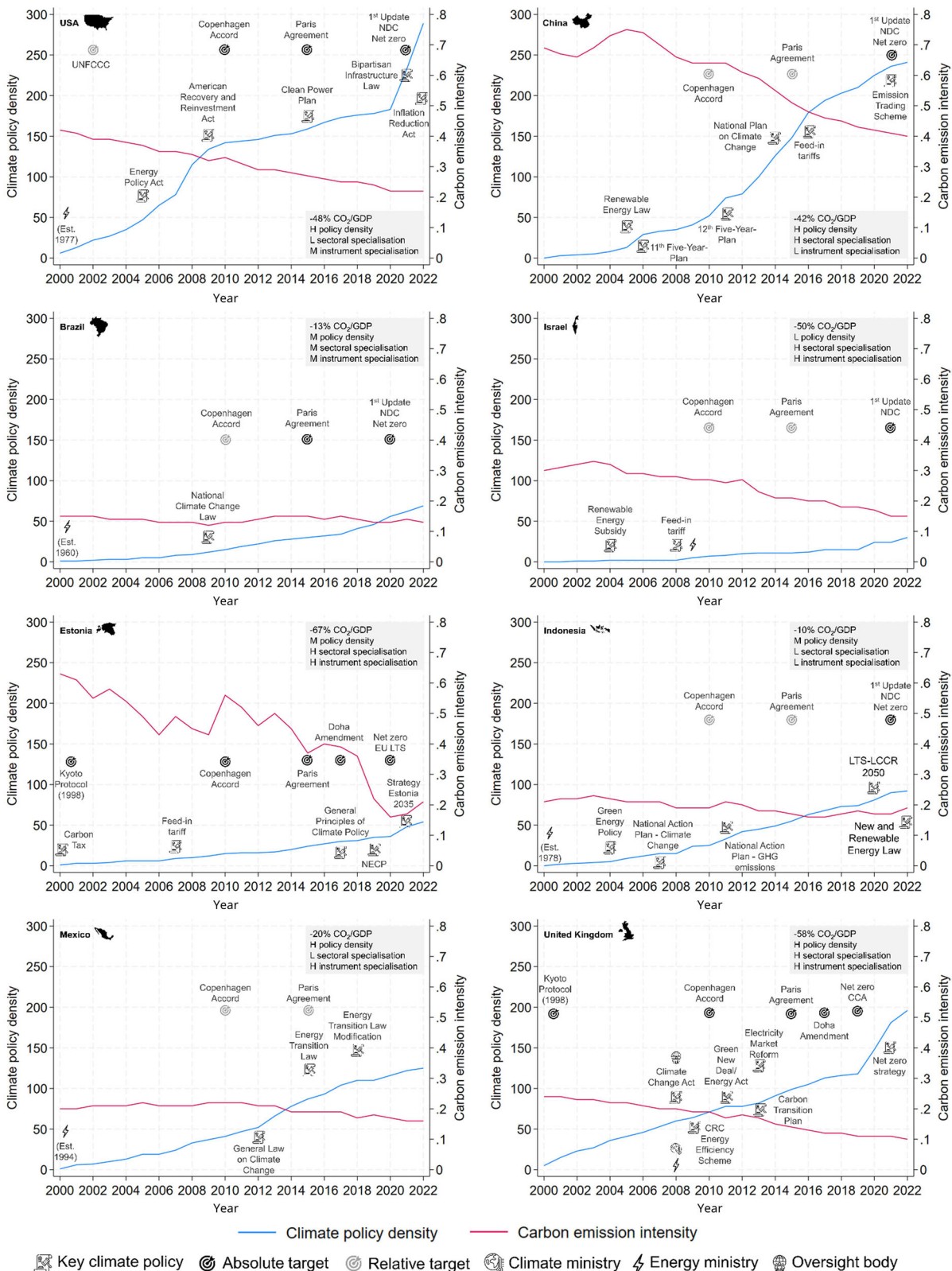

**Fig. 3 | Historical development of climate policy portfolios in select countries.** Country selection spans the H(igh), M(edium), L(low) range of values on the main independent variables: policy density, sectoral coverage, instrument type (see inset boxes on bottom or top right of each panel). Panels per country show change over 2000-2022 in fossil $CO_2$ emission intensity (red line), climate policy density (blue line), with policy portfolio milestones (icons). See Supplementary Information 2 (SI2) for full policy vignettes with data sources and country selection criteria. Panels from top left in rows: [a] USA, [b] China, [c] Brazil, [d] Israel, [e] Estonia, [f] Indonesia, [g] Mexico, [h] United Kingdom. Abbreviations: CCA = Climate Change Act, EU LTS = European Union Long-term Strategy, LTS-LCCR = Long-term Strategy on Low Carbon and Climate Resilience, NDC = Nationally Determined Contributions, UNFCCC = United Nations Framework Convention on Climate Change.

other sectors; other countries target economy-wide emissions across all sectors.

Using a diversity measure to characterise the sectoral coverage of national policy portfolios, we find that countries that concentrate on specific sectors reduce their emission intensity faster than those with broader coverage [models 3a & 3b, Table 1] (see Tables S14 and S15 in SI1). As expected, this concentration effect is stronger if coverage is weighted towards sectors with higher shares of national emissions.

Our policy vignettes for China [Fig. 3b] and Israel [Fig. 3d] show portfolio concentration on the energy supply (accounting for 46% and 61% of their respective national emissions from 2000–2022) to boost renewables deployment and accelerate the secular substitution of coal by gas in driving emission intensity reductions[32–34].

Our finding on concentrated sectoral coverage is sensitive to the 2020 Covid disruption to sectoral emission trends, particularly in transport, but is robust in the period up to 2019 [model 3c, Table 1], and in the full period to 2022 using an alternative diversity measure less distorted by the 2020 anomaly (see Methods and SI1).

### Long-term emission-reduction targets in policy portfolios

Emission-reduction targets are policy objectives designed to steer the general direction of policy portfolios including their constituent instruments. Targets set expectations but vary in form, stringency, and credibility.

We find evidence that both absolute and relative targets amplify the effect of the total stock of climate policies on reducing emission intensity [models 4a & 4b, Table 2] (see Table S3 in SI1). However, faster reductions are observed in countries with absolute targets, which predominantly have Kyoto-aligned 1990 reference years (see Methods). These results hold whether the target is measured in the year it was announced or the later year it entered into legal force. There is insufficient variability in our sample over the period 2000-2022 to isolate the effect of net-zero targets (12.5% of observations).

Policy vignettes for the USA and Mexico illustrate these findings. In 2009, the USA moved from relative to absolute targets, subsequently reinforced by further absolute targets for specific sectors [Fig. 3a][35]. This change was consistent with a steady accumulation in the number of policies over the period to 2022 concurrent with falling emissions[25].

Mexico was an early adopter of framework legislation in 2012 but adopted relative targets that only weakly signalled climate policy ambition[36]. This coincided with relatively slow progress on emission reductions [Fig. 3h].

### Governmental organisations alongside policy portfolios

Climate policy-relevant governmental organisations include: energy or climate ministries with capacity to develop and implement policy; advisory bodies for ensuring accountability and stability over political cycles; and intergovernmental or other international organisations for promoting and diffusing policy ideas and evaluation insights.

We find that countries with dedicated energy and/or climate ministries, and countries that are members of the International Energy Agency (IEA), have policy portfolios associated with faster reductions in emission intensity [models 5a & 5b, Table 2] (see Table S4 in SI1). We find similar effects for membership of the Clean Energy Ministerial (CEM) and the EU-EFTA.

We also test for the effect of independent advisory bodies. However, as these are a relatively recent organisational innovation (11.1% of observations), there is insufficient variability in our sample to isolate their effect on emissions.

## Discussion

As the urgency of delivering rapid, deep and sustained emission reductions intensifies, priorities for implementing the Paris Agreement have shifted[37]. The design of national climate policy portfolios is now firmly in the spotlight[5]: "What used to be a conversation about ambitious target setting now focuses increasingly on implementation and interventions to achieve these targets"[38].

By comparing observed emission trends with a counterfactual no-policy scenario, we attribute 27.5 $GtCO_2$ avoided emissions to the cumulative effect of all climate policy portfolios over the period 2000–2022, of which 14.6 $GtCO_2$ are in the BRIICS countries (see Fig. 4). This is equivalent to 3.1 $GtCO_2$/yr in 2022, or an average of 1.2 $GtCO_2$/yr from 2000-2022, which is in line with other attribution estimates that use a variety of methods, country and time samples (see SI1 for full discussion). Emission reductions attributed to climate policies in the literature include: 2-7 $GtCO_2$/yr (global, to 2020)[10], 1.3–5.9 $GtCO_2$/yr (133 countries, 1999–2016)[19], 1.3 $GtCO_2$/yr (39 countries, 2005-2012)[9], 1.3–2.5 $GtCO_2$/yr (renewable energy policies only)[10]. Other studies attribute emission reductions of 28% to climate policies (48 countries, 2000–2021)[13], of 0.8–2.8% to each new climate law introduced[12], or of 4–15% to carbon pricing policies only (21 countries)[24].

In their recent study using difference-in-differences methods to make more robust causal inferences than our fixed-effects model allows, Stechemesser, et al.[14] attributed total emission reductions of 0.6-1.8 $GtCO_2$ to a set of 69 structural breaks in sectoral emission trends (41 countries, 2000-2020). Their country sample and study period are similar to ours, but their policy-attributed emission reductions are an order of magnitude lower. We interpret this difference as the result of their specific focus on statistically identifiable discontinuities in sectoral emissions caused by specific combinations of policy instruments introduced or tightened in the preceding 2 years. In contrast, our models assess policy portfolios' cumulatively incremental impact over the long-term; discontinuities are subsumed within these aggregate economy-wide trends.

By exploiting policy variation in 43 countries over a 23 year period, our analysis shows that faster emission reductions are significantly associated with accumulating numbers of more stringent climate policies, specialisation on instrument types and on emission-intensive sectors, long-term objectives set by credible emission-reduction targets, and governmental organisations for policy development, implementation, monitoring and diffusion.

This contributes generalisable evidence to ongoing debates in policy science on the different elements of climate policy portfolio design.

First, our results confirm the basic insight from attribution studies that larger, more developed[19] or more stringent[13] policy portfolios are associated with faster emission reductions. Policy stocks grow due to complex policy design processes, interactions between policy goals and sectors, and competing political economic interests that all result in multiple policies or policy packages being introduced to tackle specific problems[39]. Climate policy accumulation further results from the incremental sequencing of policies over time to remove economic, societal, and political barriers to climate action[16,21]. Sequencing means policy density and policy stringency are strongly correlated as policies become more stringent as portfolios mature[13,20]. Stronger public demand for emission reductions further contributes to policy portfolios developing towards greater stringency[40,41]. This is why we interpret our main model result on policy density through the lens of stringency, which also underpins our attribution claim (see SI1 for full discussion).

Second, our results show that concentrating policy portfolios on emissions-intensive sectors (typically transport and energy supply) is more effective than aiming for economy-wide coverage. Although theoretically appealing, cross-sectoral coverage aided by greater climate policy integration faces practical or institutional constraints in the real world messiness of policymaking[42]. However, as our policy vignettes in Mexico[36], Indonesia[31], and Brazil[27] show, focusing climate policy portfolios on the energy supply sector in countries with strong

**Table 2 | Attribution models with interaction terms for long-term targets and governmental organisations alongside national climate policy portfolios (43 OECD + BRIICS countries, 2000–2022)**

| Model reference in main text | (1) | (4a) | (4b) | (5a) | (5b) |
|---|---|---|---|---|---|
| | Policy density (base model) | Absolute targets | Relative targets | Dedicated ministry | IEA |
| *Dependent variable* | emission intensity, log (CO₂/GDP) | | | | |
| Policy density (*independent variable*) | -0.000554*** | -9.47e-05 | -0.000320 | -0.00110*** | 0.000354 |
| | (0.000169) | (0.000178) | (0.000197) | (0.000206) | (0.000328) |
| Dummy: Absolute 1990 target | | 0.0157 | | | |
| | | (0.0212) | | | |
| Interaction: Absolute 1990 target * Policy density | | -0.00154*** | | | |
| | | (0.000222) | | | |
| Dummy: Relative target | | | 0.0248 | | |
| | | | (0.0303) | | |
| Interaction: Relative target * Policy density | | | -0.000770*** | | |
| | | | (0.000245) | | |
| Dummy: Energy and/or climate ministry | | | | -0.0471*** | |
| | | | | (0.0146) | |
| Interaction: Energy/climate ministry * Policy density | | | | 0.000847*** | |
| | | | | (0.000178) | |
| Dummy: IEA | | | | | -0.0670* |
| | | | | | (0.0346) |
| Interaction: IEA * Policy density | | | | | -0.00109*** |
| | | | | | (0.000338) |
| Control variables: | Table 4 | Table S3 (2) | Table S3 (4) | Table S4 (2) | Table S4 (4) |
| Constant | -17.94*** | -15.25*** | -19.58*** | -17.90*** | -13.87*** |
| | (1.318) | (1.233) | (1.560) | (1.389) | (1.734) |
| Country & Year FE | YES | YES | YES | YES | YES |
| Observations | 941 | 941 | 941 | 941 | 941 |
| *R*-squared | 0.968 | 0.970 | 0.968 | 0.968 | 0.969 |
| *F*-test (for Dummy and Interaction term) § | | 30.95 | 6.58 | 11.37 | 8.324 |
| *F*-test *P*-value | | 0.000 | 0.0015 | 0.000 | 0.000 |
| within *R*-squared | 0.2728 | 0.3175 | 0.2785 | 0.2895 | 0.2972 |
| RMSE | 0.0919 | 0.0891 | 0.0916 | 0.0909 | 0.0904 |

Models use two-way fixed effects panel regression. Dependent variable in all models is emission intensity measured as log(CO₂/GDP). The main independent variable in all models is policy density. Additional dummy variables are tested in separate models as interactions with policy density. The overall effect of policy density on emission intensity is given by the sum of the policy density and interaction coefficients (if both coefficients are statistically significant) or by the interaction coefficient if the policy density coefficient is not significant (see Methods). See Tables S3 and S4 in SI1 for full models including controls.

Notes: Coefficient significance: ***$p < 0.01$, **$p < 0.05$, *$p < 0.1$. Robust standard errors shown in parentheses below coefficients. *FE* fixed effects. *RMSE* Root Mean Square Error, *IEA* International Energy Agency.

§Interpretation of the *F*-test (for dummy and interaction term) is explained in Methods.

fossil fuel industries risks provoking resistance from incumbent interests as a form of climate policy constraint[43].

Third, our results show that specialising on instrument types in policy portfolios is associated with faster emission reductions. Different political traditions and contexts favour different policy instruments[44]. In a comparative analysis of renewable energy policy portfolios, Schmidt and Sewerin[45] similarly find that dominant instrument types vary by country, and that instrument specialisation rather than diversity is associated with higher renewable energy deployment.

Policy evaluation studies provide complementary insights on the effectiveness of specific instruments[24,46]. Over time, and with the increasing need for more stringent climate action, instrument choices have shifted from voluntary to regulatory and economic[47]. In their attribution models, Nachtigall, et al.[13] find that market-based instruments and targets are associated with faster emission reductions than non-market (regulatory) instruments. Additional robustness testing of our policy portfolio models using only subsets of policy instruments by type similarly show that portfolios weighted towards economic instrument types outperform others (see Methods and SI1).

From a normative perspective, instruments like carbon pricing offer the most economically efficient way of reducing emissions both within and across sectors[28,48]. However, carbon prices have been relatively low in many countries and carbon pricing has tended to be a more recent instrument choice[20]. The importance of regulatory and technology policies for emission reductions has been clearly evidenced in empirical assessments of low-carbon innovation[49] and in modelling analyses[50].

Consequently, rather than single instrument dominance, policy mixes - including those comprising different instrument types – tend to be more effective for multiple reasons including: mutual reinforcement and positive spillovers; complementary time-varying effects; capacity for policymakers to address multiple problems simultaneously[10]. Stechemesser, et al.[14] demonstrate empirically that policy combinations are more effective for reducing emissions than standalone policies, with carbon taxes an exception.

Fourth, politicians and governments spend a good deal of time debating and designing framework laws, targets and organisations. Our results show that certain types of emission-reduction target and

governmental organisation amplify the effectiveness of climate policy portfolios. Credible targets set strategic direction, shape long-term expectations, and help mobilise investment[51]. Our models indicate that absolute targets (reduction in reference year emissions by a target year) convey greater certainty and clarity of outcomes than targets expressed in relative terms of decoupling emissions from economic growth or counterfactual projections. Since the Paris Agreement, 147 countries accounting for 88% of global GHG emissions have now pledged net-zero targets over timescales from 2035 to 2070[52]. The credibility of these net-zero targets varies widely[51], but the long-term objectives they set in absolute terms for national climate policy portfolios are consistent with historical evidence on what works.

Sustaining and monitoring progress towards those long-term objectives falls to governmental organisations. Energy ministries, and more recently, dedicated climate ministries and independent advisory bodies support a country's capacity to develop, implement, and assess the effect of its climate policy portfolio[53]. Our results indicate the importance of energy and/or climate ministries for formulating and implementing climate policy. At the national level, few countries in our sample have established independent advisory bodies, but experiences in these countries, particularly in the UK, affirm their importance[54] (see SI2 for details).

Our models also suggest that countries' membership of intergovernmental organisations like the IEA and Clean Energy Ministerial (CEM) is beneficial. This adds an important international dimension to the organisational context that enables national climate policy effectiveness. The IEA has 32 country members from the OECD and plays an advisory and analytical role that supports diffusion of best practice in clean energy policy. In contrast the CEM is a global forum for promoting clean energy technology diffusion through its 29 member countries and participating companies[55]. Both organisations collaborate closely and are institutionally interconnected as the IEA hosts the CEM secretariat. Following Linsenmeier, et al.[56], we interpret their significance in our models as evidence of policy learning and diffusion in the case of IEA and of technology diffusion targeted by climate policies in the case of CEM. Membership of these intergovernmental organisations may also signal and strengthen domestic policy credibility. This cautions against overinterpreting their effect on country-level emission intensity reductions (we discuss potential endogeneity in our models in SI1).

The governmental organisations we have analysed as interacting synergistically with policy portfolios are but one element of a much wider governance landscape enabling climate action[17]. Codifying and integrating different measures of climate governance into the next generation of policy attribution models would help further advance our collective understanding of climate policy effectiveness.

To conclude, national climate policy portfolios and organisational architectures for climate mitigation are rapidly developing. However, given the ambition of current climate policies, an additional 16–24 GtCO2e emission reductions are needed in 2030 to stay on track with Paris Agreement goals[57]. The cumulative effect of increasingly stringent climate policy portfolios in our 43 country sample delivered a similar magnitude of fossil $CO_2$ emission reductions but over a 23 year period, reaching 3.1GtCO2/yr in 2022. What has worked historically to accelerate emission reductions now needs rapid amplification[58] through a ratcheting up of policy stringency alongside the diffusion of effective policy portfolio designs both within and beyond the major emitters.

## Methods
### Overall approach
Our econometric approach uses panel regression models with country and year fixed effects for 43 OECD + BRIICS countries over the period 2000–2022. Our dependent variable is fossil $CO_2$ emission intensity

($CO_2$ / GDP), using IEA emissions data. Our main independent variables are policy density, instrument type diversity, and sectoral coverage diversity, using coded IEA policy characteristics data. Using dummies, we also include as independent variables whether countries have long-term emission reduction targets, dedicated ministries, advisory bodies, and are members of intergovernmental organisations. We follow Eskander and Fankhauser[19] by including controls in our models for other economic, institutional, and geographic influences on emission intensity including economic development stage, climate zone, and overall governance quality.

To aid the interpretation of the econometric models, we supplement our analysis with climate policy vignettes for select countries that span the range of values on both dependent and independent variables. The policy vignettes describe key milestones and salient issues in the development of each country's climate policy portfolio over the period 2000-2022. The vignettes are provided in full in Supplementary Information 2 (SI2), and summarised graphically in the main text [Fig. 3].

### Data
Descriptive statistics for all our model variables are summarised in Table 3 for 37 OECD and 6 BRIICS (Brazil, Russia, India, Indonesia, China, South Africa) countries over the period 2000-2022.

**Dependent variable.** Our dependent variable is fossil $CO_2$ emission intensity per unit of GDP using data from the International Energy Agency (IEA)'s $CO_2$ Emissions From Fuel Combustion database. This covers $CO_2$ emissions by sector and by fuel at a highly disaggregated level for almost all countries from 1990.

On average across our 43 country sample, transport accounts for 29% of total $CO_2$ emissions over 2000–2022 and energy supply accounts for 30%. Emission shares from end-use sectors (transport, buildings, industry) vary by economic structure, building stock and heating fuels, transport infrastructure and modal splits. Emission shares from supply sectors (energy supply, energy own industry) vary by domestic resource base, energy industry materialisation, and electricity generation mix.

**Independent variables: climate policy portfolios.** Our main independent variable is policy density, the cumulative number of climate policies adopted in each country over the period 2000–2022. We also perform robustness checks with pre-2000 policy counts as initial conditions, but these do not affect our results (Table S12 in SI1). We test policy stringency as an alternative independent variable that is highly correlated with policy density and find similar results.

Our additional independent variables are diversity measures of both instrument type and sectoral coverage in national climate policy portfolios, and dummy variables for long-term emission-reduction targets.

To construct our climate policy variables we use the IEA's Policies and Measures Database (PMD) from which we identified a total of 3917 policies relevant for fossil $CO_2$ emissions for OECD and BRIICS countries over the period 2000–2022[59]. Advantages of the IEA PMD Database over the Climate Change Laws of the World database[6] used by Eskander and Fankhauser[19] are discussed in SI1.

We then coded all identified policies by instrument type and sectoral coverage. To code IEA PMD policies by instrument type, we use a standard taxonomy distinguishing legal, monitoring, regulatory, market-based, direct provision (e.g. infrastructure investment), information, planning, and voluntary[28]. There were fewer than 10 occurrences of the legal and monitoring instrument types in our dataset, so for our most granular analysis we combine these with regulatory instruments to give six instrument types overall.

For more parsimonious analysis we aggregated the instrument types into three commonly used policy instrument categories[60]:

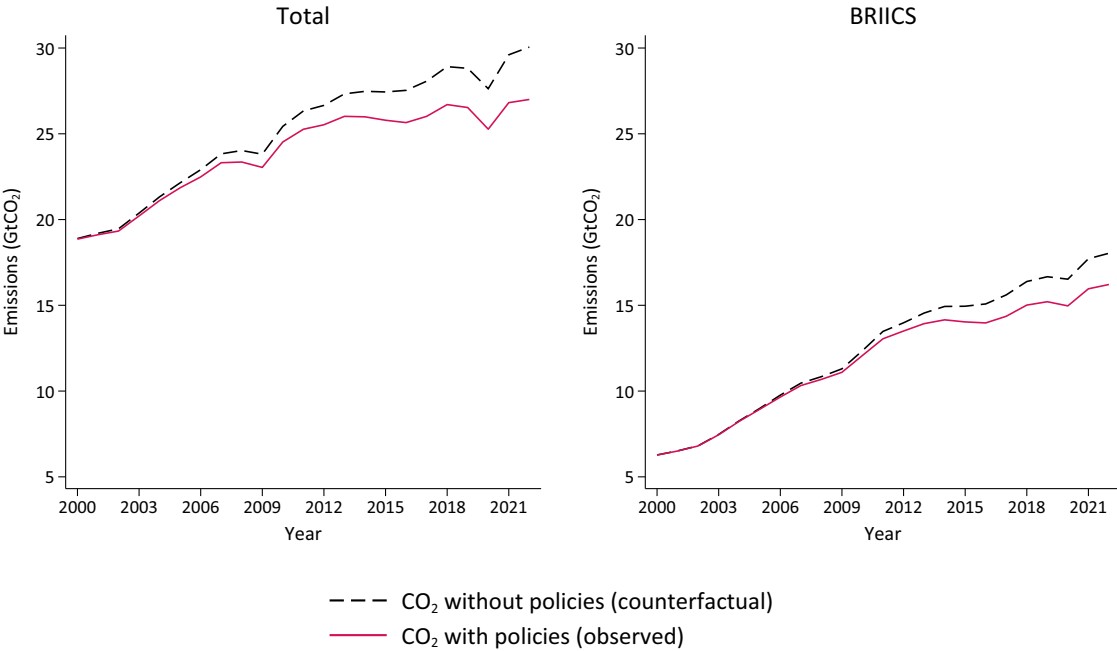

**Fig. 4 | Avoided CO2 emissions attributed to the cumulative effect of all climate policy portfolios over the period 2000–2022.** Left panel shows total avoided emissions in the 43 country sample (cumulatively 27.5 GtCO$_2$ over the study period, or 3.1 GtCO$_2$/yr in 2022); right panel shows avoided emissions in the 6 country BRIICS sample (14.6 GtCO$_2$ over the study period, or 1.8 GtCO$_2$/yr in 2022).

a.  Regulatory (incorporating legal, monitoring, and regulatory policy instrument types)
b.  Economic (incorporating market-based and direct provision policy instrument types)
c.  Voluntary (incorporating information, planning, and other voluntary policy instrument types)

To code IEA PMD policies by sectors covered, we use the standard energy-related set of two energy-supply sectors (energy industry own-use, and electricity and heat conversion) and three energy end-use sectors (buildings, transport, industry) used for GHG emission inventory reporting to the UNFCCC and in IEA energy statistics. We classified economy-wide or multi-sector policies as covering all five sectors. Policies not matched to any sector are classified as economy-wide policies only and comprise 2% of total observations.

Using our coded IEA PMD policy dataset, we calculated density and diversity measures. First, we construct our basic annual policy density variable to capture the number of climate policies adopted in a country in any given year. As past policies should continue to have an effect on emissions in years following their implementation, we use annual policy density to generate cumulative total policy density such that all past ($t$-1, $t$-2, …) and current ($t$) climate policies are counted at year $t$ within each country. This allows us to measure the cumulative effect of past and existing climate policies on fossil CO$_2$ emissions over the long term. We do not capture policy dismantling (not reported in the PMD). However this is both small and inversely correlated with new policy adoption so its effect is captured indirectly in our models (see SI1)[7].

Second, we construct diversity measures for both instrument type and sectoral coverage. To do so, we use the Herfindahl-Hirschman Index (HHI) which is widely used in economics to measure market concentration, and is commonly used in competition and regulatory economics to flag up market power and potential for anti-competitive behaviour[61]. The HHI takes values between 0 and 1, with 1 representing monopoly. In our case, a higher value of the HHI indicates lower diversity, i.e., higher concentration.

We construct the HHI for each country $i$ by taking for each year $t$ the sum of squares of the share of policy instrument types $s_{jt}^2$ (or sectors covered) within total policy density (Eq. 1). By using total policy density, we control for the cumulative (long-term) effect of specialisation on changes in fossil CO$_2$ emission intensity.

$$HHI_{it} = -\sum_{j=1}^{n} s_{jt}^2 \qquad (1)$$

For instrument types, we construct HHIs for both for the full taxonomy of six instrument types, and for the more aggregated three category taxonomy.

For sectoral coverage, we construct an unweighted HHI and a weighted HHI that weights policies per sector by that sector's share of total emissions. This allows us to test whether specialising in policies covering emission-intensive sectors has a larger effect.

As a robustness check in both cases, we also use the Shannon Index as an alternative diversity measure which is less sensitive to anomalous one-off variation that affects sectoral emission shares in the 2020 Covid year (see SI1).

For long-term emission-reduction targets, we use data from UNFCCC and Climate Action Tracker to construct dummy variables for the presence of absolute targets (% reductions relative to 1990 or other reference year emissions), relative targets (% reductions relative to business-as-usual projections or % reduction in emission intensity or emission intensity per capita), and net-zero targets. For each type of target we code alternative dummies for the year the target was announced and the later year it was ratified. Both dummies produce similar results; we use ratification year in our models.

For absolute targets, we find the 1990 baseline (in line with the Kyoto Protocol) captures most of the variation for target dummies so we use this in our models. Non-1990 absolute targets and net-zero targets contribute only 8.3% and 12.5% of observations.

**Independent variables: (inter)governmental organisations.** We use data from various sources (see SI1) to construct dummy variables for the presence of a dedicated energy and/or climate ministry, and an

**Table 3 | Summary of descriptive statistics for the 43 OECD + BRIICS country sample over the period 2000–2022**

| | Obs | Mean | S.D. | Min | Max |
|---|---|---|---|---|---|
| **Dependent variable** | | | | | |
| CO$_2$ emission intensity (kgCO$_2$ per 2015 USD PPP) | 989 | 0.24 | 0.13 | 0.05 | 0.75 |
| **Independent variables** | | | | | |
| Cumulative policy density | 989 | 39.25 | 42.49 | 0 | 289 |
| Policy instrument type diversity (HH index* 3 categories) | 963 | 0.43 | 0.15 | 0.33 | 1 |
| Policy instrument type diversity (HH index* 6 types) | 963 | 0.33 | 0.16 | 0.19 | 1 |
| Policy sectoral coverage diversity (HH index*) | 967 | 0.30 | 0.12 | 0.20 | 1 |
| Policy sectoral coverage diversity weighted (HH index**) | 967 | 0.06 | 0.04 | 0.00 | 0.54 |
| Absolute 1990 target (dummy variable) | 989 | 0.63 | 0.48 | 0 | 1 |
| Relative target (dummy variable) | 989 | 0.05 | 0.22 | 0 | 1 |
| Energy and/or climate ministry (dummy variable) | 989 | 0.51 | 0.5 | 0 | 1 |
| International Energy Agency (dummy variable) | 989 | 0.65 | 0.48 | 0 | 1 |
| **Control variables** [+] | | | | | |
| Rule of law (World Bank indicator) | 946 | 0.97 | 0.83 | -1.20 | 2.12 |
| Hodrick-Prescott GDP filter | 989 | 0.00 | 0.02 | -0.10 | 0.11 |
| GDP per capita, PPP (constant 2017 international USD)[++] | 989 | 43,217 | 23,416 | 3,094 | 140,436 |
| Imports of goods and services (% of GDP) | 989 | 42.95 | 25.62 | 9.10 | 179.92 |
| Services, value added (% of GDP) | 984 | 61.12 | 7.74 | 33.37 | 80.38 |
| Temperature variation (annual difference from long-term average 1990–2019) | 989 | 0.16 | 0.52 | -1.94 | 2.02 |

S.D. standard deviation. *Herfindahl-Hirschman index (HHI). **weighted by sectoral CO$_2$ emissions. [+] following Eskander and Fankhauser[19], see Methods. [++] also included as a squared term.

independent advisory body (like the UK's Committee on Climate Change). We also construct dummy variables for membership of intergovernmental organisations associated all or in part with clean energy or climate policy: the International Energy Agency (IEA), the Clean Energy Ministerial (CEM), and the EU (including Switzerland, and EFTA countries: Norway, Iceland, Lichtenstein). As we only consider national policies in our study, the EU dummy captures the interdependence between EU member states (and EFTA trading partners) with regard to their climate policies transposed from EU legislation into national contexts.

**Control variables.** We use the same set of control variables as Eskander and Fankhauser[19] and Nachtigall, et al.[13] to account for the political, economic, and geographic determinants of CO$_2$ emission intensity. We distinguish confounders that directly influence emission intensity from secondary controls whose effect is more diffuse. We discuss and substantiate our controls and analyse their impact on our identification strategy in SI1.

We control for three confounders related to economic development stage and structure. We use World Bank data on the log of GDP per capita and its squared term to control for the effect of a country's development stage and the non-linear (inverted U-shaped) relationship between GDP per capita and emission intensity. We include the share of total economic activity in the services sector to control for the structural shift away from more emission-intensive manufacturing and industrial activities. Additionally, we control for a country's import dependence by including the share of imports compared to the overall level of economic activity. We expect that a higher import share would indicate more services-oriented countries that import emission-intensive products from other countries with more industrial sector activity (known as 'carbon leakage').

We also include three secondary control variables for institutional quality, business cycles, and geography. For institutional quality, we use the rule of law index, estimated by the World Bank on an annual basis for all countries globally, and interpreted as a proxy for policy implementation and enforcement capacity. We expect countries with a stronger rule of law to be more effective in the adoption of climate policies. As our country sample covers only OECD and BRIICS economies, we also expect rule of law to be less decisive as a control variable than in Eskander and Fankhauser[19] which included a long tail of less developed economies with weaker and more heterogeneous policy implementation capacity. We include the Hodrick-Prescott GDP filter to control for cyclical volatility in economic activity. Although there is some empirical evidence suggesting that emissions tend to be more cyclically volatile than economic output[62], we do not form any strong expectation for this control. We incorporate the annual temperature variation from the long-term average to control for the effect of climatic conditions on emission intensity. We expect an increase in temperature above the long-term average to be associated with lower emission intensities, particularly for countries in the Northern hemisphere with winter heating seasons.

In SI1 we show our controls capture meaningful direct and indirect influences on policy density and are distinct from the country- and time-fixed effects in our models.

### Regression models: climate policy portfolios

Our method has three main steps to test observed relationships between the different elements of climate policy portfolios and emission intensity, beginning first with an application of Eskander and Fankhauser[19]'s model for policy density, before moving sequentially through layers of additional testing and complexity with the policy portfolio variables unique to this study.

**Policy density model.** Our policy density model follows the specification of Eskander and Fankhauser[19] with one main difference: we do not incorporate a short-term policy density variable as a proxy for unobserved (omitted) past climate policies. Our use of the more comprehensive IEA PMD policy dataset includes both past and existing climate polices in contrast to the CCLW dataset used by Eskander and Fankhauser[19] which focuses primarily on legislation over other policy types. We therefore have a larger policy dataset ($n = 3917$ IEA PMD policies vs $n = 1092$ CCLW legislations) over a slightly longer time period (2000-2022 vs 1996–2016) but for a smaller country sample (43 OECD + BRIICS countries vs 133 countries).

Our policy density model is shown in Eq. (2):

$$\ln\left(\frac{CO2}{GDP}\right)_{it} = \alpha + \beta_1 Pdens_{it} + \beta_2 X_{it} + \theta_t + \eta_\tau + \varepsilon_{it} \qquad (2)$$

where $\ln\left(\frac{CO2}{GDP}\right)_{it}$ stands for the natural log of emission intensity per country $i$ and year $t$; $\alpha$ is the constant parameter; $Pdens_{it}$ is the total policy density measuring the cumulative number of climate policies per country $i$ for year $t$; $X_{it}$ is a vector of control variables; $\theta_t$ are the country fixed effects; $\eta_\tau$ are year fixed effects; and residuals $\varepsilon_{it}$. The use of the natural log in the dependent variable improves model fit and facilitates the interpretation of coefficients such that a one unit increase in policy density would lead to a $(\exp(\beta_1)-1)*100\%$ change in the dependent variable.

This model is easy to augment with additional variables for other elements of national climate policy portfolios including specialisation on instrument type and sectoral coverage (see below).

We check the robustness of our model specification by observing whether the coefficients of independent variables and controls change in sign, magnitude, or significance compared with the OECD-only model from Eskander and Fankhauser[19]. (Coefficients in the two models are not directly comparable due to differences in policy density specification, country and time samples). We expect policy density to be negative and significant as an indication of climate policy effectiveness for reducing emission intensities after controlling for the level of economic activity in each country. This is indeed the case. Our policy density model results are summarised in Table 4 and shown in full in Table S11 and Table S21 in SI1.

In our model the coefficients for GDP per capita and its interaction with the square term, imports share of GDP, and services share of GDP are all statistically significant and with signs in line with expectations (see SI1 for detailed discussion of confounders and controls in our model specification). The OECD-only model of Eskander and Fankhauser[19] does not find the GDP terms to be significant which we interpret as being due to lower GDP per capita variation in their OECD-only sample. Other minor differences between the models are shown in Table 4.

As noted our policy density variable is a cumulative measure of all past and existing policies. As a robustness check, we test an alternative model (Table SE, column 2) with a 3-year lagged version of the policy density variable. The model is very similar but the coefficient of the lagged policy density variable is slightly larger in absolute terms. This supports our interpretation that accumulating policy stocks antecede emission intensity reductions rather than vice versa (see below on endogeneity).

Coefficients for the controls are also very similar between these alternative model specifications. Minor exceptions in the lagged policy density model are that the coefficient for Hodrick-Prescott GDP filter becomes weakly statistically significant, and the coefficient for temperature variation increases in magnitude and also becomes statistically significant.

We also directly replicate the model of Eskander and Fankhauser[19] with both short-term and long-term policy density variables but applied to our smaller country sample, later time period, and larger policy sample. Again, our results were very similar, with full results shown in Table S11 in SI1.

Interpreting the coefficient for policy density – a count variable - is enabled by the log transformed dependent variable. A one unit increase in policy density is associated with a change in emission intensity of magnitude given by 100% * the exponential of the policy density coefficient minus one. In our base model shown in Table 4, this gives a -0.055% change in emission intensity per unit increase in policy density.

Standardising coefficients using the within-group standard deviation can improve interpretability, particularly when comparing effect sizes across variables measured on different scales. We implement the standardisation procedure proposed by ref. 63. We residualise each predictor with respect to country and year fixed effects, calculate the within-unit standard deviation of the residuals, and use these to standardise the independent variables before re-estimating the model. A one standard deviation increase in policy density (equivalent to ~20 policies) is associated with a -1.1% change in emission intensity. This is comparable in magnitude to the standardised effects of the key confounders (GDP per capita, services share, imports share), and substantially larger than that of secondary controls (rule of law, Hodrick-Prescott filter, temperature variation). See S1 for full results and discussion.

**Policy stringency as alternative to policy density.** Policy stringency and policy density are strongly correlated. This was demonstrated by Schaffrin, et al.[22] with their measure of policy strictness for three countries' climate policies, and confirmed for a large country sample by the OECD using their own stringency measure, the CAPMF (e.g., Fig 3.2 in ref. 64). Stringency is defined as "the degree to which climate actions and policies incentivise or enable GHG emissions mitigation at home or abroad".[13]

The OECD's Climate Actions and Policies Measurement Framework (CAPMF)[64] provides one index on the stringency of climate policies, with sectoral sub-indices (including for buildings, industry, transport, and electricity supply). The index measures the degree to which sectoral climate actions and policies incentivise or enable GHG emissions mitigation. As a country example for the UK, Fig 3.5 in ref. 64 shows that stringency increased with the adoption of more policies over time.

The CAPMF provides a generalisable measure of policy stringency characterising national climate policy portfolios. However, we cannot include CAMPF as an additional portfolio characteristic due to collinearity: it has a 71% correlation with policy density. There are also two mismatch problems. Although the timeframe of the CAPMF is 1990-2022, covering our model period, its OECD country sample does not include Brazil and the United States. The policies covered by CAPMF also do not include many of those coded as voluntary or informational in our policies dataset from IEA PMD.

Consequently, we use CAPMF as an alternative independent variable to policy density for robustness testing our main model, despite the mismatch of policy and country samples. We take the mean of the sectoral policy stringency indices from CAPMF weighted by each sector's share of total emissions. This is similar to our procedure for constructing emission-weighted diversity indices in our testing of policy sectoral coverage.

We show the full results in Table S20 in SI1. Like policy density, the coefficient for the weighted sectoral policy stringency index is negative and statistically significant at $p < .01$. A more stringent climate policy corpus is associated with faster reductions in emissions intensity. This complements our main finding on the effect of climate policy density.

**Policy portfolio specialisation models.** Having established our base policy density model, we then augmented it with additional independent variables that measure different elements of climate policy portfolios. First, we control for the effect of specialisation in sectoral policy coverage and instrument type on emission intensities. We incorporate in our base model the HHI measures of diversity. This augmented model is shown in Eq. (3):

$$\ln\left(\frac{CO2}{GDP}\right)_{it} = \alpha + \beta_1 Pdens_{it} + \beta_2 HHI_{it} + \beta_3 X_{it} + \theta_t + \eta_\tau + \varepsilon_{it} \qquad (3)$$

where $\ln\left(\frac{CO2}{GDP}\right)_{it}$ is the natural log of emission intensity per country $i$ and year $t$; $\alpha$ is the constant parameter; $Pdens_{it}$ is the total policy density measuring the cumulative the number of climate policies per country $i$

**Table 4 | Policy density model including all controls over the period 2000–2022**

| Model reference in main text | (1) | | |
|---|---|---|---|
| | Policy density model | Eskander and Fankhauser[19] model over the period 2000-2016 for comparison: coefficient signs & significance | |
| Country samples | OECD + BRIICS | 133 countries | OECD only |
| | Emission intensity (log $CO_2$/GDP) (dependent variable) | | |
| Policy density § (independent variable) | -0.000554*** | -0.0179*** | -0.0083*** |
| | (0.000169) | (0.0014) | (0.0012) |
| Policy density, just last 3 years (independent variable) | | -0.0078*** | -0.0046** |
| | | (0.0021) | (0.0019) |
| Control variables: | | | |
| Rule of law | -0.0374 | -0.6164*** | -0.2051 |
| | (0.0301) | (0.1168) | (0.2685) |
| Hodrick-Prescott GDP filter | -0.354 | 0.3679* | -0.0110 |
| | (0.249) | (0.2124) | (0.2305) |
| GDP per capita (log) | 3.735*** | 1.1623*** | -0.2547 |
| | (0.288) | (0.2570) | (1.0765) |
| GDP per capita (log) squared | -0.200*** | -0.0840*** | -0.0065 |
| | (0.0153) | (0.0138) | (0.0578) |
| Imports share of GDP | -0.00410*** | 0.0018*** | -0.0013* |
| | (0.000647) | (0.0006) | (0.0007) |
| Services share of GDP | -0.00782*** | -0.0029** | -0.0073** |
| | (0.00220) | (0.0012) | (0.0031) |
| Temperature variation | -0.0176** | -0.0123* | -0.0101* |
| | (0.00792) | (0.0067) | (0.0060) |
| Federal systems | | 0.0059 | 0.0196 |
| | | (0.0407) | (0.0325) |
| Constant | -17.94*** | -4.5877*** | 2.6982*** |
| | (1.318) | (1.2092) | (4.9767) |
| Country & Year FE | YES | YES | YES |
| Observations | 941 | 2394 | 738 |
| within R-squared | 0.2729 | 0.214 | 0.234 |
| RMSE | 0.0919 | | |

See Table S11 and Table S21 in Supplementary Information 1 (SI1) for models with alternative lagged climate policy density variables as robustness checks.

Note: *FE* fixed effects. *RMSE* root mean square error. Robust standard errors shown in parentheses. Significance: ***$p < 0.01$, **$p < 0.05$, *$p < 0.1$.

§ In Eskander & Fankhauser's model, policy density is lagged one year and excludes policies introduced in the past 3 years.

for year $t$; $HHI_{it}$ stands for the Herfindahl-Hirschman Index per country $i$ for year $t$; $X_{it}$ stands for all controls; $\theta_t$ are the country fixed effects; $\eta_\tau$ are year fixed effects; and residuals $\varepsilon_{it}$.

We estimate four separate models, one for each of the four HHI variables (see Table 3). Full results are provided in Table S15 in SI1 with model summaries in the main text. We also run robustness checks using the alternative Shannon Index as a diversity measure and find similar results (Table S14 in SI1).

Models 2a and 2b in Table 1 (main text) include the HHIs for six instrument types and three instrument categories respectively. Coefficients for these HHI variables are negative, statistically significant, and are of comparable size. A higher specialisation on instrument type (higher HHI) is associated with larger reductions in emission intensity. The effect of both policy density and the controls remains virtually identical in both models, and is similar to the base model.

Models 3a and 3b in Table 1 (main text) include the HHI for sectoral policy coverage weighted by the sectoral share of emissions, and the non-weighted HHI. Model 3c in Table 1 shows the weighted HHI for the period 2000-2019 prior to the Covid disruption. After controlling for the 2020 anomaly in sectoral emission shares (see SI1 for details), HHI coefficients are significant and negative. A higher sectoral concentration of policies is associated with larger reductions in emission intensity. The coefficient for the weighted sectoral HHI is almost twice that for the unweighted sectoral HHI. Policy concentration on emission-intensive sectors is more effective. The coefficient for the policy density variable reduces more alongside the unweighted sectoral HHI than for the weighted sectoral HHI, but the augmented model is broadly consistent with the base model. The controls also have very similar effects across the different model specifications.

**Policy density model interaction effects with long-term targets and organisations.** We use dummy variables to test the interaction between policy density and (1) long-term emission-reduction targets, (2) dedicated energy and/or climate ministries, (3) independent advisory bodies, (4) membership of intergovernmental organisations including the International Energy Agency (IEA) and the Clean Energy Ministerial (CEM).

Following Wooldridge[65], we use the model shown in Eq. (4):

$$\ln\left(\frac{CO2}{GDP}\right)_{it} = \alpha + \beta_1 Pdens_{it} + \beta_2 X_{it} + \delta_0 dummy_{it} + \delta_1 Pdens*dummy_{it} + \theta_t + \eta_\tau + \varepsilon_{it}$$

(4)

where $\ln\left(\frac{CO2}{GDP}\right)_{it}$ stands for the natural log of emission intensity per country i and year t; $\alpha$ is the constant parameter; $Pdens_{it}$ is the total policy density measuring the cumulative number of climate policies per country i for year t; $X_{it}$ stands for all controls; $dummy_{it}$ accounts for the presence/absence of targets or organisations; $Pdens*dummy_{it}$ is the interaction term between policy density and the dummy; $\theta_t$ are the country fixed effects; $\eta_\tau$ are year fixed effects; and residuals $\varepsilon_{it}$.

Interpreting the coefficients in these dummy interaction models is somewhat complex, but in summary the overall estimated effect of total policy density on emission intensity when dummy=1 is given by summing coefficients $\beta_1$ and $\delta_1$ (on the condition that both coefficients are significant). If the coefficient for policy density is not significant, we treat it as 0, so we effectively sum 0 to the coefficient for the interaction term. In such cases, the fact that policy density becomes non-significant does not imply that the target or organisation does not help reduce emission intensity. This is because the policy density variable is highly correlated with the interaction term and so they compete for the same effect. (See SI1 for full discussion of the interaction terms and their interpretation).

Models 4a and 4b in Table 2 (main text) summarise the results for absolute and relative emission-reduction targets respectively. Full results are provided in Table S3 in SI1.

For absolute targets (with 1990 reference year), the coefficient for the interaction term is negative, statistically significant, and has about three times larger an effect than the coefficient for policy density in the base model. The coefficient for policy density becomes statistically non-significant, as variation has been picked up by the interaction term, i.e. the countries with absolute targets (62% of the sample).

As a robustness check, we tested an additional model (Table S3 in SI1, column 3) that includes the dummy for absolute targets but not the interaction term. Coefficients for both policy density and dummy are negative and statistically significant. The policy density coefficient becomes non-significant when we include the interaction term.

For relative targets, the coefficient for the interaction term is statistically significant and smaller in magnitude than in the model for absolute targets, indicating relative targets have less of an effect on emission-intensity reductions. In addition, the positive and statistically significant dummy coefficient indicates that these countries with relative targets (5% of the sample) start off with higher emission intensities than the rest.

Both these results hold whether the targets are measured in the year they were announced or the later year when they entered into legal force through ratification.

For the governmental organisations with the exception of dedicated energy and/or climate ministries (50.6% of observations) and membership of intergovernmental organisations (e.g., CEM 29.6% of observations), there was insufficient variation to isolate their effect on emission intensities. Net-zero targets are observed in only 12.5% of observations and independent bodies in 11.1%.

Models 5a and 5b in Table 2 (main text) summarise the results for dedicated energy and/or climate ministries and membership of the IEA respectively. Full results are provided in Table S4 in SI1.

For dedicated ministries, the coefficient for the interaction term is positive, statistically significant, and the sum of the two coefficients is negative but smaller in size than the policy density coefficient which is negative and significant. For membership of the IEA, the coefficient for the interaction term is statistically significant, negative, and larger in magnitude than the coefficient for policy density in the base model, indicating that membership of this international body is associated with a strengthened effect of policy accumulation on emission intensity. Similar results are found for membership of the CEM and EU-EFTA (Table S4 in SI1).

## Policy density model: robustness checks & alternative model specifications

We run numerous robustness checks on our policy density model including:

- inclusion of pre-2000 climate policies as initial conditions (Table S12 in SI1)
- exclusion of China (due to large differences in policy counts between IEA PMD and CCLW databases) (Table S12 in SI1)
- 1-year lag on all independent and control variables (Table S13 in SI1)
- replacement of emission intensity with absolute emissions as dependent variable (Table 16 in SI1)

We also test alternative model specifications with subsets of our policy portfolio variables including:

- replacement of HHI diversity measures with Shannon Index diversity measures (Table S14 in SI1)
- policy density for each instrument type (Table S17 in SI1)
- emission intensity for each sector (Table S18 in SI1)

Finally, we test different combinations of control and independent variable including:

- parsimonious controls (Tables S5 and S6 in SI1)
- full combinations of independent variables (Table S19 in SI1)

Results are discussed in SI1. We found no material changes in model results or coefficients in all our robustness tests, and conclude our policy density model is robust to methodological variation and alternative specification. In SI1 we also include full discussion and analysis of confounders and our use of controls.

## Endogeneity and reverse causality

In main text we report associations between climate policy portfolios and emission-intensity reductions. Our two-way fixed effects models are not able to establish causality but we interpret our results as showing climate policies are antecedent to emission reductions. As noted, Stechemesser, et al.[14] use difference-in-difference (DiD) methods to show that climate policies lead to discontinuous reductions in sectoral emissions within two years of introduction. This shows causality from policies to emission reductions. Our aim is different in assessing the path-dependent effect of climate policy stocks on cumulatively incremental emission reductions (with our additional testing of the interacting role of long-term targets and governmental organisations). We align with the methodological framework established by Eskander and Fankhauser[19] in the climate attribution literature and replicate their model specifications for comparison.

In SI1 we address endogeneity issues and provide multiple lines of reasoning to counter reverse causal interpretations of our model results. These include the Durbin-Wu-Hausman test (for endogeneity) and the Dumitrescu-Hurlin test (for Granger causality). We also include arguments and evidence from lagged models, fixed effects, weak correlations between emission intensity with covariates, recent literature, our policy vignettes, and our choice of econometric model. In combination these support our attribution of emission intensity reductions to national climate policy portfolios, controlling for the effect of other influences on emissions.

## $CO_2$ emissions avoided due to climate policy portfolios

We calculate avoided $CO_2$ emissions due to climate policy by estimating a counterfactual with no climate policies, following the methodological framework proposed in ref. 19. We set policy density equal to zero in Eq. (2) for our base policy density model, and then subtract it from the full estimated Eq. (2):

$$\ln\left(\frac{\widetilde{CO2}_{it}}{GDP_{it}}\right) - \ln\left(\frac{\widehat{CO2}_{it}}{GDP_{it}}\right) = \hat{\beta}_1 Pdens_{it} \quad (5)$$

where we use $\widetilde{CO2}_{it}$ for the counterfactual emissions and $\widehat{CO2}_{it}$ for the esstimated emissions.

However:

$$\ln\left(\frac{\widetilde{CO2}_{it}}{GDP_{it}}\right) - \ln\left(\frac{\widehat{CO2}_{it}}{GDP_{it}}\right) = \ln\left(\frac{\widetilde{CO2}_{it}}{\widehat{CO2}_{it}}\right) \quad (6)$$

Therefore, combining Eqs. (5) and (6), we get:

$$\widetilde{CO2}_{it} = \widehat{CO2}_{it} \exp\left(-\hat{\beta}_1 Pdens_{it}\right)$$
$$\approx CO2_{it} \exp(-\hat{\beta}_1 Pdens_{it}) \quad (7)$$

in which estimated emissions can be replaced by observed emissions.

Finally, we aggregate emissions estimated over countries $i$ and time $t$ to calculate total counterfactual emissions under no climate policies as shown in Eq. (8).

$$\widetilde{CO2}_{total} = \sum_i \sum_t \widetilde{CO2}_{it} \quad (8)$$

Figure 4 shows the results for the full 43 country sample (left panel) and for the BRIICS country bloc (right panel).

## Reporting summary

Further information on research design is available in the Nature Portfolio Reporting Summary linked to this article.

## Data availability

In Methods we summarise our variables and the data used in their construction. In Supplementary Information 1 (SI1), we provide further detail on variable construction and data sources.

The data for our independent variables have been stored in the heiDATA repository under the accession code https://doi.org/10.11588/DATA/XTDNAU. The data used to construct our control variables are publicly available and can be accessed via the World Bank (see Table S1 in SI1 for details). The data for our dependent variables on fossil $CO_2$ emission intensity are available under restricted access through the IEA's Greenhouse Gas Emissions from Energy database (upgrade of the former IEA $CO_2$ Emissions from Fuel Combustion database); access can be obtained through the IEA.

## Code availability

The Stata.do file used in our analysis has been storied in the heiDATA repository under the accession code https://doi.org/10.11588/DATA/XTDNAU for replication purposes.

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

## Acknowledgements

We acknowledge support from the following projects and funders: T.A. – EDITS (Ministry of Economy, Trade, and Industry (METI), Japan); CW – iDODDLE (ERC Grant #101003083); A.J. – CAST (UK ESRC Grant #ES/S012257/1); A.J., S.B. & J.T. – DeepDcarb (ERC Grant #882601); J.T. – ACCELZ (Research Council of Norway Grant #335073); NV – CROSSEU (EU Horizon Grant #101081377). We also thank Brendan Moore for sharing his data on advisory bodies, and Jan Ivar Korsbakken for his advice on emissions data.

## Author contributions

T.A., S.B., C.W. contributed equally. T.A. designed and ran the econometric models, with support from N.V. S.B. analysed and coded the policy variables and wrote the policy narrative vignettes, with support from J.T. and A.J. S.B. and T.A. designed the Figures. C.W. designed and led the overall study. C.W. wrote the manuscript. T.A. and C.W. wrote SI1. S.B. wrote SI2. All authors commented on and edited the manuscript.

## Competing interests

The authors declare no competing interests.
