## [Transparent Peer Review file · Nature Communications]

Climate policy portfolios that accelerate emission reductions

Corresponding Author: Professor Charlie Wilson

Version 0:

Reviewer comments:

Reviewer #1

(Remarks to the Author)

The study and its methodology are transparent and well written. I also like the presentation of national figures in the paper and the thorough analysis of national projections in the supplementary information. The study finds quantitative evidence that national climate policy portfolios that specialise in certain instrument types and sectors are associated with faster reductions in fossil CO2 emission intensity. The link with policy instruments is particularly interesting.

This study builds on the methodology of Eskander and Fankhauser (2020), but the authors have extended it to a longer time horizon (2000-2019) and cover more policies. The in-depth country analysis of this study is also an added value compared to the 2020 study.

I think the work could be important in this area, although a better presentation in a chart of the main instrument types and sectors associated with faster reductions in fossil CO2 emission intensity for a selected group of countries would be welcome. As it is, the main policies are included in the figures, but this is widely known.

The disadvantage of the method is that it still uses the period that is not very recent, as it misses the last five years where even more policies have been developed, as we also see in the figure, as it misses Fit-for-55 for the EU and IRA for the US. I can understand why you did not focus on the period until 2022 because of the impact of COVID, but the current emissions trend for some countries with many climate policies are lower compared to 2019 levels (Canada, US, EU).

There is also an interesting paper in Science on climate policies that have achieved large emission reductions: Global evidence from two decades, which looks at 1500 climate policies implemented between 1998 and 2022 in 41 countries across six continents. It would be good if the authors would also compare some of their results with this study, although I can understand that this study was published after your submission.

I would recommend publication if the authors adequately address the main comments listed below:

The main difficulty I have is with the intensity of climate policy: It is obviously difficult to measure the stringency and GHG reduction impact of all types of climate policies. Nevertheless, the lack of a measure of policy stringency is a major limitation of this paper, because the number of policies is not a good measure of the quality and GHG impact of climate policies. Countries may have many policies and a high policy density, but this does not guarantee that they will reduce CO2 emissions. For example, a single strong national policy based on an ETS with nationwide coverage of all GHGs could be much more effective than multiple policies that have not reduced emissions at all in the end. It all depends on the quality and stringency of the policies and their coverage across sectors to determine the GHG impact. I therefore suggest that the issue of stringency and the GHG impact of policies should be addressed in some way. I assume that the earlier work by Eskander and Fankhauser also addressed this issue.

I also don't quite understand why the authors use policy intensity in terms of numbers, as this does not take into account total GHG emissions, nor sectoral coverage, and compare it to a country's emissions intensity (i.e. nationwide emissions). I would expect policy density to also be relative to the size of a country's emissions, as well as sectoral coverage. Perhaps you could elaborate on this.

I think the two tables are not always very attractive to the Nature Communications audience, especially those with many blank cells. A better presentation of Table 2 would be welcome. It would also be good if the authors could present the most effective policies according to their analysis for the major emitting countries.

Please use the Nature communications citation and references format.

Specific comments:

You mention twice: "National climate policy portfolios that specialise in certain types of instruments and sectors are

associated with faster reductions ...". It would be good to give an example of what you mean by certain?

Reviewer #2

(Remarks to the Author)

I would like to thank the authors for a very interesting and timely study on a very important topic. The question of the impact of individual policies and policies cumulatively is highly relevant and hence deserves academic attention. I have a number of concerns that I would ask the authors to address.

Major comments:

1. My main concern about this study is the lack of discussion surrounding causal inference and endogeneity. The paper does not feature a discussion of the basis for the causal inference used in the paper. Are the authors claiming that the policy density is plausibly exogenous? Does the paper claim causality (and how would one avoid claims of reverse causality with higher emissions increasing pressures on governments and then leading to more policies)? Or is this a descriptive assessment? This is especially relevant for wording such as "attributing" different effects as well as for the headline statements that claim that those policies have reduced emissions by such a huge amount. Could those policies be correlated with general economic development or other variables and dynamics? Furthermore, a number of co-variables are likely to be highly correlated (see discussion below with regards to the Energy Ministerial), which is also not discussed. I would expect the authors to discuss questions like these explicitly and to clarify the causality basis that they are making their claims by.
2. I am highly sceptical about the results concerning the Clean Energy Ministerial as well as the results concerning the presence of an energy ministry. The Clean Energy Ministerial is highly correlated with general institutions in a country (a government/society more focused on climate policy is much more likely to sign up to this). Here, as above, causality is in my mind difficult to claim – as even the direction of causality is unclear. Is the membership leading to more policy diffusion or is policy knowledge in a government leading to joining such an institution? The CEM also (in my experience) plays a negligible role in day-to-day policy-making but is often a publicity exercise. Similarly, the presence of an energy ministry is arbitrary in my view. Every government (that I know of) has a ministry dealing with energy issues, even if it is paired with other dossiers and independent of whether "Energy" appears in the name of a ministry (as I gather from their source Tosun 2018). This is also not a static component – consider for example the German vs the UK case where in Germany the word "Energy" was dropped from the name of the department because climate was added to the ministries responsibility. In the UK Energy remained in the name, but the responsibilities of the ministries changed from BEIS, to DECC, to DENSY – does Germany deal with Energy less than the UK? I am highly sceptical of those singular dummy variables and would hence suggest to remove them (at the very least from the main text). The same applies to the IEA membership (especially the reverse causality argument) but it is more sensible than the other two.
3. I notice that the authors have used a dummy for EU membership – but it is unclear to me how the authors have dealt with EU policies on the policy density for EU member countries. Please clarify.
4. In the models displayed in Table 2, the authors never consider a full model (i.e. adding multiple variables into the model rather than just one variable at a time and then removing it again). This is especially odd as their individual models suggest that the additional variables matter – hence this is suggesting that each individual model (and the coefficients of the individual variables) suffer(s) from omitted variable bias. I would recommend to show a full model (at least adding the dummy variables if adding all HHI indicators does not make sense).

Minor comments:

5. In line 860 the authors state that the residuals are iid. Is this tested? Or assumed? Given the lack of testing for autocorrelation, cointegration and other common empirical issues, I would assume that this is their assumption (similar in S1 line 161).
6. In their cumulative policy density measure, do the authors also account for discontinued policies (e.g. the Australian carbon pricing scheme that only ran for a few years)?
7. In Figure 1, I'm unclear what the ranges left and right of the schematic mean. I would suggest to clarify this.
8. In Table 2, I would recommend adding a row at the top with the dependent variable. From what I gather from the note below the table it is always the same (CO2 intensity) but the headings on first reading suggest they are different. I would suggest to add a row clarifying the dependent variable.
9. Unless I have missed it, I would recommend to add specific equations and further explanation (including maybe examples) for the calculation of the HHI in the S1. This would help comprehension of the HHI variables.
10. The two paragraphs between line 60 and 62 seem disjointed. I am unclear whether the studies cited in line 60 are supporting the statements in line 62, especially as they are not cited in that paragraph anymore.
11. Line 73 suggests the authors have built the underlying policy database while in the methods it seems that the authors have used the existing Policies and Measures database from the IEA. Perhaps it would be good to clarify in the main text that additional analysis went into the database.
12. In line 856 the subscript i should likely not be capitalised.

Reviewer #3

(Remarks to the Author)

The authors investigate the effectiveness of policies in reducing CO2 emissions. Based on a panel dataset of 43 countries

2000-2019, the authors test whether the number of policies, specialization on instrument types, sectoral coverage, emission reduction targets, and government institutions affect CO2 emission intensities.

In my opinion, this is a promising manuscript, to which I have the following comments to give:

1. As the title and the beginning of the discussion rightly suggest, the ultimate goal of climate policy is to reduce the absolute level of emissions, not intensities. Accordingly, it would be critical for the authors to look at the level of emissions as outcome variable as well. This could contribute valuable insights into possible rebound effects of climate policies. If policies incentivize efficiency improvements, possibly in connection with an economic stimulus, policies could increase emissions despite relative decoupling of GDP from CO2 emissions. Also a possible interaction effect of policies and absolute targets on absolute emissions would be interesting to see. Any relatively time-invariant country characteristics (size, population, institutions, ...), that affect the average level of emissions and GDP, are absorbed by the unit fixed effects and are thus not a valid reason to only investigate intensities.

2. While I think the fixed effects model specification generally is a good choice, the control variables could be problematic in the sense of "bad control". For instance, one way to achieve a lower GDP CO2 intensity is to shift economic activity to the service sector while offshoring polluting manufacturing. This means climate policies could affect both the control (service value added, imports, ...) and outcome variable (emissions). I would like to see a parsimonious specification (with only the fixed effects and policy measures) as a baseline and a more detailed discussion of possible mechanisms.

3. The authors state that there are no significant differences between instrument types (regulatory, economic, voluntary) but I was not able to find the corresponding results. It would be surprising if voluntary instruments were as effective as mandatory types.

4. How common is the exit of policies in the data? When governments change so often does their ideological stance. For instance, Trump tried to undo environmental policies of the Obama administration. Is this relevant for the results?

Version 1:

Reviewer comments:

Reviewer #1

(Remarks to the Author)

The paper has improved a lot, and I think the paper presents interesting insights in the impact of policies. I think the authors have addressed the many comments in a good way with a very detailed response. Although for some comments they were not able to address those, but this was due to good reasons as they indicated.

I think one of my main omission of the analysis was policy stringency is missing, because the number of policies is not a good measure of the quality and GHG impact of climate policies. The authors have included literature that showed that policy stringency and policy density are also strongly correlated, so I think well addressed.

The work is original, but soon after their submission Stechemesser et al. published a similar analysis in Science. I asked for a comparison with Stechemesser et al. , but I think this is still not done yet. In their response to my reviewer's comments they mentioned that "Stechemesser et al identify total emission reductions in the range 0.6-1.8 GtCO2 from their set of 69 policy-attributed structural breaks in emission trends. This is an order of magnitude smaller than the estimate of 27.5 GtCO2 reductions in this study across all policy portfolios including continuously incremental reductions as well as major discontinuities."

I was surprised to see this difference. However, the authors did not discuss the much lower reduction found by Stechemesser et al in their paper.

Action required: It would be good if the authors would discuss this major difference, and the possible reasons, as the reader may expect a comparison with Stechemesser et al., as this study has a similar research objective. The comparison could also move to the SI, and only highlight the main conclusions in the main text.

(Remarks on code availability)

Reviewer #2

(Remarks to the Author)

I would like to thank the authors for their efforts in revising and improving this study. I feel that the study has substantially improved. I would like to highlight that the authors have considerably improved their discussions of endogeneity and attribution in the supplementary part of their paper. I think this discussion was necessary and has benefitted the overall paper.

I appreciate the efforts by the authors to defend their attribution claims in the study and indeed I believe there is a good discussion (mainly the Granger causality testing) surrounding the temporal setting that policies are antecedent to emission reductions.

However, I remain critical of the causal validity of the counterfactual construction and the headline number of cumulative reductions of 27 GtCO2. I do not believe that simple precedent from another study is enough to justify the term "attribution". I remain partly convinced that the policy density measure is more a representation of how liberal a society is and how important climate change is for society – but I'd be very careful to translate their results into policy advice. Maximising policy

density is clearly not what we should be doing – we should carefully design a small number of (mixed) effective policies with clear aims and effect pathways rather than aim to simply increase policy density.

At the very least, I would recommend to the authors to further review their language in lines 88, 100 and 295 and to directly compare their headline statements with Stechemesser et al. in the main text, as they do in at the bottom of page 4 of the rebuttal to reviewer 1.

I would also suggest to clarify that their results do not challenge the validity of the Tinbergen rule – but rather that their results indicate that policy-makers are not actually adhering to it. As currently framed, they seem to suggest that their results show that the Tinbergen rule is not valid.

I would further remove the implicit reference to the EKC in line 144 – there remains substantial academic debate around the presence of an EKC in carbon emissions, so I would rephrase that element.

I recommend to avoid using the abbreviation “IV” in Table 1 for independent variables – this in econometrics generally refers to an instrumental variable and could be confusing.

I would also ask the authors to show the full numbers in the two rightmost columns in Table Methods 2.

(Remarks on code availability)

Reviewer #3

(Remarks to the Author)

The authors have responded to my comments and mostly implemented appropriate changes in the manuscript. I still have some major comments about the identification strategy and interpretation of results.

1. The selection of control variables is not sufficiently justified and could substantially bias the coefficient of interest, yielding misleading results, even if well intentioned. In response to my previous concern about bad control, the authors provided to additional models ED16 in which they separately exclude the service and import shares. In S11 Section 2.9, the authors state that the estimate of beta1 "remains virtually unchanged" when omitting these variables. This is true for the service share but without the import share the point estimate of beta1 decreases by 35% and notably increases in variance. In my opinion, this is a relevant change that suggests a mediating role of imports.

A rigorous engagement with the functional relationships of the included variables is still missing, possibly in the form of a directed acyclic graph (Cinelli et al 2024). As parsimonious specification in a two-way fixed effects setting, I would expect models with only the explanatory variable of interest, especially since the FE already absorb a very large share of the total variance in the outcome (~70% in this case).

1.1 There are some variables that are possible mediators, ie variables that could be affected by the treatment and in turn can affect the outcome, namely GDP, the service share, and the import share. I would like to see these variables regressed on policy density, including also lagged density since adjustment to supply chains and sectoral composition likely take several years. If policy density affects these variables, then they are bad controls, bias beta1, and should not be included (Angrist and Pischke 2009, Cinelli et al 2024).

1.2 There are some variables which are basically time-invariant and are already accounted for by the unit FE, eg institutions. In that sense, the standard deviation reported in supplementary table 1 is misleading because it does not reflect the variance that is actually used to estimate the models. It would be helpful to report also the within SD here, ie the SD of the FE residuals (Mummolo and Peterson 2018). I would assume that almost all variance in the variables related to institutions and economic structure is absorbed by the FE.

1.3 Temperature varies randomly, given the unit and time FE. Regressing temperature anomalies on policy density should yield an insignificant effect which might be a good placebo test. As such it is no confounder / omitted variable and it is neither necessary to include it as control nor does including it bias the estimate of beta1.

1.4 Since no track changes are provided, it is not immediately obvious what the authors changed in the manuscript. However, I was not able to find any discussion of possible mediators in the main text (besides the moderating role of targets).

2. Related to 1.2, in the current presentation of the results it is not clear how large the effect sizes are. Currently, the interpretation seems to be "An increase in policy density by 1, decreases emissions intensity by 0.05%". Is this a practically relevant effect size? Is a marginal change by 1 in policy density still plausible after partialing out the the FE? It would be very helpful to report also estimates that are standardized using the within SD as suggested by Mummolo and Peterson (2018). This way all variables would be on the same scale and the reader could easily assess the relevance of the effect sizes.

2.1 In response to my previous comment, the authors provide interesting additional estimates by type of instrument in ED13. Here it is striking that the effect legal interventions is MUCH larger than any of the other types (eg 100 times larger than of market-based economic instruments). Some other instruments do not significantly affect CO2 intensity at all. This seems like the effect could be driven entirely by legal instruments. Is this the case, ie is the coefficient still significant when omitting legal instruments from the policy density index?

2.2 It would be less confusing to present the regression results in Table 1 in a standard format (models in columns, not rows). The caption should explain how to interpret the coefficients (as in line 663ff). Also it could be clarified here what is included in the 3 types and 8 types.

3. The link to the replication materials does not work for me so I cannot comment on the availability of the materials.

References

Angrist, J. D., & Pischke, J. S. (2009). *Mostly harmless econometrics: An empiricist's companion*. Princeton university press.

Cinelli, C., Forney, A., & Pearl, J. (2024). A crash course in good and bad controls. *Sociological Methods & Research*, 53(3), 1071-1104.

Mummolo, J., & Peterson, E. (2018). Improving the interpretation of fixed effects regression results. *Political Science Research and Methods*, 6(4), 829-835.

(Remarks on code availability)

Link not working

Version 2:

Reviewer comments:

Reviewer #1

(Remarks to the Author)

The new manuscript has successfully addressed the comments that I identified related to the comparison of Stechemesser et al, resulting in a much stronger and more robust paper.

I want to thank the authors for detailed replies and the comprehensive additional work they have done to address both referees' comments. The replies and additional sensitivity analysis have made clear that the main messages coming from the analysis are robust enough for this to be a meaningful contribution.

(Remarks on code availability)

Reviewer #2

(Remarks to the Author)

I would like to thank the authors for their consideration of my comments and would like to congratulate them for their important and insightful addition to the literature.

I have no more concerns and recommend acceptance.

(Remarks on code availability)

Reviewer #3

(Remarks to the Author)

The authors have comprehensively addressed my comments and I have no further comments to give.

(Remarks on code availability)

Climate policy portfolios that accelerate emission reductions

Revised submission: Jan 2025

Response to Reviewers.

We respond in detail to all the reviewers' comments below with a summary of the main changes in our revised submission. Prompted by the reviewers' recommendations and concerns, we have both extended the time frame of our analysis from 2019 to 2022, rerun all our models, and conducted new analyses along multiple dimensions including absolute emissions as dependent variables, policy stringency indices, subsets of instrument type, alternative diversity indices, and both full and more parsimonious model specifications. We have also conducted endogeneity and causality tests to strengthen our attribution of emission intensity reductions to policy accumulation rather than vice versa. We have updated all figures and tables with these new results, accompanied by full discussion in new sections of SI1, and summary explanations in main text and Methods. The revised submission is significantly strengthened as a result, and we appreciate the reviewers for encouraging this.

REVIEWER COMMENTS

Reviewer #1 (Remarks to the Author):

R1.1 The study and its methodology are transparent and well written. I also like the presentation of national figures in the paper and the thorough analysis of national projections in the supplementary information. The study finds quantitative evidence that national climate policy portfolios that specialise in certain instrument types and sectors are associated with faster reductions in fossil CO2 emission intensity. The link with policy instruments is particularly interesting. This study builds on the methodology of Eskander and Fankhauser (2020), but the authors have extended it to a longer time horizon (2000-2019) and cover more policies. The in-depth country analysis of this study is also an added value compared to the 2020 study. I think the work could be important in this area, although a better presentation in a chart of the main instrument types and sectors associated with faster reductions in fossil CO2 emission intensity for a selected group of countries would be welcome. As it is, the main policies are included in the figures, but this is widely known.^[SEP]

Response:

We tried lots of different figure options to combine the instrument types and sectors for select countries with the vignettes on policy density & policy milestones against aggregate CO2/GDP trends shown in our original Fig 2. All the options we tried became overly crowded with different dimensions in the data, so we have updated our original Fig 2 to the extended 2000-2022 timeframe, and included additional country Figs in SI2 visualising the complement data on instrument type shares and sectoral coverage shares. We also provide the underlying data for all countries (including those in the vignettes) in Table ED11. An example of these new country Figs in SI2 is shown here for the US.

Figure S4. Instrument types and changes in emission intensity per sector in the United States.

Note: Based on policy data from the IEA Policies and Measures Database (PMD) and carbon emission data from the IEA CO2 Emissions From Fuel Combustion Database.

Main changes in revised submission:

- new Fig in SI2 with instrument type and sectoral shares
- updated Fig2 in main text to 2022

R1.2 The disadvantage of the method is that it still uses the period that is not very recent, as it misses the last five years where even more policies have been developed, as we also see in the figure, as it misses Fit-for-55 for the EU and IRA for the US. I can understand why you did not focus on the period until 2022 because of the impact of COVID, but the current emissions trend for some countries with many climate policies are lower compared to 2019 levels (Canada, US, EU).^{[L]_{SEP}}

Response:

We have updated all our independent, control and dependent variables to 2022, and re-run all our models (including additional robustness checks) for the period 2000-2022.

(Some of our independent and control variables - sectoral CO2 emissions and climate variability - are not yet available for 2023 so we were unable to include this additional year, except in lagged models of our dependent variable CO2/GDP).

All our findings reported originally for the period 2000-2019 hold in the extended model for 2000-2022. In some cases our original findings are strengthened as the reviewer suggested given recent climate policy activity in the US and elsewhere. Indeed, extending the policy count time series sees the US overtake China for the largest number of policies in 2022, capturing the IRA-related policy activity under the Biden presidency. 41 instruments in our extended policy instrument database are directly linked to the IRA (see also IEA policy report here). The effect of the EU's Fit-for-55 strategy is also evident but less marked - we think because it was proposed in 2021 but not passed till 2023, and was also a more cumulative strengthening of a large pre-existing (prior to 2021) policy corpus in EU countries.

In the revised manuscript, we have updated all our model results (main text, methods, SI1, Extended Data tables) and our policy vignettes (main text, SI2) for the period 2000-2022. Our extended main model to 2022 now shows the cumulative CO2 emissions avoided from the adoption of all policy portfolios in our 43 country sample is 27.5 GtCO2 (up from 15.9 GtCO2 in our original model to 2019).

Covid impact in 2020:

The major impact of Covid-related disruptions in 2020 (and to a lesser extent in 2021) affect both total CO2 emissions and GDP in our dependent variable, so our extended models are broadly robust to the Covid discontinuity in emission trends.

The one exception is in our HHI diversity measure of policy sectoral coverage when this is weighted by sectoral emissions. In our original model to 2019, the emission-weighted HHI diversity measure had a significant negative coefficient, in line with expectations: policy portfolios weighted towards high emitting sectors are more effective (*ceteris paribus*). In our extended model to 2022, this coefficient becomes non-significant. We think the reason is that the sectoral shares of emissions in 2020 deviates from the longer-term trend, for example, due to the drop in transport sector emissions under lockdown-related travel restrictions. We include the country data for this effect in Table ED11. The main diversity index we used - the HHI - is sensitive by design to anomalous changes in sectoral shares. This causes the HHI diversity measure of emission-weighted policy sectoral coverage to drop in significance (shown in Table ED3a).

In further robustness testing, we find that the alternative Shannon Index diversity measure which is less sensitive to anomalous changes in sectoral shares remains negative and statistically significant in both the original 2000-2019 model and the extended 2000-2022 model (shown in Table ED3b). We have included this as a new section of SI1.

Main changes in revised submission:

- all model results, tables, figures & policy vignettes updated for period 2000-2022
- additional tests for effect of Covid 2020 anomaly in main text and Table ED3a (sectoral coverage HHI for 2000-2019 for comparison) and Table ED11 (transport sector share of emissions pre- and during 2020).

- additional robustness test using alternative Shannon Index as a diversity measure in SI1 and Table ED3b

R1.3 There is also an interesting paper in *Science* on climate policies that have achieved large emission reductions: Global evidence from two decades, which looks at 1500 climate policies implemented between 1998 and 2022 in 41 countries across six continents. It would be good if the authors would also compare some of their results with this study, although I can understand that this study was published after your submission.

Response:

Indeed, the Stechemesser et al. paper came out in *Science* a few weeks after we submitted our manuscript and is a clear new reference point in the policy attribution literature. They use a difference-in-difference (DiD) approach to evaluate the effect of combinations of policies in 41 countries over the period 2000-2020. Their country sample and time period are very similar to ours, but there are three important differences with their aim and methodology which mean our study is non-overlapping but strongly complementary.

First, their aim is to evaluate policies responsible for major near-term reductions in emissions. Consequently their dependent variable is structural breaks - statistically identifiable discontinuities - in sectoral emission trends.

Second (and as a consequence of this aim), their methodology uses difference-in-differences testing to attribute the discontinuity to specific combinations of policy instruments introduced or tightened in the preceding 2 years in the corresponding sector. The difference-in-differences approach allows them to demonstrate causality more robustly from policy instrument combinations to discontinuous reductions in emissions. They find policy instruments as part of a mix are generally more effective than those implemented alone, pointing to the importance of complementarity in portfolio design. (Taxation is an exception that works well as a stand-alone policy). They also find that effective policy mixes by country and by sector so caution against one-size fits all prescriptions.

In contrast, our approach using fixed effect models fitted to panel data is designed to identify the *cumulative* (non-discontinuous) effect of national policy portfolios - as opposed to the discrete and immediate effect of specific policy instrument combinations in the specific cases they identify in which these cause major reductions in emissions. So whereas Stechemesser et al. evaluate the near-term effect of particular combinations of sectoral policy in 69 specific cases of emission discontinuity, our approach evaluates the longer-term effect of policy portfolios in general cases of emission reduction trends.

This cumulative focus of our approach using national policy portfolios also allows us to test and demonstrate the interacting effect of framework targets and organisations that play out over the longer-term. This is a further complementary novelty of our work alongside Stechemesser et al.

It is interesting to note that they identify total emission reductions in the range 0.6-1.8 GtCO₂ from their set of 69 policy-attributed structural breaks in emission trends. This is an order of magnitude smaller than our estimate of 27.5 GtCO₂ reductions across all policy portfolios including continuously incremental reductions as well as major discontinuities. This points to the

importance of combining both approaches to policy attribution capturing both ‘big bang for your buck’ immediate effects of specific policy combinations and ‘slow burn’ cumulative effects of policy portfolios.

In this context, we also note that policy evaluation studies are designed to identify *ex post* the effect of specific policies on emission reductions as well as other policy objectives. We refer to these in our discussion section. We have extended this discussion to include a new paper by Döbbeling-Hildebrandt et al. published in *Nature Communications* after we had submitted our original manuscript. They meta-analyse 80 *ex post* evaluation studies of 21 carbon pricing schemes worldwide to generalise the (de-biased) effect of this specific type of policy instrument to the order of -4% to -15% emission reductions.

Main changes in revised submission:

- updated literature review (intro & discussion), with links to Stechemesser et al. findings on on policy mixes, and on sector-specific policy combinations, and with links to Döbbeling-Hildebrandt et al. findings on evaluating specific policies

Stechemesser, A., N. Koch, E. Mark, E. Dilger, P. Klösel, L. Menicacci, D. Nachtigall, F. Pretis, N. Ritter, M. Schwarz, H. Vossen and A. Wenzel (2024). "Climate policies that achieved major emission reductions: Global evidence from two decades." *Science* 385(6711): 884-892. doi:10.1126/science.adl6547

Döbbeling-Hildebrandt, N., K. Miersch, T. M. Khanna, M. Bachelet, S. B. Bruns, M. Callaghan, O. Edenhofer, C. Flachsland, P. M. Forster, M. Kalkuhl, N. Koch, W. F. Lamb, N. Ohlendorf, J. C. Steckel and J. C. Minx (2024). "Systematic review and meta-analysis of ex-post evaluations on the effectiveness of carbon pricing." *Nature Communications* 15(1): 4147. 10.1038/s41467-024-48512-w

I would recommend publication if the authors adequately address the main comments listed below:

R1.4 [17] The main difficulty I have is with the intensity of climate policy: It is obviously difficult to measure the stringency and GHG reduction impact of all types of climate policies. Nevertheless, the lack of a measure of policy stringency is a major limitation of this paper, because the number of policies is not a good measure of the quality and GHG impact of climate policies. Countries may have many policies and a high policy density, but this does not guarantee that they will reduce CO2 emissions. For example, a single strong national policy based on an ETS with nationwide coverage of all GHGs could be much more effective than multiple policies that have not reduced emissions at all in the end. It all depends on the quality and stringency of the policies and their coverage across sectors to determine the GHG impact. I therefore suggest that the issue of stringency and the GHG impact of policies should be addressed in some way. I assume that the earlier work by Eskander and Fankhauser also addressed this issue.

Response:

We agree that not accounting for policy stringency is an important omission. The Eskander & Fankhauser paper on which we build also did not account for policy stringency. We had noted the importance of further work on stringency in the discussion of our original manuscript, but we appreciate the reviewer pushing us on this as we have been able to make progress using an OECD dataset.

Quantitative stringency measures are not commonly compiled due to difficulties in standardising stringency assessments across heterogeneous policy contexts. To the extent they are available, this is typically only for particular instruments, particularly carbon pricing for which the effective \$/tCO₂ price is a comparable measure across jurisdictions and pricing instruments including taxes and emissions trading schemes (World Bank 2023, see also <https://carbonpricingdashboard.worldbank.org>; and an alternative ‘Real Carbon Price Index’ published by Monash University, see also <https://www.realcarbonindex.org/indices>).

The only measure of policy stringency we could find is the OECD Climate Actions and Policies Measurement Framework (CAPMF) by Nachtigall et al. (2022). CAPMF provides one index on the stringency of climate policies, with sectoral sub-indices (including for buildings, industry, transport, and electricity supply). The index measures the degree to which sectoral climate actions and policies incentivise or enable GHG emissions mitigation.

The timeframe of the CAPMF is 1990-2022, matching our extended model period, but the OECD country sample does not include Brazil and the United States. Additionally, the policies covered by CAPMF do not include many of those coded as ‘voluntary’ or informational in our sample.

Policy stringency and policy density are also strongly correlated. This was demonstrated by Schaffrin (2015) with their novel measure of policy ‘strictness’ for three countries’ climate policies. The OECD have recently confirmed this with the CAPMF as their own stringency measure - see, for example, Fig 3.2 in the OECD’s report on CAPMF (Nachtigall et al. 2022). As a country example for the UK, Fig 3.5 shows that stringency increased with the adoption of more policies over time. The policy density variable in our sample without US and Brazil has a 71% correlation with the CAPMF.

So for these three reasons - correlation with policy density, mismatch in policy instrument scope (misspecification), mismatch in geographic scope - we limit our use of CAPMF to robustness testing our main models (reported in SI1 and referred to in main text and methods).

As we cannot include the policy stringency index in our models alongside policy density due to collinearity, we use the OECD CAMPF as an alternative independent variable. Specifically, we take the mean of the sectoral policy stringency indices from CAMPF weighted by each sector’s share of total emissions. (This is similar to our procedure in constructing emission-weighted diversity indices in our testing of policy sectoral coverage).

We show the full results in Table ED12. Like policy density, the coefficient for the weighted sectoral policy stringency index is negative and statistically significant at $p < .01$. This shows that a more stringent climate policy corpus is associated with faster reductions in emissions intensity.

Despite the limitations noted for the stringency index including its exclusion of voluntary instruments as well as Brazil and the United States, we find it performs very well, offering complementary insights to our main finding on the effect of climate policy density on emission intensity.

Main changes in revised submission:

- new SI1 on policy stringency model with links to full model results in Table ED12

- summary of findings and revised discussion in main text

IEA (2024). *Energy Efficiency 2023*. Paris, France, International Energy Agency.

Nachtigall, D., Lutz, L., Rodríguez, M. C., Hašič, I., & Pizarro, R. (2022). The climate actions and policies measurement framework: A structured and harmonised climate policy database to monitor countries' mitigation action. (OECD Environment Working Papers, Issue No. 203). <https://doi.org/10.1787/2caa60ce-en>

Schaffrin, A., S. Sewerin and S. Seubert (2015). "Toward a Comparative Measure of Climate Policy Output." *Policy Studies Journal* 43(2): 257-282. <https://doi.org/10.1111/psj.12095>

World Bank (2023). *State and Trends of Carbon Pricing 2023*. Washington, DC, World Bank. <http://hdl.handle.net/10986/39796>

R1.5 [11] also don't quite understand why the authors use policy intensity in terms of numbers, as this does not take into account total GHG emissions, nor sectoral coverage, and compare it to a country's emissions intensity (i.e. nationwide emissions). I would expect policy density to also be relative to the size of a country's emissions, as well as sectoral coverage. Perhaps you could elaborate on this.

Response:

We follow the precedent of Eskander & Fankhauser (2020) in using policy density (cumulative total number of policies) as our main independent variable, building off wider political science literature on policy density (Shaffrin 2015) as well as studies of policy accumulation and policy sequencing (Pahle et al. 2018, Nascimiento et al. 2023). This literature emphasises policy density as a useful measure for explaining variation in countries' progress in reducing emissions over time.

By using CO₂/GDP rather than absolute CO₂ emissions as our dependent variable, we also account for variation in economic performance and country size - although in terms of the economy rather than emissions. In response to the reviewer's concern, we ran new models with absolute fossil CO₂ emissions as the dependent variable, with controls to account for variation in countries' development stage. For this, standard controls are log GDP per capita and its interaction with the square term (as the relationship between development stage and emissions is non-linear), but we also ran additional models with log GDP or log population as direct controls of country size. These new model results are reported in full in Table ED15, and are in line with expectations: higher policy densities are associated with faster reductions in CO₂ emissions, controlling for variation in country size and development stage. (For full discussion of these results with CO₂ emissions as the dependent variable, see R3.2).

In terms of sectoral coverage, all 43 countries in our sample have policies targeting emissions in all six emitting sectors. We show the sectoral share data in full in Table ED11. (There are two exceptions: Iceland has no climate policies in the energy industry sector; Switzerland has no climate policies in the agricultural sector). So it's not possible to weight policy density by sectoral coverage. However, we do account for the concentration of policies on higher emitting sectors in our emissions-weighted diversity measure of policy sectoral coverage. As the reviewer suggests, national policy portfolios concentrated on sectors accounting for a higher share of total emissions are associated with faster reductions in emission intensity. These model results are reported in full in Tables ED3a and ED3b, and summarised in the main text and methods.

To test this sectoral effect further in response to the reviewer's concern, we ran further models for each sector (except energy industry) using sectoral CO2 emissions normalised by total GDP as the dependent variable, and sectoral policy counts as the independent variable. For each model, we also test sectoral policy stringency (using the sectoral CAMPF index) as an alternative independent variable - see response R1.4 for details on the CAMPF). This approach is limited in that the GDP normalisation is for the whole economy not the sector (for which sufficient time series data were not available to match our country x time panel). However, all the sectoral model results show negative and significant coefficients for policy density and for the alternative policy stringency variable as expected, and consistent with the main economy-wide results. The one exception is that policy density in the buildings sector model is negative but not significant. As the alternative specification using policy stringency does show a significant coefficient, we think this exception is a model artefact due to reduced heterogeneity as some smaller countries have very few buildings sector policies. We report these sectoral model results in full in Table ED18.

We also interpret the reviewer's concern as being about endogeneity - that higher emissions lead to more policies (in order to tackle those emissions). We respond in detail to endogeneity concerns in R2.1 below. A summary of our response is that we combine lagged models, new endogeneity and causality tests, model selection, and grounding in literature to substantiate our arguments that policy density drives reductions in emission intensity, not the other way round. However, with our fixed effect panel model we are unable to conclusively rule out endogeneity concerns. (See R2.1 for full discussion).

Main changes in revised submission:

- new model results for absolute CO2 emissions as dependent variable in Table ED15, with new section in SI1 discussing the results
- new model results for sectoral policies in Table ED18, with summary of findings and links to model results in Methods and SI1

Eskander, S. M. S. U., & Fankhauser, S. (2020). Reduction in greenhouse gas emissions from national climate legislation. *Nature Climate Change*, 10(8), 750-756. <https://doi.org/10.1038/s41558-020-0831-z>

Nascimento, L., M. den Elzen, T. Kuramochi, S. Woollands, I. Dafnomilis, M. Moisisio, M. Roelfsema, N. Forsell and Z. Araujo Gutierrez (2023). "Comparing the Sequence of Climate Change Mitigation Targets and Policies in Major Emitting Economies." *Journal of Comparative Policy Analysis: Research and Practice*: 1-18. [10.1080/13876988.2023.2255151](https://doi.org/10.1080/13876988.2023.2255151)

Pahle, M., D. Burtraw, C. Flachsland, N. Kelsey, E. Biber, J. Meckling, O. Edenhofer and J. Zysman (2018). "Sequencing to ratchet up climate policy stringency." *Nature Climate Change* 8(10): 861-867. [10.1038/s41558-018-0287-6](https://doi.org/10.1038/s41558-018-0287-6)

Schaffrin, A., S. Sewerin and S. Seubert (2015). "Toward a Comparative Measure of Climate Policy Output." *Policy Studies Journal* 43(2): 257-282. <https://doi.org/10.1111/psj.12095>

R1.6 I think the two tables are not always very attractive to the Nature Communications audience, especially those with many blank cells. A better presentation of Table 2 would be welcome.

Response:

We have moved the original Table 1 with the descriptive statistics for the variables in our model to the Methods section as Methods Table 1. We think it's important to keep this table visible as the underlying basis for our model results, even though it's not the most attractive.

We have revised the original Table 2 with our model results to remove the blank spaces, though our previous format followed the convention when reporting econometric model results. We hope the reviewer agrees this new format is visually more acceptable. The revised table is now Table 1 in the main text.

Main changes in revised submission:

- original Table 1 descriptive statistics moved to Methods
- new format for original Table 2 model results

R1.7 It would also be good if the authors could present the most effective policies according to their analysis for the major emitting countries.^[1]

Response:

Our study aim and approach is concerned with the cumulative effect of climate policy portfolios, including through the interaction with targets and governmental organisations. Although our models report clear and strong findings in each of these aspects, our approach is not designed to identify the most effective policies *within* each national portfolio.

This would require a difference-in-difference (DiD) method in which each policy is analysed as a discrete intervention causing emissions to diverge from counterfactual without-intervention trends. Indeed this is the method used by Stechemesser et al. in their recent *Science* paper for a set of 69 cases in which they had identified structural breaks in emission trends. From there they used DiD methods to identify which policies or combinations of policies in the previous 2 years could be attributed as causing the break (see R1.1 for full discussion). Similar approaches using DiD are used in policy evaluation studies, but focused on specific policy interventions which are the instrumental variable necessary for DiD. However, these methods would not be consistent with our study aim.

Our detailed policy vignettes for select countries provide a narrative account of policy accumulation that include discussions of effectiveness given each country's institutional context, based on literature. (The vignettes are summarised in Fig 3 in main text, and shown in full in SI2, updated to 2022 in this revised submission). However, we are reluctant to use these narratives to make strong claims about specific policies which we can not support in our modelling.

In our response to R3.6, we explain in full how we ran additional models for policy density by instrument type in order to see whether we could identify which instrument type was more effective in general terms (as opposed to in specific countries). We ran a separate model including controls for policies coded in 1 of the 8 instrument types:

- *regulatory*: legal interventions, monitoring, and regulations
- *economic*: market-based and direct provision
- *voluntary*: planning, informational, and other voluntary

Full results are shown in Table ED13. Given that our dependent variable is emissions intensity (CO₂/GDP), we did not expect to find a clear effect given we were including as our independent variable only a subset of all policies (coded as being of a particular instrument type). However, 5 of the 8 models did show negative and significant coefficients for the policy density by instrument type variable, with the largest coefficient in absolute magnitude being for legal interventions and monitoring, both of which are categorised as ‘regulatory’.

However, given the general limitation of these models including only a subset of policies in trying to explain economy-wide emission intensity, we are reluctant to over-interpret this result.

Main changes in revised submission:

- new model results for policy density by instrument type in Table ED13
- clarifying text in Methods and discussion on study aim and limited ability to evaluate effectiveness of specific policy instruments in specific countries

R1.8 Please use the Nature communications citation and references format.^[1]

Addressed in the revised submission.

Specific comments:^[1]

R1.9 You mention twice: "National climate policy portfolios that specialise in certain types of instruments and sectors are associated with faster reductions ...". It would be good to give an example of what you mean by certain?^[1]

Response:

This was unclear wording on our part which we have corrected. Our model results using diversity measures of policy sectoral coverage and policy instrument type as additional independent variables alongside policy density show that in both cases specialisation (the opposite of diversification) is associated with faster reductions in emission intensity. This general finding holds across our 43 country panel over the extended 2000-2022 time period. Our emission-weighted models of policy sectoral coverage do show that policy portfolios weighted towards higher emitting sectors are in general more effective, but which sector these are varies by country and over time. In our additional testing of policy density by specific instrument types (see R1.7 above), we also show that legal, monitoring, and economic instruments are associated with reductions in emission intensity but we are reluctant to over-interpret these results given the limitations of explaining economy-wide emission trends using only a subset of policies.

Our study aim and modelling approach (discussed in detail in R1.3 and R1.7) mean we cannot specify further which instrument types or which sectoral coverage are most effective. This will vary by policy tradition and emission profile in each country, and is the focus of policy evaluation studies using a different methodology designed to answer this question - with the recent Stechemesser et al. paper an excellent example (see R1.3).

Main changes in revised submission:

- rewording of findings on specialisation to avoid misinterpretation

Reviewer #2 (Remarks to the Author):

R2.0 I would like to thank the authors for a very interesting and timely study on a very important topic. The question of the impact of individual policies and policies cumulatively is highly relevant and hence deserves academic attention. I have a number of concerns that I would ask the authors to address. Major comments:

R2.1 My main concern about this study is the lack of discussion surrounding causal inference and endogeneity. The paper does not feature a discussion of the basis for the causal inference used in the paper. Are the authors claiming that the policy density is plausibly exogenous? Does the paper claim causality (and how would one avoid claims of reverse causality with higher emissions increasing pressures on governments and then leading to more policies)? Or is this a descriptive assessment? This is especially relevant for wording such as “attributing” different effects as well as for the headline statements that claim that those policies have reduced emissions by such a huge amount. Could those policies be correlated with general economic development or other variables and dynamics? Furthermore, a number of co-variates are likely to be highly correlated (see discussion below with regards to the Energy Ministerial), which is also not discussed. I would expect the authors to discuss questions like these explicitly and to clarify the causality basis that they are making their claims by.

Response:

On wording, we agree it is ambitious to claim causality and we have made sure we do not do this in the revised submission. Generally we frame our analysis as showing associations between policy portfolio variables and emission intensity reductions, with literature supporting the inference that policies are antecedent. However we recognise that the term ‘attribution’ - the field of climate policy science to which our study contributes - does also imply causation. In the applied economics and political science literature, causal claims are only possible with Difference-in-Differences (DiD) methods which we discuss in R1.3 with reference to a recent *Science* paper using this approach. In our work, we follow precedent (e.g., Eskander & Fankouser 2020) in positioning our approach as an attribution study.

We have multiple lines of reasoning to support our attribution of emission intensity reductions to accumulating policy density rather than vice versa: (1) lagged models, (2) endogeneity tests, (3) Granger non-causality tests, (4) fixed effects, (5) weak correlations with covariates, (6) supporting literature and policy vignettes, (7) our choice of econometric model. We discuss each in turn.

(1) Lagged models: We examine the lagged effect of policy density on emission intensity. A significant lagged effect indicates that an increase in the stock of policies drives emission intensity reduction, rather than the reverse. We report a 3 year lagged model in Table ED2 showing a negative and significant coefficient for the policy density variable in line with expectations. We get broadly similar results for 1, 2, and 4 year lags.

However, we recognise that policy density does not change much over some time periods in some countries, so the lagged policy density variable could absorb the effects of other variables. In a further model reported in Table ED8 we include a 1 year lag on all controls as well as the policy density variable. Below we show that our controls are only weakly correlated with our policy density variable.

(2) Endogeneity tests: We ran standard endogeneity tests used in applied econometrics, specifically the Durbin-Wu-Hausman test, following Wooldridge (2003, p. 483-484). Full results are included in a new section of SI1 with model results in Table ED14.

As an instrument for policy density, we use the sectoral stringency index (OECD CAPMF discussed in full in R1.4) which we know from literature controls for a similar effect to policy density on emission intensity, also indicated by the high correlation between the two variables (71%) in our panel.

We start with the baseline regression model B1 (column 1, Table ED14) then run model B2 (column 2, Table ED14) in which emission intensity is replaced by policy density as the dependent variable, while simultaneously introducing the OECD CAMPF sectoral stringency index as an independent variable. We save the residuals from model B2 and add them in the baseline regression model (B3, column 3, in Table ED14) as an additional regression, where we observe that the coefficient for the model B2 residuals does not reject the null hypothesis that the coefficient $\delta=0$. As a final test, we perform an F-test on the coefficient for the model B2 residuals that fails to reject the null. This further shows that the specific regressor is not endogenous, giving us further evidence to reject the reverse causal explanation that emission intensity results in higher policy density.

We summarise these results here, and refer the reviewer to Table ED14 for the full models with controls.

Summary of Table ED14. Durbin-Wu-Hausman endogeneity test for policy density.

	Model B1	Model B2	Model B3
	Log(CO2/GDP)	Policy density	LOG(CO2/GDP)
Policy density	-0.000555***		-0.000436
	(0.000169)		(0.000569)
Stringency index		15.59***	
		(1.387)	
Model B2 residuals			9.38e-06
			(0.000617)
Controls	YES	YES	YES
Observations	941	898	898
Within R-squared	0.2729	0.3113	0.273
*** denotes statistical significance at 1%, ** at 5%, and * at 10%. Parentheses indicate standard errors for each coefficient.			

(3) Granger non-causality tests: As an additional sensitivity analysis of the relationship between total policy density and emission intensity, we specify Granger non-causality following Dumitrescu and Hurlin (2012). Results in the Table below show that the null hypothesis of Granger non-causality is rejected at 1% statistical significance level, a result that holds when we add a lag operator of order 3.

Summary Table of Dumitrescu-Hurlin (2012) test on whether policy density Granger-causes emission intensity reduction.

	Lag 0	Lag 3
--	-------	-------

	W-stat	Zbar-stat	P-value	W-stat	Zbar-stat	P-value
Policy density	6.5810	25.8779	0.0000	10.3218	19.6010	0.0000
Notes: Null hypothesis that policy density does not Granger-cause reduction in the natural logarithm of emissions intensity. Rejection of the null indicates there is Granger causality.						

We note that although the Dumitrescu-Hurlin test, like most statistical tests of causality, is useful for detecting potential temporal causal relationships between independent and dependent variables, it has limitations in establishing reverse causality for which arguments based on prior hypotheses and causal reasoning are important.

(4) Fixed effects: Empirically, policy density is affected by differing policy traditions across countries, which is captured in our models by the country fixed effects. Policy density also increases over time (shown in our SI1 plots per country) due to increasing issue salience and prioritisation as well as secular trends (e.g. economic development and strengthening policy capacity). This is captured in our models by year fixed effects. Both these model features help to avoid our findings being the result of spurious correlations.

(5) Weak correlation between co-variates: There is generally low or very low correlation between policy density and the control variables used in our models over the period 2000-2022. The highest correlation is 22.8% for services share of GDP; all others are below 20% (temperature variation 19.5%, imports share of GDP -16.6%, GDPpercap log 14.0%, rule of law 0.08%, Hodrick-Prescott GDP filter -0.01%). These are all substantially lower than the correlation with the OECD CAMPF sectoral stringency index (71%), which we use as an instrument to test for endogeneity. In general we find that policy stocks are not correlated with general economic development or other variables and dynamics.

We address correlation between policy density and our governmental organisation dummy variables including CEM in more detail in Response 2.2 below.

(6) Supporting literature and policy vignettes: Consistent with the literature, we are confident that policy density is indeed plausibly exogenous, and that our analysis goes beyond a descriptive assessment.

All the climate policy portfolio elements we include in our models are designed to reduce emissions in line with policy priorities post-Kyoto Protocol (1997) in many OECD countries and post-Paris Agreement (2015) in all OECD+BRIICS countries.

Our policy vignettes (see SI2) indicate that the implementation of new policies increasing the total stock is clearly antecedent to observed emission intensity reductions, rather than the other way round.

Stehenesser et al (2024) use DiD methods to show that combinations of policies lead to discontinuous reductions in sectoral emissions within two years of introduction. This shows causality from policies to emission reductions.

Like Eskander and Fankhauser (2020), our aim is different in assessing the path dependent effect of climate policy stocks on cumulative incremental emission reductions (with our additional testing of the interacting role of long-term targets and governmental organisations). We align with the methodological framework established by Eskander and Frankauser (2020) in the climate

attribution literature, and in our Methods and SI1 we show how we replicate their model specifications for comparison.

(7) Our choice of econometric model: Our choice of the two-way fixed effects (TWFE) estimator is grounded in evidence from the applied econometrics literature, that emphasises its capability in establishing causality. Influential studies, such as Angrist and Pischke (2009) in applied economics and Imai and Kim (2019) in political science, underscore the effectiveness of fixed effects models for identifying causal relationships in panel data. Millimet and Bellemare (2023) and Wooldridge (2021) provide a comprehensive overview of the econometric properties of the TWFE estimator, elaborating on its strengths and limitations in the context of causal inference with panel datasets.

We make these arguments to support our decision to use the TWFE estimator, incorporating cross-sectional and time dummies, given our study aim of assessing the overall impact of policy density on emission intensity. With all the controls in our model, we are confident the estimated effect on emission intensity is clearly attributable to changes in policy density.

As noted, the main alternative method which allows for robust causal claims is the Difference-in-Difference (DiD) model (Baker et al, 2022). As in the Stechemesser et al. 2024 study discussed in R1.3, the DiD approach allows researchers to estimate the average treatment effect by comparing outcomes between a control group and a treatment group subject to a distinct policy intervention.

However, given we examine the effect of accumulating policy stocks on emissions intensity, a similar approach focusing on distinct individual policies would not work. Directly comparing individual policies across countries would invoke concerns over robustness, given the fundamental differences in the policies' design, which are tailored to each country's specific characteristics and needs. Such comparisons would fall under the umbrella term of 'bad comparison' in which dissimilar policies are evaluated against each other. Baker et al. (2022) argue that when staggered timing of treatment effects (i.e. policies introduced at different dates) is combined with treatment effect heterogeneity (i.e. varying types of policies across different countries), DiD estimates become biased and consequently misleading. This is particularly true in the case of dynamic treatment effects (Sun and Abraham, 2021). For these reasons we do not consider DiD methods appropriate for our research focus on cumulative national climate policy portfolios.

In sum:

From these multiple arguments we are confident that both conceptually and methodologically our study goes beyond a descriptive analysis in being able to attribute emission intensity reductions to accumulating stocks of climate policies.

Main changes to revised submission:

- checked and removed any language on causation
- clarification of reasons for claiming attribution (main text, Methods, and SI1)
- new SI1 section on endogeneity & causality tests

Angrist, J. D., & Pischke, J.-S. (2010). The Credibility Revolution in Empirical Economics: How Better Research Design Is Taking the Con out of Econometrics. *Journal of Economic Perspectives*, 24(2), 3–30. <https://doi.org/10.1257/jep.24.2.3>

Baker, A. C., Larcker, D. F., & Wang, C. C. Y. (2022). How much should we trust staggered difference-in-differences estimates? *Journal of Financial Economics*, 144(2), 370-395. <https://doi.org/https://doi.org/10.1016/j.jfineco.2022.01.004>

Dumitrescu, E.-I., & Hurlin, C. (2012). Testing for Granger non-causality in heterogeneous panels. *Economic Modelling*, 29(4), 1450-1460. <https://doi.org/https://doi.org/10.1016/j.econmod.2012.02.014>

Eskander, S. M. S. U., & Fankhauser, S. (2020). Reduction in greenhouse gas emissions from national climate legislation. *Nature Climate Change*, 10(8), 750-756. <https://doi.org/10.1038/s41558-020-0831-z>

Imai, K., & Kim, I. S. (2019). When Should We Use Unit Fixed Effects Regression Models for Causal Inference with Longitudinal Data? *American Journal of Political Science*, 63(2), 467-490. <https://doi.org/https://doi.org/10.1111/ajps.12417>

Millimet, D., & Bellemare, M. F. (2021). Fixed Effects and Causal Inference. IZA Discussion Paper No. 16202. <http://dx.doi.org/10.2139/ssrn.4467963>

Nachtigall, D., Lutz, L., Rodríguez, M. C., Hašič, I., & Pizarro, R. (2022). The climate actions and policies measurement framework: A structured and harmonised climate policy database to monitor countries' mitigation action. (OECD Environment Working Papers, Issue No. 203). <https://doi.org/10.1787/2caa60ce-en>

Sun, L., & Abraham, S. (2021). Estimating dynamic treatment effects in event studies with heterogeneous treatment effects. *Journal of Econometrics*, 225(2), 175-199. <https://doi.org/https://doi.org/10.1016/j.jeconom.2020.09.006>

Wooldridge, J. M. (2021). Two-Way Fixed Effects, the Two-Way Mundlak Regression, and Difference-in-Differences Estimators. Available at SSRN: <https://ssrn.com/abstract=3906345>.

Wooldridge, J. M. (2003). *Introductory Econometrics: A Modern Approach*. (2nd Edition ed.). Cengage Learning. <https://doi.org/https://doi.org/10.1198/jasa.2006.s154>

R2.2. I am highly sceptical about the results concerning the Clean Energy Ministerial as well as the results concerning the presence of an energy ministry. The Clean Energy Ministerial is highly correlated with general institutions in a country (a government/society more focused on climate policy is much more likely to sign up to this). Here, as above, causality is in my mind difficult to claim – as even the direction of causality is unclear. Is the membership leading to more policy diffusion or is policy knowledge in a government leading to joining such an institution? The CEM also (in my experience) plays a negligible role in day-to-day policy-making but is often a publicity exercise.

Response:

This comment touches on several issues that warrant clarification. The first part refers to a correlation between general institutions in a country and CEM membership. It is true that CEM membership is voluntary and as such no country is forced to join it. It is also true the CEM membership indicates that climate action is considered salient in a country, but this is where the picture starts to become more nuanced.

Countries choose to join the CEM for different reasons: some join because they are already actively adopting and implementing ambitious climate policies, and others join because they recognise the need to do so (Tosun & Rinscheid, 2023). What the CEM does not accept is that its members make no effort to promote clean energy technologies. Becoming and remaining a member of the CEM therefore requires a certain level of commitment. This commitment does not imply the adoption and implementation of common clean energy policies or the development of more ambitious national policies. It means that countries participate in workstreams that they consider relevant to their clean energy transition (Tosun et al., 2023). Some of the workstreams

aim at improving the implementation (not the formulation) of domestic policies, but most of them are designed to facilitate public-private collaboration in the diffusion of clean energy technologies that can contribute to reducing GHG emissions.

As an example the Transforming Solar workstream focuses on the solar PV manufacturing value chain and the development of resilient supply chain capacity and linkages among trading partners. Consequently, CEM membership provides access to collaborations aimed at stimulating the diffusion of clean energy technologies that can help reduce GHG emissions (Carey and Yang, 2023). This effect may be mediated through domestic climate or clean energy policies, but it may also be a direct effect through technology diffusion.

For this reason, we consider it important to control for a potential effect of CEM membership in our models - *not as a main effect* but through an interaction effect with our main policy density variable. These are the results we report.

The reviewer is right that we can not rule out reverse causality (discussed in detail in R2.1 above), but by testing the interaction between CEM membership and policy accumulation we capture a pathway to GHG emission intensity reduction that exists in addition to the policy pathway based on the diffusion of clean energy technology.

In our original submission we were imprecise in this respect, and have clarified the text to refer to CEM's role in clean energy technology diffusion. We fully agree with the reviewer that CEM membership does not play a role in domestic policy formulation.

Given the reviewer's concerns about our CEM results, we have also relegated them to SI1 and Extended Data Table ED5. In the revised submission, we include the IEA results in main text (Table 2) as IEA membership is more directly related to policy exchange and diffusion (see R2.4 below).

Main changes to revised submission:

- removal of CEM interaction effect models from main text
- clarification in main text on CEM role in clean energy technology diffusion (not policy diffusion)
- new section of SI1 with discussion of CEM role in clean energy technology diffusion

Tosun, J., & Rinscheid, A. (2023). What drives engagement in the Clean Energy Ministerial? An assessment of domestic-level factors. *Journal of European Public Policy*, 30(3), 469-487.

Tosun, J., Heinz-Fischer, C., & Luo, R. (2023). Who takes the lead? A disaggregate analysis of the EU's engagement in the Clean Energy Ministerial and Mission Innovation. *Journal of Cleaner Production*, 382, 135240.

Carey, E., & Yang, X. (2023). From Paris to Glasgow and beyond: what future for clean energy technology deployment under Article 6?. In *A Research Agenda for Energy Politics* (pp. 127-154). Edward Elgar Publishing.

R2.3 Similarly, the presence of an energy ministry is arbitrary in my view. Every government (that I know of) has a ministry dealing with energy issues, even if it is paired with other dossiers and independent of whether "Energy" appears in the name of a ministry (as I gather from their source Tosun 2018). This is also not a static component – consider for example the German vs the

UK case where in Germany the word “Energy” was dropped from the name of the department because climate was added to the ministries responsibility. In the UK Energy remained in the name, but the responsibilities of the ministries changed from BEIS, to DECC, to DENZ – does Germany deal with Energy less than the UK? I am highly sceptical of those singular dummy variables and would hence suggest to remove them (at the very least from the main text).

Response:

This comment is useful for explaining the nature of our dummy variable coding for energy ministries and to elaborate on our rationale for including it in one of our models as an interaction with policy density.

From a theoretical viewpoint, the creation of dedicated ministries for certain policies is relevant for our analysis in two ways as demonstrated empirically by Klüser (2023). First, the creation of an issue-specific ministry creates new bureaucratic capacity both to formulate policies that fall within the policy domain of a given ministry and to implement those policies. Second, the creation of an issue-specific ministry indicates that an issue is salient to a particular government.

Issue salience tends to vary over time. A strong indicator that the design of government ministries is influenced by issue salience is the change in their names. Our data on energy ministries vary over time and show that in some countries there were episodes when there was no ministry with “energy” in its name. It is true that “energy” is sometimes replaced by “climate”, “energy efficiency”, “energy security” or some other formulation, but this also reflects the salience of energy-climate as a policy domain, both for formulation and implementation.

Following Klüser (2023), dedicated “energy” or “climate” ministries increase the chances that formulated policies are fully implemented, and so increase the impact of policies on emissions. Conversely, if a new government removes “energy” or “climate” from the name of a ministry this indicates a lower priority for the issue and is likely to reduce the capacity of the respective ministry to formulate and implement policy which may weaken the effect of the policy corpus on emission reductions.

To better capture variation in ministry naming over time, including the German case mentioned by the reviewer, in our revised submission we have replaced the energy ministry dummy with an energy and/or climate ministry dummy. This accounts for 50.6% of observations. We then reran the models (extended to 2022) to test the interaction between this dummy and policy density. In line with our expectations, the interaction term was negative and significant.

Main change in revised submission:

- replacement of energy ministry dummy with energy and/or climate ministry dummy
- rerun models with policy density X ministry interaction term - summarised in main text, and shown in full in Table ED5

Klüser, K. J. (2022). From bureaucratic capacity to legislation: how ministerial resources shape governments’ policy-making capabilities. *West European Politics*, 46(2), 347-373. <https://doi.org/10.1080/01402382.2022.2030602>

R2.4 The same applies to the IEA membership (especially the reverse causality argument) but it is more sensible than the other two.

Response:

The 32 current member countries of the IEA are all in the OECD, so membership is an indirect measure of country development stage and income level, controlled for in our models by the GDP and GDP-squared terms. Originally the IEA was established to represent the interests of oil-importing countries (following the 1970s OPEC embargo) but has more recently been active in promoting pathways, policies, and clean energy innovation programmes in line with net-zero goals (including through its hosting of the CEM; see R2.2).

We think it is legitimate to test whether IEA membership interacts with policy density for two reasons. First, the IEA has an explicit policy evaluation function in energy efficiency and demand, in clean energy innovation, in renewable energy and low-carbon technologies (e.g., country focus chapters in the annual Energy Efficiency and Energy Technology Perspectives reports). Second, the IEA's work aims to diffuse best practice policies and policy insights among its member countries.

Our modelling tests whether this interaction between IEA membership and accumulating policy stocks affects emission intensity reduction tests.

As with the CEM, the reviewer is right that we cannot rule out reverse causality - we discuss causality and endogeneity in detail in R2.1 above.

Main changes in revised submission:

- inclusion of IEA interaction term model results in main text, with explanation of IEA role in policy diffusion

R2.5 I notice that the authors have used a dummy for EU membership – but it is unclear to me how the authors have dealt with EU policies on the policy density for EU member countries. Please clarify.

Response:

We only consider national policies in our study. EU policies are not directly included in our measure of policy density. However, EU legislation is captured indirectly through the domestic policies of EU member states when they transpose EU legislation into national law. This implies an interdependence between EU member states with regard to their climate policies (in addition to the many economic interdependencies), which we control for with our dummy for EU membership. More specifically, this dummy also includes EFTA countries, which have to transpose EU legislation, including climate policies, in order to be allowed to participate in the EU single market.

Main changes in revised submission:

- Clarification in Methods of how EU policies are coded in our main policy density variable.

R2.6 In the models displayed in Table 2, the authors never consider a full model (i.e. adding multiple variables into the model rather than just one variable at a time and then removing it again). This is especially odd as their individual models suggest that the additional variables

matter – hence this is suggesting that each individual model (and the coefficients of the individual variables) suffer(s) from omitted variable bias. I would recommend to show a full model (at least adding the dummy variables if adding all HHI indicators does not make sense).

Response:

In our original work we had run full models as suggested by the reviewer, but we did not include them in our original submission due to amplified collinearity problems with multiple independent variables in the same model. This overspecification led to: unstable coefficient estimates and difficulty in determining the individual effects of each variable; loss of efficiency, reducing the precision of the estimated coefficients, and increasing standard errors; and overfitting.

This applies to the full set of policy variables (density, portfolios specialise on instrument type, portfolios concentrated on sectoral coverage) as well to the dummy variable for long-term targets and governmental organisational (energy and/or climate ministries, membership of CEM, IEA, EU-EFTA).

We also tried full models with all the dummy variable interaction terms (but not the additional policy variables) together with our main policy density independent variable. As well as collinearity problems, from an applied econometrics perspective, including all dummies and interaction effects simultaneously in the same model specification makes the model practically uninterpretable as the dummies' effects are interrelated.

Following the reviewer's recommendation, we ran further tests to see how complete we could get without compromising model quality and interpretability. In Table ED17, we report full combined models with our main policy density variable, one of the four policy portfolio variables (instrument type and sectoral coverage), and five dummy variables - *but no interaction terms* - for long-term targets (relative and absolute), for energy and/or climate ministries, and for IEA and EU-EFTA membership.

All coefficient signs and magnitudes are in line with those of the parsimonious models reported in Table 2 of the main text. Across all models, the coefficient for total policy density remains strongly significant. The same holds for each of the policy portfolio variables that continue to perform well in their respective combined model. Significance of the energy and/or climate ministry dummy drops away, as does the EU-EFTA membership (which is highly correlated with IEA membership).

However, overall, these combined models give us further confidence in the results reported in main text.

Main changes in revised submission:

- new full combine model results reported in Table ED17
- new section of SI1 with discussion of full, combined model results, with summary in main text

Minor comments:

R2.7 In line 860 the authors state that the residuals are iid. Is this tested? Or assumed? Given the lack of testing for autocorrelation, cointegration and other common empirical issues, I would assume that this is their assumption (similar in S1 line 161).

Response:

Correct, this is assumed. For transparency, we removed this statement from the main manuscript and SI1 and we now simply refer to ε_{it} as the regression residuals. Overall, Root Mean Square Error (RMSE) estimates reported across model specifications in Table 1 (main manuscript) and Extended Data tables show minimal differences between predicted and observed values.

R2.8 In their cumulative policy density measure, do the authors also account for discontinued policies (e.g. the Australian carbon pricing scheme that only ran for a few years)?^[SEP]

Response:

The IEA Policies and Measures Database does not consistently report when policies end, so we were unable to capture “discontinued” policies in our policy density measure.

However, we believe this is not an issue for two reasons. First, the “discontinuing” or “dismantling” of policies is less frequent than one would expect. For example, Schaub et al. (2024) show that only a small fraction of climate policies terminate over the course of time. Second, the dismantling of climate policies correlates with low adoption of new climate policies. So even if we do not capture the dismantling of some policies during a period with a government with low climate ambition, the effect of dismantling is included in the slow or negligible increase in policy density during this period.

The reviewer is right in pointing to the Australian case, where a change in government in 2013 ended the carbon pricing scheme. This dismantling of policies also coincided with few newly adopted climate policies. The figure below for AUT shows this inverse correlation between policy adoption and dismantling in Australia (taken from Figure S14 in the Supplementary File of Schaub et al., 2022).

Graphs by 3-letter Country Code

In another example, the first Trump administration dismantled climate policies while also adopting comparatively few new climate policies. This was in contrast to the previous Obama administration and particularly the subsequent Biden administration.

In sum, our measure of policy density captures the comparatively rare occurrence of policy dismantling indirectly. Even if we could incorporate the dismantling of policies in our density measure, we would not expect our results to change significantly.

Main changes in revised submission:

- clarification in Methods of policy dismantling not being coded in our policy density variable + explanatory text on the inverse correlation between dismantling and new adoption

Schaub, S., Tosun, J., & Jordan, A. J. (2024). Climate action through policy expansion and/or dismantling: Country-comparative Insights: An introduction to the Special Issue. *Journal of Comparative Policy Analysis: Research and Practice*, 26(3-4), 215-232.

Schaub, S., Tosun, J., Jordan, A., & Enguer, J. (2022). Climate policy ambition: Exploring a policy density perspective. *Politics and Governance*, 10(3), 226-238.

R2.9 In Figure 1, I'm unclear what the ranges left and right of the schematic mean. I would suggest to clarify this.

Response:

The ranges in the schematic were supposed to indicate the variables we tested in our models; however we agree they were not clear so we have removed them. We explain the variables and their underlying measurements in detail in Methods and in SI.

Main changes in revised submission:

- revised Figure 1 in main text

R2.10 In Table 2, I would recommend adding a row at the top with the dependent variable. From what I gather from the note below the table it is always the same (CO2 intensity) but the headings on first reading suggest they are different. I would suggest to add a row clarifying the dependent variable.

Response:

Table revised as suggested.

R2.11 Unless I have missed it, I would recommend to add specific equations and further explanation (including maybe examples) for the calculation of the HHI in the S1. This would help comprehension of the HHI variables.

Response:

Full equations and explanations of the HHI (and alternative Shannon Index) diversity measures included in SI1 as suggested.

R2.12 The two paragraphs between line 60 and 62 seem disjointed. I am unclear whether the studies cited in line 60 are supporting the statements in line 62, especially as they are not cited in that paragraph anymore.

Response:

We have changed the text to clarify this. The ‘portfolio size matters’ was the simple summary of the previous paragraph including its citations. The segue into the next paragraph shifts focus from numbers of policies to portfolio design characteristics.

R2.13 Line 73 suggests the authors have built the underlying policy database while in the methods it seems that the authors have used the existing Policies and Measures database from the IEA. Perhaps it would be good to clarify in the main text that additional analysis went into the database.

Response:

We have clarified in the main text on how we derived the policy data. We used the IEA Policies and Measures database as our data source. We then assigned the identified policies to sectors and instrument types based on our own classification. And then we created our policy density measures. This involved transforming data at the policy level into aggregated data at the country year level.

R2.14 In line 856 the subscript i should likely not be capitalised.

Response:

We couldn't find this in our submission! We have checked all formulas and equations and think they are all correct, but we will double check with the editorial team should the manuscript be accepted for publication.

Reviewer #3 (Remarks to the Author):

R3.1 The authors investigate the effectiveness of policies in reducing CO2 emissions. Based on a panel dataset of 43 countries 2000-2019, the authors test whether the number of policies, specialization on instrument types, sectoral coverage, emission reduction targets, and government institutions affect CO2 emission intensities. In my opinion, this is a promising manuscript, to which I have the following comments to give:

[Authors' note: we have moved R3.4 up to here, as we respond to R3.2 + R3.4 together]

R3.2 As the title and the beginning of the discussion rightly suggest, the ultimate goal of climate policy is to reduce the absolute level of emissions, not intensities. Accordingly, it would be critical for the authors to look at the level of emissions as outcome variable as well.

R3.4 Also a possible interaction effect of policies and absolute targets on absolute emissions would be interesting to see. Any relatively time-invariant country characteristics (size, population, institutions, ...), that affect the average level of emissions and GDP, are absorbed by the unit fixed effects and are thus not a valid reason to only investigate intensities.

Response:

We agree that ultimately climate policy has to reduce absolute levels of emissions. We use emission intensity as our dependent variable following the precedent in similar attribution studies (e.g., Eskander and Fankhauser 2020) because it accounts for changes in emissions driven by a country's overall economic performance and size. (GDP and CO2 are correlated at 92% in our sample). Using emission intensity rather than absolute emissions therefore helps reduce confounding factors when interpreting the effect of climate policy.

In response to the reviewer's comment, we ran additional models reported in full in Table ED15 and summarised in SI1 for which we use the natural logarithm of absolute CO2 emissions as the dependent variable instead of emissions intensity. Overall, results remain consistent.

We start by comparing our baseline model (column 1, Table ED15) with a new model specification with \ln CO2 as the dependent variable (column 2, Table ED15). The independent and control variables remain the same across both models. Policy density remains negative and statistically significant with a smaller (in absolute terms) coefficient compared to the baseline model. This is reasonable from an econometric perspective, as this model is mis-specified due to the absence of controls for economic activity, the primary driver of emissions. To address this misspecification, we include the natural logarithm of GDP as an additional control variable (column 3, Table ED15). Again, results remain consistent, with the coefficient for total policy density slightly larger in magnitude than in the baseline model.

For the interaction effect between policy density and absolute emission-reduction targets on absolute levels of CO2 emissions, we add this in (column 4, Table ED15). The coefficient for the interaction effect remains strongly significant, similar to in our baseline model with emission intensity as the dependent variable (main text, and Table ED4).

As additional sensitivity tests, we replace the natural logarithm of GDP with the natural logarithm of population as alternative control variables in our models (columns 5 & 6, Table ED15). Again, results remain remarkably consistent. The coefficients for policy density and the

interaction effect with long-term targets are slightly larger in magnitude than the models with GDP as a control variable.

Overall, we find that policy density is associated with more rapid emission reductions regardless of whether emission intensity or absolute emissions are specified as the dependent variable in our models. To avoid misspecification issues, we focus on the emission intensity models in main text, but include the full set of absolute emission models in a new section of SI1 and Table ED15.

Main changes in revised submission:

- new absolute CO2 emission model results reported in Table ED15
- new section of SI with discussion of absolute CO2 emission model results, with summary in main text

Eskander, S. M. S. U., & Fankhauser, S. (2020). Reduction in greenhouse gas emissions from national climate legislation. *Nature Climate Change*, 10(8), 750-756. <https://doi.org/10.1038/s41558-020-0831-z>

R3.3. This could contribute valuable insights into possible rebound effects of climate policies. If policies incentivize efficiency improvements, possibly in connection with an economic stimulus, policies could increase emissions despite relative decoupling of GDP from CO2 emissions.

Response:

We absolutely agree that the macroeconomic effects of climate policies on growth are critically important to understand, and central to discussions on decoupling, green growth, and degrowth. However, our econometric models are not able to isolate these recursive relationships in the CO2/GDP and climate policy time trends.

As noted in the previous response, our additional models with CO2 as the dependent variable are consistent in showing the effectiveness of climate policies - the main aim of our study. So to the extent the reviewer's observation of the potential rebound effects from efficiency-improving climate policy holds, these effects are captured, but not isolatable, in the models. (We address endogeneity concerns that emission reductions may lead to climate policies in detail in R2.1).

To explore this further within our data, we reanalysed our policy dataset and found 13-15% of all policies were explicitly linked with energy efficiency or energy savings. We reasoned that these would be the main types of policy associated with possible rebound, but are a minority in our policy stock variable.

We also note literature offering a more circumspect view of the magnitude of possible rebound from such efficiency policies, particularly Gillingham et al. (2013, 2016).

So although we cannot address the reviewer's concern directly, overall we think our main insights on climate policy effectiveness are robust.

Using CO2 emissions as the dependent variable emphasises absolute emissions trends that are influenced by not only country size and structural changes but also other time-varying factors such as energy demand and economic growth trajectories. While fixed effects mitigate some

confounding influences, they cannot fully capture the dynamic interplay between absolute emissions, policy-induced efficiency gains, and economic output.

Gillingham, K., Kotchen, M. J., Rapson, D. S., & Wagner, G. (2013). Energy policy: The rebound effect is overplayed [10.1038/493475a]. *Nature*, 493(7433), 475-476. <http://dx.doi.org/10.1038/493475a>

Gillingham, K., Rapson, D., & Wagner, G. (2016). The Rebound Effect and Energy Efficiency Policy. *Review of Environmental Economics and Policy*, 10(1), 68-88. <https://doi.org/10.1093/reep/rev017>

R3.5. While I think the fixed effects model specification generally is a good choice, the control variables could be problematic in the sense of "bad control". For instance, one way to achieve a lower GDP CO2 intensity is to shift economic activity to the service sector while offshoring polluting manufacturing. This means climate policies could affect both the control (service value added, imports, ...) and outcome variable (emissions). I would like to see a parsimonious specification (with only the fixed effects and policy measures) as a baseline and a more detailed discussion of possible mechanisms.

Response:

To address the reviewers' concern, we conducted additional sensitivity tests by removing the control variable for the share of the services sector within a country. We then compared the modified model with our baseline model. If the effect of shifting economic activity to the services sector while offshoring polluting manufacturing was associated with climate policy density, removing the control should change results from the baseline model. However the coefficient for policy density, as well as coefficients for the other controls, all remain virtually unchanged.

As an additional sensitivity test, we run a modified model that removes the control variable for the share of imports in the country's balance of payments. This should allow us to indirectly account for carbon leakage that - as the reviewer notes - may occur when a higher share of imports with a larger carbon footprint replaces domestically produced goods. Again, the modified and baseline models are remarkably consistent, although in this case, the coefficient for total policy density in the modified model reduces slightly in magnitude.

We show full model results in Table ED16. Overall, we do not find evidence contrary to our initial findings. We reason that if significant carbon leakage effects were present, policy accumulation would lose relevance as an explanatory variable on emission intensity reductions which is not the case.

Main changes in revised submission:

- new parsimonious model sensitivity tests reported in Table ED16
- new section of SI with discussion of model results

R3.6. The authors state that there are no significant differences between instrument types (regulatory, economic, voluntary) but I was not able to find the corresponding results. It would be surprising if voluntary instruments were as effective as mandatory types.

Response:

In our original submission, we focused on the characteristics of national policy portfolios rather than the relative contributions of specific instrument types. We did include results on portfolio designs that were specialised or concentrated on instrument types compared to those with a more diverse mix of instrument types, finding the former to be more effective. However, as the reviewer notes, we did not report which instrument type was the most effective as our models were not designed to evaluate this. (See R1.3 for a full discussion of this limitation of our modelling approach).

In response to the reviewer's concern, we have now run additional models for policy density by instrument type. Due to collinearity, we ran a separate model including controls for each of the 8 instrument types:

- *regulatory*: legal interventions, monitoring, and regulations
- *economic*: market-based and direct provision
- *voluntary*: planning, informational, and other voluntary

Full results are shown in Table ED13. Given that our dependent variable is emissions intensity (CO₂/GDP), we did not expect to find a clear effect given we were including as our independent variable only a subset of all policies (coded as being of a particular instrument type). However, five of the eight models show negative and significant coefficients for the policy density by instrument type variable. The largest coefficient in absolute magnitude is for legal interventions and monitoring, both of which are categorised as 'regulatory'. Three of eight models show negative but non-significant coefficients. Two are for informational and other voluntary instruments, both categorised as 'voluntary'. The third is for the regulations (one of the instrument types in the 'regulatory' category).

The null results for the voluntary instrument types is not surprising given the weaker incentives these entail. This is also in line with the reviewer's expectation in the comment.

The null result for regulations is surprising. We think this result has to do with increased heteroskedasticity (noise) within our coding of regulations as an instrument type which spans a wide range from binding performance and emission standards to weaker reporting, safety, or compliance type regulations. For instance, such regulations include safety and performance requirements for heaters, air conditioners or refrigerators in buildings (such as regulation "GB 4706.13-2014 on the safety on household and similar electrical appliances" adopted in China in 2016 or regulation "SI 994-1 on safety and performance requirements of air conditioners" adopted in Israel in 2009), mandatory reporting requirements for energy efficiency actions (such as a respective regulation adopted in 2019 in the Netherlands) or guidelines for the estimation of greenhouse gas emissions (such as a regulation adopted in 2017 in Australia which gives technical guidelines for GHG emissions estimation in energy facilities). Our country vignettes in SI2 further show that the largest shares of regulations relate to the buildings and transport sector where emissions reductions have been relatively low, especially when comparing to emissions reductions in energy supply.

As also discussed in R1.4, we are also unable to control for the differing stringency of regulations within our policy sample.

However, given the general limitation of these models trying to explain economy-wide emission intensity but with only a subset of all policies, we are reluctant to over-interpret this result.

Main changes in revised submission:

- new model results in Table ED13
- explanatory text in Methods and SI1 (linked from main text)

R3.7. How common is the exit of policies in the data? When governments change so often does their ideological stance. For instance, Trump tried to undo environmental policies of the Obama administration. Is this relevant for the results?

Response:

Reviewer 2 had very similar questions, so we have combined our response in full in R2.8 (see above). In general our measure of policy density captures the comparatively rare occurrences of policy dismantling indirectly because policy dismantling inversely correlates with new climate policy adoption.

Climate policy portfolios that accelerate emission reductions

2nd revised submission: Sep 2025

(1st revised submission: Jan 2025, original submission Aug 2024)

Response to Reviewers.

We respond in detail to all the reviewers' comments below with a summary of the main changes in our second revised submission. Prompted by the reviewers' recommendations and concerns, we have updated the manuscript, method and SI sections with new analyses, results and argumentation particularly on (1) the relationship between policy density and stringency, (2) our use of control variables, (3) our use of attribution terminology, and (4) how our results compare with other recent publications on the effectiveness of climate policy. This revised submission is further strengthened as a result, and we appreciate the reviewers for encouraging this.

green text = editors' and reviewers' comments

black text = our response

blue text (indented) = examples of new or revised text from manuscript

REVIEWER COMMENTS

Reviewer #1 (Remarks to the Author):

R1.1 The paper has improved a lot, and I think the paper presents interesting insights in the impact of policies. I think the authors have addressed the many comments in a good way with a very detailed response. Although for some comments they were not able to address those, but this was due to good reasons as they indicated. I think one of my main omission of the analysis was policy stringency is missing, because the number of policies is not a good measure of the quality and GHG impact of climate policies. The authors have included literature that showed that policy stringency and policy density are also strongly correlated, so I think well addressed.

R1.2 The work is original, but soon after their submission Stechemesser et al. published a similar analysis in Science. I asked for a comparison with Stechemesser et al., but I think this is still not done yet. In their response to my reviewer's comments they mentioned that "Stechemesser et al identify total emission reductions in the range 0.6-1.8 GtCO₂ from their set of 69 policy-attributed structural breaks in emission trends. This is an order of magnitude smaller than the estimate of 27.5 GtCO₂ reductions in this study across all policy portfolios including continuously incremental reductions as well as major discontinuities." I was surprised to see this difference. However, the authors did not discuss the much lower reduction found by Stechemesser et al in their paper.

R1.3 Action required: It would be good if the authors would discuss this major difference, and the possible reasons, as the reader may expect a comparison with Stechemesser et al., as this study has a similar research objective. The comparison could also move to the SI, and only highlight the main conclusions in the main text.

Response:

We have included a new SI1 section with an extended literature review of attribution studies estimating emission reductions from climate policies, including the Stechemesser et al. study. This allows us to situate our attribution estimates in the literature, and discuss differences as a function of aim, methodology, and country and time samples. We also

summarise this in main text as the reviewer suggested. The main text summary is reproduced here:

By comparing observed emission trends with a counterfactual no-policy scenario, we attribute 27.5 GtCO₂ avoided emissions to the cumulative effect of all climate policy portfolios over the period 2000-2022, of which 14.6 GtCO₂ are in the BRIICS countries (see Figure Methods 1). In 2022, this is equivalent to 3.5 GtCO₂/yr which is in line with other attribution estimates that use a variety of methods, country and time samples (see SI1 for full discussion). Emission reductions attributed to climate policies in the literature include: 2-7 GtCO₂/yr (global, to 2020) (Hoppe, Hinder et al. 2023), 1.3-5.9 GtCO₂/yr (133 countries, 1999-2016) (Eskander and Fankhauser 2020), 1.3 GtCO₂/yr (39 countries, 2005-2012) (Maamoun 2019), 1.3-2.5 GtCO₂/yr (renewable energy policies only) (Hoppe, Hinder et al. 2023). Other studies attribute emission reductions of 28% to climate policies (48 countries, 2000-2021) (Nachtigall, Lutz et al. 2024), of 0.8%-2.8% to each new climate law introduced (Eskander and Fankhauser 2023), or of 4-15% to carbon pricing policies only (21 countries) (Döbbeling-Hildebrandt, Miersch et al. 2024).

In their recent study using difference-in-differences methods to make more robust causal inferences than our fixed-effects model allows, Stechemesser, Koch et al. (2024) attributed total emission reductions of 0.6-1.8 GtCO₂ to a set of 69 structural breaks in sectoral emission trends (41 countries, 2000-2020). Their country sample and study period are similar to ours, but their policy-attributed emission reductions are an order of magnitude lower. We interpret this as the result of their tighter focus on statistically identifiable discontinuities in sectoral emissions caused by specific combinations of sectoral policy instruments introduced or tightened in the preceding 2 years. In contrast, our models assess policy portfolios' cumulatively incremental impact over the long-term; discontinuities are subsumed within these aggregate economy-wide trends.

The new SI1 section includes further details on attribution methods and studies. For example:

In Chapter 14 of the IPCC's WG3 Sixth Assessment Report, Babiker, Bertoldi et al. (2022) define policy attribution as: "*the extent to which emission-relevant outcomes (including the emission intensity of GDP) charted for countries as well as sectors and technologies may be reasonably attributed to policies implemented prior to the observed changes.*"

Most ex-post policy impact assessments are for specific policy instruments in particular contexts (Hoppe, Hinder et al. 2023). Very few studies assess collective or 'global' impacts of policy stocks comprising multiple instrument types. Attribution methodologies used to identify the effect of mitigation policies controlling for confounding factors like socioeconomic conditions, fossil fuel prices, and trade-related policies, include "*statistical attribution methodologies, including experimental and quasi-experimental design, instrumental variable approaches, and simple correlational methods*" (Babiker, Bertoldi et al. 2022).

We align with the statistical attribution framework for 'global' policy impacts established by Eskander and Fankhauser (2020) in the climate policy attribution literature, extending their testing of policy density (cumulative policy stocks) to additionally assess the effect of policy stringency (see also Nachtigall, Lutz et al.

(2024)), policy instrument types, policy sectoral coverage, targets and policy-relevant organisations.

In their review of both 'global' and instrument-specific attribution literature, Hoppe, Hinder et al. (2023) estimate a plausible range of emission reduction of 2-7 GtCO₂e/yr compared to a no-policy counterfactual, equivalent to 4-5% of total GHG emissions in 2020. Evidence from multiple studies span an attribution range of 1.3-5.9 GtCO₂/yr for different country samples and time periods:

- Eskander and Fankhauser (2020) attribute reductions of 5.9 GtCO₂/yr (in 2016), or 38 GtCO₂-eq. cumulatively, to climate laws (1999-2016, 133 countries);
- Maamoun (2019) attribute reductions of 1.3 GtCO₂/yr (in 2012), equivalent to -7%/yr, to climate policies pursuant to Kyoto Protocol emission-reduction targets (2005-2012, 39 countries with in UNFCCC Annex B);
- Babiker, Bertoldi et al. (2022) attribute reductions of 1.8-3 GtCO₂/yr to multiple policy instruments including energy-efficiency programmes and renewables diffusion (unspecified time horizon & country sample);
- Grubb, Okereke et al. (2022) attribute reductions of 4-5 GtCO₂/yr (in 2019) to the cumulative impact of policies (2010-2019, global);
- UNFCCC (2020) attribute projected reductions of ~3.8 GtCO₂/yr to the 38% of >2500 implemented policies and measures for which estimates are available (36 countries in UNFCCC Annex 1);
- Hoppe, Hinder et al. (2023) draw on IRENA data to attribute emission reductions of 1.3-2.5 GtCO₂/yr to renewable energy policy.

Major attribution studies since the IPCC assessment, including those for specific policy instruments (carbon pricing) or targeted sectors (renewable energy), are:

- Stechemesser, Koch et al. (2024) attribute reductions of 0.6-1.8 GtCO₂ from discontinuous breaks in emission trends to 69 sets of policies in combination (2000-2020, 41 countries);
- Nachtigall, Lutz et al. (2024) attribute emission reductions of 28% to climate policies compared to a no-policy counterfactual, emphasising the increasing stringency of policy portfolios over time (2000-2021, 48 countries);
- Eskander and Fankhauser (2023) attribute emission intensity reductions over the long-term (>3 years) of 0.84–2.78% (territorial or production-based) or 0.91–2.61% (consumption-based) to each new climate law introduced (1996–2018, 111 countries);
- Döbbeling-Hildebrandt, Miersch et al. (2024) meta-analyse 80 *ex post* evaluation studies of 21 carbon pricing schemes worldwide to generalise a de-biased effect of this specific policy instrument to the order of -4% to -15% emission reductions.

Reviewer #2 (Remarks to the Author):

R2.1 I would like to thank the authors for their efforts in revising and improving this study. I feel that the study has substantially improved. I would like to highlight that the authors have considerably improved their discussions of endogeneity and attribution in the supplementary part of their paper. I think this discussion was necessary and has benefitted the overall paper. I appreciate the efforts by the authors to defend their attribution claims in the study and indeed I believe there is a good discussion (mainly the Granger causality testing) surrounding the temporal setting that policies are antecedent to emission reductions.

R2.2 However, I remain critical of the causal validity of the counterfactual construction and the headline number of cumulative reductions of 27 GtCO₂. I do not believe that simple precedent from another study is enough to justify the term “attribution”. I remain partly convinced that the policy density measure is more a representation of how liberal a society is and how important climate change is for society – but I’d be very careful to translate their results into policy advice. Maximising policy density is clearly not what we should be doing – we should carefully design a small number of (mixed) effective policies with clear aims and effect pathways rather than aim to simply increase policy density.

Response:

We appreciate the reviewer’s concern and we certainly want to avoid the main policy insight from our analysis to be about simple increasing policy stocks to maximise numbers density. We have revised the manuscript and SI in several way to address this concern.

First, we have made more prominent our analysis of policy stringency as a correlate of policy density to emphasise that increasing stringency explains emission reductions rather than just policy stocks *per se*. (This includes a new section in Methods on our policy stringency model). We also draw on the policy sequencing literature to explain why stringency and density tend to correlate as countries build on ‘easy wins’ to strengthen their climate action. New or substantially revised text in our manuscript include:

[in Results]:

Building on prior attribution studies (see SI1), we find that both policy density (Eskander and Fankhauser 2020) and policy stringency (Nachtigall, Lutz et al. 2024) are associated with faster reductions in fossil CO₂ emission intensity. We interpret this result through the lens of policy sequencing (Linsenmeier, Mohommad et al. 2022) which contends that policy portfolios become increasingly stringent as they develop (Pahle, Burtraw et al. 2018, Linsenmeier, Mohommad et al. 2022). (Stringency is the calibration of policy instruments, e.g., the specific level of a carbon tax, and is broadly equivalent to strictness (Schaffrin, Sewerin et al. 2015) or the extent to which climate policies incentivise or enable emission reductions (Creutzig, Becker et al. 2024)).

...

[in Discussion]:

Our results confirm the basic insight from attribution studies that larger, more developed (Eskander and Fankhauser 2020) or more stringent (Nachtigall, Lutz et al. 2024) policy portfolios are associated with faster emission reductions. Policy stocks grow due to complex policy design processes, interactions between policy goals and sectors, and competing political economic interests that all result in multiple policies or policy packages being introduced to tackle specific problems (Adam, Hurka et al. 2019). Climate policy accumulation further results from the incremental sequencing

of policies over time to remove economic, societal, and political barriers to climate action (Pahle, Burtraw et al. 2018, Nascimento, den Elzen et al. 2023). Sequencing means policy density and policy stringency are strongly correlated as policies become more stringent as portfolios mature (Linsenmeier, Mohommad et al. 2022, Nachtigall, Lutz et al. 2024). Stronger public demand for emission reductions further contributes to policy portfolios developing towards greater stringency (Schaffer, Oehl et al. 2021). This is why we interpret our main model result on policy density through the lens of stringency, which also underpins our attribution claim (see S11 for full discussion).

Second, we are now also clearer in our results and discussion that it is the design and composition of climate policy portfolios that explain emission reduction outcomes rather than simply policy density. Indeed, we see this as the principal contribution of our study. This includes more discussion on the importance of policy combinations or mixes as the reviewer notes. New or substantially revised text in our Discussion section on policy mixes includes:

Our results show that specialising on instrument types in policy portfolios is associated with faster emission reductions. Different political traditions and contexts favour different policy instruments (Meckling and Nahm 2018). In a comparative analysis of renewable energy policy portfolios, Schmidt and Sewerin (2019) similarly find that dominant instrument types vary by country, and that instrument specialisation rather than diversity is associated with higher renewable energy deployment.

Policy evaluation studies provide complementary insights on the effectiveness of specific instruments (Peñasco, Anadón et al. 2021, Döbbeling-Hildebrandt, Miersch et al. 2024). Over time, and with the increasing need for more stringent climate action, instrument choices have shifted from voluntary to regulatory and economic (Schmidt and Fleig 2018). In their attribution models, Nachtigall, Lutz et al. (2024) find that in general market-based instruments and targets are associated with faster emission reductions than non-market (regulatory) instruments. Additional robustness testing of our policy portfolio models using only subsets of policy instruments by type similarly show that portfolios weighted towards economic instrument types outperform others (see Methods and S11).

From a normative perspective, instruments like carbon pricing offer the most economically efficient way of reducing emissions both within and across sectors (Sterner and Coria 2013, World Bank 2023). However, carbon prices have been relatively low in many countries and carbon pricing has tended to be a more recent instrument choice (Linsenmeier, Mohommad et al. 2022). The importance of regulatory and technology policies for emission reductions has been clearly evidenced in empirical assessments of low-carbon innovation (Grubb, Drummond et al. 2021) and in modelling analyses (Bertram, Luderer et al. 2015).

Consequently, rather than single instrument dominance, policy mixes - including those comprising different instrument types – tend to be more effective for multiple reasons: mutual reinforcement and positive spillovers; complementary time-varying effects; capacity for policymakers to address multiple problems simultaneously (Hoppe, Hinder et al. 2023). Stechemesser, Koch et al. (2024) demonstrate empirically that policy combinations are more effective for reducing emissions than standalone policies, with carbon taxes an exception.

Third, we position our attribution approach more clearly within the literature through a new section in S11 (reproduced in our response to R1.3) which also includes the difference-in-

differences (DiD) study by Stechemesser, Koch et al. (2024) whose design allows robust causal inference of climate policy impacts on emissions. As the reviewer notes, our attribution methodology using two-way fixed-effect models does not directly establish causality so we provide multiple other lines of evidence to back up our attribution claim. We also further strengthen our model specification in an extended new SI1 section on controls and secondary confounding variables (reproduced in our responses to R3, particularly R3.3-R3.6). This includes further model testing designed to clean up the attribution of emission intensity reductions to climate policy portfolio elements.

R2.3 At the very least, I would recommend to the authors to further review their language in lines 88, 100 and 295 and to directly compare their headline statements with Stechemesser et al. in the main text, as they do in at the bottom of page 4 of the rebuttal to reviewer 1.

Response:

We've implemented all the reviewer's suggestions on language and have also included the comparison of our results with Stechemesser et al. in the main text. In all the instances of implied causality the reviewer notes, we've either removed the claim completely, or changed the language to emphasise association not causation. We then limit our attribution claim to the conclusions, supported by literature (see R2.2). This text is as follows:

[in Introduction]:

Attribution studies use econometric methods to identify the general effect of climate policies on emissions, controlling for other determinants of emission trends. These studies have demonstrated the effect of larger policy stocks (Eskander, Fankhauser et al. 2021, Eskander and Fankhauser 2023), of more stringent policies (Nachtigall, Lutz et al. 2024), and of specific policy combinations (Stechemesser, Koch et al. 2024). The difference-in-differences modelling used by (Stechemesser, Koch et al. 2024) allowed stronger causal attribution but was focused on a set of 69 discontinuities in emission trends from recent policy introductions rather than the cumulative effect of policy portfolios over the long-term.

[in Discussion]:

By comparing observed emission trends with a counterfactual no-policy scenario, we attribute 27.5 GtCO₂ avoided emissions to the cumulative effect of all climate policy portfolios over the period 2000-2022, of which 14.6 GtCO₂ are in the BRIICS countries (see Figure Methods 1). In 2022, this is equivalent to 3.5 GtCO₂/yr which is in line with other attribution estimates that use a variety of methods, country and time samples (see SI1 for full discussion). Emission reductions attributed to climate policies in the literature include: 2-7 GtCO₂/yr (global, to 2020) (Hoppe, Hinder et al. 2023), 1.3-5.9 GtCO₂/yr (133 countries, 1999-2016) (Eskander and Fankhauser 2020), 1.3 GtCO₂/yr (39 countries, 2005-2012) (Maamoun 2019), 1.3-2.5 GtCO₂/yr (renewable energy policies only) (Hoppe, Hinder et al. 2023). Other studies attribute emission reductions of 28% to climate policies (48 countries, 2000-2021) (Nachtigall, Lutz et al. 2024), of 0.8%-2.8% to each new climate law introduced (Eskander and Fankhauser 2023), or of 4-15% to carbon pricing policies only (21 countries) (Döbbling-Hildebrandt, Miersch et al. 2024).

In their recent study using difference-in-differences methods to make more robust causal inferences than our fixed-effects model allows, Stechemesser, Koch et al. (2024) attributed total emission reductions of 0.6-1.8 GtCO₂ to a set of 69 structural

breaks in sectoral emission trends (41 countries, 2000-2020). Their country sample and study period are similar to ours, but their policy-attributed emission reductions are an order of magnitude lower. We interpret this as the result of their tighter focus on statistically identifiable discontinuities in sectoral emissions caused by specific combinations of sectoral policy instruments introduced or tightened in the preceding 2 years. In contrast, our models assess policy portfolios' cumulatively incremental impact over the long-term; discontinuities are subsumed within these aggregate economy-wide trends.

R2.4 I would also suggest to clarify that their results do not challenge the validity of the Tinbergen rule – but rather that their results indicate that policy-makers are not actually adhering to it. As currently framed, they seem to suggest that their results show that the Tinbergen rule is not valid.

Response:

We've dropped all reference to the Tinbergen rule as our modelling is not designed to test it. We've refocused the discussion of our policy density finding on its correlation with policy stringency as a better explanation of the emissions impact. We also discuss increasing policy density in light of evidence on the effectiveness of policy combinations and policy mixes.

R2.5 I would further remove the implicit reference to the EKC in line 144 – there remains substantial academic debate around the presence of an EKC in carbon emissions, so I would rephrase that element.

Response:

We've dropped all reference to EKC's, recognising the ongoing debates around their applicability to GHG emissions.

R2.6 I recommend to avoid using the abbreviation "IV" in Table 1 for independent variables – this in econometrics generally refers to an instrumental variable and could be confusing.

Response:

We've dropped shorthand notation to IVs (as well as DVs).

R2.7 I would also ask the authors to show the full numbers in the two rightmost columns in Table Methods 2.

Response:

We've included the full model results from Eskander & Fankhauser in Table Methods 2.

Reviewer #3 (Remarks to the Author):

R3.1. The authors have responded to my comments and mostly implemented appropriate changes in the manuscript. I still have some major comments about the identification strategy and interpretation of results.

R3.2. The selection of control variables is not sufficiently justified and could substantially bias the coefficient of interest, yielding misleading results, even if well intentioned. In

response to my previous concern about bad control, the authors provided to additional models ED16 in which they separately exclude the service and import shares. In SI1 Section 2.9, the authors state that the estimate of beta1 "remains virtually unchanged" when omitting these variables. This is true for the service share but without the import share the point estimate of beta1 decreases by 35% and notably increases in variance. In my opinion, this is a relevant change that suggests a mediating role of imports.

R3.3. A rigorous engagement with the functional relationships of the included variables is still missing, possibly in the form of a directed acyclic graph (Cinelli et al 2024). As parsimonious specification in a two-way fixed effects setting, I would expect models with only the explanatory variable of interest, especially since the FE already absorb a very large share of the total variance in the outcome (~70% in this case).

Response:

We have strengthened our identification strategy including, as the reviewer suggests, through the use of directed acyclic graphs (DAG) to clarify the functional relationships among variables. We went back through relevant literature including Pearl (2009) and subsequent work by Cinelli and others (Cunningham 2021, Celli 2022, Hünernmund and Bareinboim 2023, Cinelli, Forney et al. 2024) to inform our DAG that illustrates our research design and explicitly maps the assumed relationships among all variables included in our model specification. We first summarise the additional analysis we conducted on controls and model specification before explaining our DAG procedure in more detail.

Summary of additional analysis on controls and model specification

From an econometric perspective, the inclusion of GDP per capita, services share, and import share aligns with standard practices in applied panel data analysis to block confounding pathways. In a two-way fixed-effects framework, controlling for observed time-varying confounders is necessary when the treatment is also time-varying and potentially correlated with omitted variables that affect the outcome (Angrist and Pischke 2010, Wooldridge 2021). As emphasised by Cinelli and Hazlett (2019), even in high-dimensional settings, control strategies grounded in causal assumptions help prevent omitted variable bias and enhance identification of causal effects. We use directed acyclic graphs (DAGs) to explain and justify the role of our control variables (Cunningham 2021, Celli 2022, Cinelli, Forney et al. 2024).

Our sequential inclusion of controls mirrors the logic of a Frisch-Waugh-Lovell (FWL) decomposition, whereby we partial out the effect of other covariates to isolate the variation in policy density that is orthogonal to plausible confounders. The modest changes in the coefficient magnitude and standard errors in later model stages (columns 6–9 of Table R1 below) suggest that the control set reduces bias without introducing excessive multicollinearity or instability. This pattern is consistent with an “unbiased conditional” specification.

Directed Acyclic Graphs (DAGs): Confounders & backdoor paths

We present the basic DAG in Figure R1a and then discuss the causal assumptions and roles of each variable.

Figure R1. Directed acyclic graph (DAGs) of our research design: left panel [a] shows confounders; right panel [b] shows secondary controls.

[a]

[b]

In our model, Y denotes the outcome (dependent) variable i.e., emission intensity; D is the treatment (independent) variable capturing policy density; X represents GDP per capita; SER indicates the share of the services sector in the economy; and IMP denotes the share of imports; HF represents the Hodrick-Prescott filter; RoL denotes the Rule of Law index; and TEMP denotes the temperature variation from the long-term mean.

The direct path from D to Y captures the causal effect of interest: the impact of policy density (D) on emission intensity (Y):

(D) Policy Density \rightarrow (Y) Emission Intensity

To obtain an unbiased estimate of the causal effect, it is necessary to block the following backdoor paths using appropriate controls (pp79-85 in Pearl 2009, pp99-102 in Cunningham 2021). These paths (Figure R1a) represent non-causal associations that could bias the estimation of the treatment effect:

1. $D \leftarrow X \rightarrow Y$
2. $D \leftarrow SER \rightarrow X \rightarrow Y$
3. $D \leftarrow SER \rightarrow IMP \rightarrow X \rightarrow Y$
4. $D \leftarrow SER \rightarrow IMP \rightarrow Y$

Backdoor path 1:

1. $D \leftarrow X \rightarrow Y$, in which X is GDP per capita (including its square term).

This backdoor path introduces spurious correlations between D and Y due to variation in the confounder X that affects both D and Y. Here the confounder X is GDP per capita and its squared term, a proxy for a country's development stage. Including the GDP squared term captures the non-linear (U-shaped) effect of development stage on emission intensity. In our study, climate policy density serves as a proxy for this transition so we expect a similar nonlinear relationship as for emission intensity. GDP per capita is a confounder as it correlates with the both the treatment and the outcome although in a non-causal way (see also Table R2).

Backdoor path 2:

2. $D \leftarrow SER \rightarrow X \rightarrow Y$, in which SER is the share of services in economic activity.

Controlling for GDP per capita closes one backdoor path, but introduces a second backdoor path from D to Y that arises due to variation in the share of economic activity in the services sector (SER) which is therefore a potential confounder. A higher services share is closely associated with a country's level of development. More developed economies tend to have larger services sectors that yield greater value-added activity than manufacturing or agriculture (Herrendorf, Rogerson et al. 2014). In addition, specialisation in services such as real estate, finance, and insurance prompts regulatory oversight that contributes to higher

policy density. The delivery of services typically involves complex transactions and institutional arrangements, requiring more sophisticated regulatory environments to ensure efficiency, stability, and consumer protection (Rodrik 2000). This is also shown in Table R2.

To close this second backdoor path and avoid omitted variable bias, we control for the share of the services sector in the economy. Structural change within the services sector could also directly reduce emission intensity, although to a lesser extent than the backdoor path shown in the DAG that we account for in our approach.

Backdoor paths 3 and 4:

3. $D \leftarrow SER \rightarrow IMP \rightarrow X \rightarrow Y$, in which IMP is the share of imports in a country's economy.

4. $D \leftarrow SER \rightarrow IMP \rightarrow Y$

Controlling for services introduces two additional backdoor paths that must be addressed to avoid bias. First, specialisation in services can generate comparative trade advantages for these types of exports, by extension increasing the share of imports of goods, specifically of manufactured products. Policy density may then be jointly influenced by rising services-sector specialisation and increased goods imports as governments respond to trade practices (e.g. dumping) by introducing protective regulations to safeguard domestic industries. As Rodrik (2018) argues, trade agreements become more about domestic rules and regulations than tariffs and non-tariff barriers.

Second, increased reliance on services-based economic activity and corresponding imports of manufactured goods is associated with reduced domestic emission intensity. This reflects a form of carbon leakage through which more emission-intensive production is offshored to developing countries (Eskander and Fankhauser 2023).

Controlling for imports closes two backdoor paths: between services and policy density, and between services and emission intensity.

Bad Controls & Colliders

According to Cunningham (2021), it is common for a collider to entirely flip the sign of the coefficient of interest when introduced in the model: *“Angrist and Pischke (2009) talk about this problem in a different way using language called “bad controls”. Bad controls are not merely conditioning on outcomes. Rather, they are any situation in which the outcome had been a collider linking the treatment to the outcome of interest.”*

Starting with a model specification without fixed effects and including only our treatment variable, we can see in column 1-Table R1 that the coefficient for policy density is negative and significant as we expect. Controlling for country effects in column 2-Table R1, the coefficient remains negative and significant. These first two models are called ‘biased unconditional’ in a relevant example used in Cunningham (2021, p110).

Once we introduce year fixed effects in column 3-Table R1 the coefficient flips its sign to become positive. This is not reasonable and the equivalent model in Cunningham (2021, pp106-110) is called ‘biased biased’. Year dummies controlling for time fixed effects effectively function as a collider in our case. This is because when we control for time, we introduce a backdoor path for policy density and emission intensity that is spurious and distorts the nature of the causal relationship. Policy density tends to increase over similar time periods for group of countries that are part of political unions and common economic markets. In the EU, for example: “More than half of the national measures are EU regulations. In some cases, the ratio is such that EU regulations account for 80 percent of national climate change policies (Netherlands and Belgium)” (Jahn 2024). If policy density

increases at similar rates in specific years for groups of countries within our sample, then simple models of emission intensity reduction while controlling for yearly effects will understate the effect of policy density (e.g., the baseline parsimonious TWFE model in column 3-Table R1).

To counteract this, we first need to introduce the confounder discussed above that accounts for countries' development stage (column 4-Table R1). This confounder captures better underlying variation among countries within the same group and accounts for additional nuances in the effect over policy density conditional on countries' economic status. We can observe that the coefficient for policy density in column 4 becomes negative, as one would expect, although it remains statistically non-significant. This model is still 'biased biased' because we haven't yet controlled for all backdoor paths opened due to the introduction of the new confounder variable (see above). Introducing the additional two confounders that account for share of services and share of imports, we cut off all backdoor paths and obtain a negative and statistically significant coefficient for policy density. The model in column 6-Table R1 is the one that Cunningham (2021, p110) calls 'unbiased conditional'.

In terms of standard errors variation, we can observe that there is minimal variation between column 3 (0.000149) and column 6 (0.000154). When we introduce the final controls, we observe a small correction in the magnitude of the coefficient, and minimal change in its standard error variation (0.000163 in column 7 and 8, and 0.000169 in column 9).

We provide additional sensitivity analysis in table R2 that further supports our reasoning. Once we drop time fixed effects, the coefficient for policy density always remains negative and statistically significant. This indicates that the change observed in Table R1 is due to the discussed collider effect, and by mainly shutting off backdoor pathways, we obtain 'unbiased conditional' estimates. Introducing controls one by one in the parsimonious model without time fixed effects further confirms our expectations based on DAG formation presented in Figure R1 and confirms findings for model specification including time fixed effects. We can also see that the coefficients for policy density are in general larger in terms of magnitude when excluding time fixed effects in Table R2 than the corresponding ones in Table R1. The larger coefficients without time fixed effects represent overestimated effects that include spurious correlation from shared time trends. In other words, omitting time fixed effects would fail to control for unobserved variation in emission intensities from 2000 to 2022, leading to biased model estimates.

Secondary control variables - potential backdoor paths

We include a set of secondary controls in Figure R1b, which are not directly related to policy density, but may potentially have a spurious influence on the identified direct treatment effect if left uncontrolled. These variables are not part of the main DAG pathways shown in in Figure R1a. They help ensure robustness by capturing broader contextual variation not absorbed by our main controls or fixed effects. More specifically, introducing GDP per capita may raise concerns about opening alternative backdoor paths related to cyclical fluctuation in the business cycle and the nature of institutions. To capture the former, we introduce the Hodrick-Prescott GDP filter that is commonly used to account for cyclical volatility in economic activity. To capture institutional quality, we include the World Bank's rule of law index which varies across countries and time, reflecting path dependency in a country's institutional development (Acemoglu and Robinson 2002). We expect both these secondary controls to be correlated with GDP per capita and emission intensity, and not to be significantly associated with policy density. Sensitivity analysis confirms this (see our response to R3.4 below).

Finally, we introduce temperature variation from the long-term mean. This variable is exogenous to the treatment variable and helps capture non-linear temperature anomalies

that could be associated with emission intensity fluctuations. It is not plausibly influenced by policy density, and its inclusion strengthens the robustness of our estimates without introducing endogeneity (see our response to R3.6 below).

Summary of Revisions to Manuscript & SI:

We have extended and strengthened the explanation of controls in our Methods section, and have included an edited version of this full response as a dedicated new section in SI1 on our selection and justification of controls and their effect on our identification strategy for the effect of policy density on emission intensity.

We distinguish the three confounders (GDP per capita, imports share, and services share) from the three secondary control variables (HP filter, rule of law, long-term temperature variation), and link to corresponding new model results in relevant Extended Data Tables.

Table R1. Sensitivity analysis with sequential addition of controls to the baseline model (also included as Table ED16(a) in Extended Data).

VARIABLES	(1)	(2)	(3)	(4)	(5)	(6)	(7)	(8)	(9)
	LOG(CO2/G DP)	LOG(CO2/G DP)	LOG(CO2/G DP)	LOG(CO2/G DP)	LOG(CO2/G DP)	LOG(CO2/G DP)	LOG(CO2/G DP)	LOG(CO2/G DP)	LOG(CO2/G DP)
Policy density)	-0.00127*** (0.000388)	-0.00450*** (0.000184)	0.000479*** (0.000149)	-0.000177 (0.000150)	-0.000247* (0.000148)	-0.000454*** (0.000154)	-0.000514*** (0.000163)	-0.000505*** (0.000163)	-0.000554*** (0.000169)
GDP per capita (log)				3.907*** (0.251)	4.258*** (0.280)	3.889*** (0.252)	3.866*** (0.287)	3.822*** (0.285)	3.735*** (0.288)
GDP per capita (log) squared				-0.211*** (0.0131)	-0.229*** (0.0147)	-0.210*** (0.0134)	-0.207*** (0.0151)	-0.204*** (0.0150)	-0.200*** (0.0153)
Services share of GDP					-0.00856*** (0.00211)	-0.00948*** (0.00211)	-0.00871*** (0.00218)	-0.00896*** (0.00221)	-0.00782*** (0.00220)
Imports share of GDP						-0.00435*** (0.000611)	-0.00434*** (0.000635)	-0.00435*** (0.000637)	-0.00410*** (0.000647)
Rule of law							-0.0424 (0.0298)	-0.0436 (0.0298)	-0.0374 (0.0301)
Hodrick-Prescott GDP filter								-0.330 (0.254)	-0.354 (0.249)
Temperature variation									-0.0176** (0.00792)
Constant	-1.544*** (0.0213)	-1.411*** (0.00849)	-1.616*** (0.00723)	-19.25*** (1.199)	-20.43*** (1.288)	-18.41*** (1.143)	-18.52*** (1.329)	-18.34*** (1.323)	-17.94*** (1.318)
FE - Country	NO	YES	YES	YES	YES	YES	YES	YES	YES
FE - Year	NO	NO	YES	YES	YES	YES	YES	YES	YES
Observations	1,027	1,027	1,027	1,027	1,017	1,017	974	974	941
R-squared	0.013	0.890	0.954	0.964	0.965	0.967	0.967	0.967	0.968

Notes: Dependent variable is the log of emission intensity. Robust standard errors are reported in parentheses. *** p<0.01, ** p<0.05, * p<0.1

Table R2. Sensitivity analysis with sequential addition of controls to the baseline model excluding time fixed effects (also included as Table ED16(b) in Extended Data).

VARIABLES	(1)	(2)	(3)	(4)	(5)	(6)	(7)	(8)
	LOG(CO2/GDP)	LOG(CO2/GDP)	LOG(CO2/GDP)	LOG(CO2/GDP)	LOG(CO2/GDP)	LOG(CO2/GDP)	LOG(CO2/GDP)	LOG(CO2/GDP)
Policy density	-0.00127*** (0.000388)	-0.00450*** (0.000184)	-0.00272*** (0.000165)	-0.00237*** (0.000155)	-0.00220*** (0.000141)	-0.00221*** (0.000155)	-0.00217*** (0.000154)	-0.00196*** (0.000153)
GDP per capita (log)			6.324*** (0.313)	6.710*** (0.314)	5.580*** (0.283)	5.690*** (0.303)	5.727*** (0.303)	5.267*** (0.299)
GDP per capita (log) squared			-0.347*** (0.0160)	-0.364*** (0.0158)	-0.303*** (0.0148)	-0.309*** (0.0157)	-0.312*** (0.0158)	-0.287*** (0.0156)
Services share of GDP				-0.0177*** (0.00225)	-0.0175*** (0.00223)	-0.0180*** (0.00238)	-0.0163*** (0.00237)	-0.0146*** (0.00228)
Imports share of GDP					-0.00662*** (0.000690)	-0.00647*** (0.000716)	-0.00669*** (0.000709)	-0.00593*** (0.000652)
Rule of law						0.0516* (0.0296)	0.0539* (0.0296)	0.0345 (0.0297)
Hodrick-Prescott GDP filter							0.665*** (0.206)	0.715*** (0.202)
Temperature variation								-0.0503*** (0.00763)
Constant	-1.544*** (0.0213)	-1.411*** (0.00849)	-29.42*** (1.526)	-30.55*** (1.520)	-25.22*** (1.313)	-25.75*** (1.413)	-25.90*** (1.412)	-23.88*** (1.387)
FE - Country	NO	YES	YES	YES	YES	YES	YES	YES
FE - Year	NO	NO	NO	NO	NO	NO	NO	NO
Obs.	1,027	1,027	1,027	1,017	1,017	974	974	941
R-squared	0.013	0.890	0.944	0.949	0.955	0.955	0.956	0.960

Notes: Dependent variable is the log of emission intensity. Robust standard errors are reported in parentheses. *** p<0.01, ** p<0.05, * p<0.1

R3.4 There are some variables that are possible mediators, ie variables that could be affected by the treatment and in turn can affect the outcome, namely GDP, the service share, and the import share. I would like to see these variables regressed on policy density, including also lagged density since adjustment to supply chains and sectoral composition likely take several years. If policy density affects these variables, then they are bad controls, bias beta1, and should not be included (Angrist and Pischke 2009, Cinelli et al 2024).

Response:

As discussed above, we reason that GDP per capita, services share, and imports share are confounders (not mediators as suggested by the reviewer). Their inclusion is necessary to reduce omitted variable bias and improve causal identification.

In response to the reviewer’s suggestion, we estimate simple regression models of these variables on policy density (Table R3) and lagged policy density (Table R4). Consistent with the relationships outlined in our DAG (Figure R1), we find that GDP per capita (and its squared term), services share, and imports share are significantly associated with policy density. In contrast, the Hodrick-Prescott GDP filter and rule of law are not significantly associated with policy density, confirming again our expectations (Figure R2). Instead, they are correlated with GDP per capita (Table R5) and emission intensity, supporting our decision to treat them as secondary controls rather than confounders.

Table R3. Sensitivity analysis: testing for associations between treatment variable (policy density) and control variables.

VARIABLES	(1) Policy density	(2) Policy density	(3) Policy density	(4) Policy density	(5) Policy density
GDP per capita (log)	498.5*** (62.60)				
GDP per capita (log) squared	-25.13*** (3.108)				
Services share of GDP		2.288*** (0.552)			
Imports share of GDP			-0.945*** (0.142)		
Hodrick-Prescott GDP filter				0.612 (42.58)	
Rule of law					-6.795 (6.513)
Constant	-2,412*** (318.6)	-99.32*** (33.64)	82.09*** (6.248)	41.17*** (0.645)	49.74*** (6.317)
FE - Country	YES	YES	YES	YES	YES
FE - Year	YES	YES	YES	YES	YES
Observations	1,032	1,022	1,029	1,028	989
R-squared	0.818	0.806	0.810	0.797	0.807

The dependent variable is policy density.

Robust standard errors are reported in parentheses. *** p<0.01, ** p<0.05, * p<0.1.

Table R4. Sensitivity analysis: testing for associations between lagged treatment variable (policy density) and control variables.

VARIABLES	(1) L1 Policy density	(2) L1 Policy density	(3) L1 Policy density	(4) L1 Policy density	(5) L1 Policy density
-----------	-----------------------------	-----------------------------	-----------------------------	-----------------------------	-----------------------------

GDP per capita (log)	476.3***				
	(65.88)				
GDP per capita (log) squared	-23.82***				
	(3.241)				
Services share of GDP		2.113***			
		(0.559)			
Imports share of GDP			-0.863***		
			(0.140)		
Hodrick-Prescott GDP filter				-10.90	
				(41.51)	
Rule of law					-1.688
					(7.009)
Constant	-2,326***	-91.10***	76.43***	38.85***	42.58***
	(338.3)	(34.13)	(6.159)	(0.626)	(6.788)
FE - Country	YES	YES	YES	YES	YES
FE - Year	YES	YES	YES	YES	YES
Observations	989	979	986	985	946
R-squared	0.814	0.801	0.807	0.795	0.805

The dependent variable is one-year lagged policy density.

Robust standard errors are reported in parentheses. *** p<0.01, ** p<0.05, * p<0.1.

Table R5. Sensitivity analysis: testing for associations between GDP per capita and secondary control variables.

VARIABLES	(1) GDP per capita	(2) GDP per capita
Hodrick-Prescott GDP filter	1.606***	
	(0.278)	
Rule of law		0.345***
		(0.0340)
Constant	10.51***	10.18***
	(0.00364)	(0.0334)
FE - Country	YES	YES
FE - Year	YES	YES
Observations	1,028	989
R-squared	0.969	0.974

The dependent variable is GDP per capita.

Robust standard errors are reported in parentheses. *** p<0.01, ** p<0.05, * p<0.1

Summary of Revisions to Manuscript & SI:

As noted in the previous response, we have extended and strengthened the explanation of confounders and secondary controls in our Methods section, and as a dedicated new section in SI1 on our selection of model variables and their effect on our identification strategy.

R3.5 There are some variables which are basically time-invariant and are already accounted for by the unit FE, eg institutions. In that sense, the standard deviation reported in supplementary table 1 is misleading because it does not reflect the variance that is actually used to estimate the models. It would be helpful to report also the within SD here, ie the SD of the FE residuals (Mummolo and Peterson 2018). I would assume that almost all variance in the variables related to institutions and economic structure is absorbed by the FE.

Response:

Following Eskander and Fankhauser (2023), we reason that country-fixed effects include time-invariant differences in socioeconomic context, political culture, institutional quality, and resource endowment between countries. With respect to institutional quality, we acknowledge that institutions are typically slow-moving and path-dependent, but over the 2000–2022 period we have observed meaningful institutional shifts in several countries particularly within the BRIICS including through crises, regime changes, democratisation, autocratization, and reforms (see Figure R4). These institutional shifts may impact the effectiveness with which climate policy density affects emission intensities (e.g., through changes to policy credibility, monitoring, and enforcement). Considering this variation, we include the World Bank’s rule of law index as a control. The rule of law index from the Worldwide Governance Indicators (WGI) is reported annually and varies across countries and time (Versteeg and Ginsburg 2017).

In response to the reviewer’s concern, we computed and report both the overall standard deviation and the within-unit standard deviation (i.e., fixed-effects-residualised) following Mummolo and Peterson (2018). This allows us to examine the proportion of meaningful variation that is actually used for identification in the fixed effects models. Table R6 follows the structure of Table 1 in Mummolo and Peterson (2021) to show within-unit variation for rule of law. While reduced, it remains meaningful (retaining 17% of its total variance), supporting our choice to include it in our models. Similarly, variables controlling for economic structure (GDP per capita and its squared term, services share, and imports share) retain on average 20% of their variation once we account for fixed effects. The treatment variable, policy density, retains just under half of its variation, highlighting its suitability for our research design. As expected, secondary controls, Hodrick-Prescott filter and temperature variation retain over two-thirds of their variance, consistent with their role in capturing cyclical and exogenous fluctuations.

Figure R4. Variation in Rule of Law index 2000-2022 in BRIICS, with UK as comparison.

Table R6. Standard deviation of independent variables before and after controlling for fixed effects (also included as Table ED19(a) in Extended Data).

	(1) Coefficient	(2) SD	(3) Within unit SD	(4) Δ%
--	--------------------	-----------	-----------------------	-----------

Policy density	-0.000554	44.95	20.24	-54.99
GDP per capita (log)	3.735	0.64	0.12	-81.83
GDP per capita (log) squared	-0.200	13.12	2.25	-82.87
Services share to GDP	-0.00782	7.71	1.67	-78.31
Imports share to GDP	-0.00410	25.86	5.36	-79.27
Rule of Law	-0.0374	0.83	0.13	-83.80
Hodrick-Prescott GDP filter	-0.352	0.019	0.013	-33.45
Temperature variation	-0.0176	0.52	0.40	-23.70

Column 1 shows the coefficients estimated in column 9 of Table R1. Column 2 indicates the standard deviation of the independent variables for the full sample. Column 3 shows the within-unit standard deviation meaning that variables are standardised for country and year fixed effects (Mummolo and Peterson 2018). Column 4 show the % difference (Δ) between standard deviations reported in column 3 and 2.

Summary of Revisions to Manuscript & SI:

We have included an edited version of this response (including Table R6) as a section in our SI1 discussion of controls and our model specification including both rule of law as a secondary control variable and country fixed effects.

R3.6 Temperature varies randomly, given the unit and time FE. Regressing temperature anomalies on policy density should yield an insignificant effect which might be a good placebo test. As such it is no confounder / omitted variable and it is neither necessary to include it as control nor does including it bias the estimate of beta1.

Response:

We incorporate temperature anomalies from the long-term mean in our model to account for climate variability, given that it might not be fully captured by year fixed effects, which account only for linear time trends. Recent data from NASA (<https://svs.gsfc.nasa.gov/5452>) show an increase in both the frequency and intensity of temperature anomalies over the last decade, with the warming trend accelerating at different rates across countries.

In our view, regressing temperature variation on policy density does not constitute a robust placebo treatment test in this context. This is because both policy density and temperature variation are in linear form, while our baseline model estimates the treatment effect on a log-transformed outcome. Consequently, we expect that this simple placebo test model would yield spurious results as it does not account for the accelerating relationship. In this, we follow the caution advised by Hartman and Hidalgo (2018), who critique use of underpowered placebo treatment tests for causal validation in applied political science. Similarly, Eggers, Tuñón et al. (2024) emphasise the importance of matching the logic of the placebo test to the research design.

Nonetheless, we agree that it is important to ensure the robustness of our research design, so we follow the reviewer's suggestion to regress temperature variation on policy density (Table R7, column 1). As expected, results appear statistically significant, which we interpret as a model misspecification. To correct for accelerating trends and potential heteroskedasticity (particularly relevant given increasing temperature variability across countries in recent years), we log-transform temperature variation. This log-linearised model confirms that the coefficient for temperature variation is statistically non-significant, as expected, thereby supporting the exogeneity of this control.

To further support our argument, we re-estimate the model using as dependent variable the log of policy density and as independent variable (i) temperature variation (column 3), and (ii) the log of temperature variation (column 4). The absent of significant evidence in both

models confirms that once either or both variables are log-transformed, temperature variation shows no systematic association with policy density.

Table R7. Sensitivity analysis: testing for associations between treatment variable (policy density) and control variable for temperature variation from long-term mean.

VARIABLES	(1) Policy density	(2) Policy density	(3) LOG(Policy density)	(4) LOG(Policy density)
Temperature variation	-4.775*** (1.469)		-0.0285 (0.0206)	
Temperature variation (log)		-0.783 (0.799)		0.00529 (0.0118)
Constant	40.03*** (0.686)	44.71*** (1.193)	3.106*** (0.0102)	3.325*** (0.0174)
FE - Country	YES	YES	YES	YES
FE - Year	YES	YES	YES	YES
Observations	989	620	968	611
R-squared	0.797	0.845	0.945	0.950

The dependent variable in columns 1 and 2 is policy density; in columns 3 and 4, it is the log of policy density.

Robust standard errors are reported in parentheses. *** $p < 0.01$, ** $p < 0.05$, * $p < 0.1$.

To further test for robustness, we examine whether the inclusion of temperature variation affects identification of the treatment effect. Specifically, we compare the coefficient for policy density in the full model with temperature variation (Table R1, column 8) to the corresponding coefficient in the reduced model without temperature variation (Table R1, column 7,) using an F-test of coefficient equality. We estimate χ^2 statistic equal to 0.70 with p -value 0.40. This means that the test fails to reject the null hypothesis of no difference, suggesting that including temperature variation in the model does not significantly alter the estimated treatment effect. This supports our decision to include it as a control without compromising causal identification. Consequently, we retain the original temperature variation variable in our baseline model specification. This is also consistent with the precedents on which we build and so facilitates comparability of our model estimates with prior studies (Eskander and Fankhauser 2020, Eskander and Fankhauser 2023).

Summary of Revisions to Manuscript & SI:

We have included an edited version of this response (including tables) in our SI1 discussion of controls and our model specification including temperature variation from long-term mean as a secondary control variable.

R3.7 Since no track changes are provided, it is not immediately obvious what the authors changed in the manuscript. However, I was not able to find any discussion of possible mediators in the main text (besides the moderating role of targets).

Response:

Our revisions to the manuscript in the first round were extensive and we uploaded a track changed version, so we are mystified this was not made available to the reviewer. We have checked this with the journal production team, and trust our tracked change version of the manuscript after this second round of revisions is accessible. As a back-up, we summarise here our discussion of mediators which is substantially strengthened in this second revision (see R3.2 & R3.3 on model specification and DAGs).

Summary of Revisions to Manuscript & SI:

In SI1, we include a new dedicated section on model specification that includes our conceptualisation of confounders in both basic and full DAGs, with further analysis of backdoor paths and colliders. We reference this full discussion in Methods. As the reviewer notes, we also have extended discussion in main text, Methods, and SI1 on long-term targets and organisational dummies as moderators of climate policy portfolios' effect on emission intensities.

R3.8. Related to R3.5, in the current presentation of the results it is not clear how large the effect sizes are. Currently, the interpretation seems to be "An increase in policy density by 1, decreases emissions intensity by 0.05%". Is this a practically relevant effect size? Is a marginal change by 1 in policy density still plausible after partialing out the the FE? It would be very helpful to report also estimates that are standardized using the within SD as suggested by Mummolo and Peterson (2018). This way all variables would be on the same scale and the reader could easily assess the relevance of the effect sizes.

Response:

Our treatment variable, policy density, is a count variable that captures the accumulation of discrete policies. Consequently we prefer to present effect sizes in terms of a one-unit change, following (Mummolo and Peterson 2018): *"Some might wish to convey treatment effects in the common scenario of a one-unit shift. This may be especially useful if the treatment being studied is dichotomous or a count variable that takes only integer values, in which case unit shifts arguably make more substantive sense to consider."*

This is also consistent with prior studies that express effect sizes in relation to unit increases in policy density (Eskander and Fankhauser 2020, Nachtigall, Lutz et al. 2022, Eskander and Fankhauser 2023).

However, we agree that standardising coefficients using the within-group standard deviation can improve interpretability, particularly when comparing effect sizes across variables measured on different scales. In response to the reviewer's comment, and in line with the growing use of this method in applied political science, we implement the standardisation procedure proposed by (Mummolo and Peterson 2018). We residualise each predictor with respect to country and year fixed effects, calculate the within-unit standard deviation of the residuals, and use these to standardise the independent variables before re-estimating the model.

Results of this re-estimated model are presented in Table R8 below. We find that a 1 standard deviation increase in policy density (equivalent to 20 polices, see Table R6) is associated with a 1% reduction in emission intensity. As shown in Table R8, the standardised effect of policy density is comparable in magnitude to that of key confounders (GDP per capita, services share, imports share), and substantially larger than that of secondary controls (rule of law, Hodrick-Prescott filter, temperature variation).

Table R8. Standardised coefficients using the within-group standard deviation (also included as Table ED19(b) in Extended Data).

VARIABLES	(1) Original Coefficients - Table R1 column 9	(2) Standardised Coefficients (within SD)
Total policy density	-0.000554	-0.0112
GDP per capita (log)	3.735	0.4369
GDP per capita (log) squared	-0.200	-0.4493
Services share to GDP	-0.00782	-0.0131

Imports share to GDP	-0.00410	-0.0220
Rule of law	-0.0374	-0.0050
Hodrick-Prescott GDP filter	-0.352	-0.0046
Temperature variation	-0.0176	-0.0070

Column 1 shows the coefficients for the model specification in Table R1, column 9. Column 2 shows the standardised coefficients using the within-group standard deviation as proposed by Mummolo and Peterson (2018).

Summary of Revisions to Manuscript & SI:

We have included additional text in the manuscript and Methods, Supporting Information, and Extended Data to capture this analysis. For example in Methods, we explain the magnitude of effect size in relation to the confounders and secondary controls. In SI1, we provide further explanation in a new dedicated section linking to full model results in Table ED19.

R3.9 In response to my previous comment, the authors provide interesting additional estimates by type of instrument in ED13. Here it is striking that the effect legal interventions is MUCH larger than any of the other types (eg 100 times larger than of market-based economic instruments). Some other instruments do not significantly affect CO2 intensity at all. This seems like the effect could be driven entirely by legal instruments. is this the case, ie is the coefficient still significant when omitting legal instruments from the policy density index?

Response:

To directly address the reviewer's concern, we perform the proposed test by removing legal instruments from both the policy density index (Table R10-column 4). When comparing results with the baseline model specifications (Table R10-column 2), we can see that the coefficient changes only marginally from -0.116 to -0.117. This confirms that the main result is not driven by legal instruments.

To investigate further we went back to our policy coding to reexamine the descriptive statistics for all the instruments in our dataset, but with a particular focus on legal and also monitoring instrument types as these had much fewer occurrences ($n < 10$) than the other instrument types (all $n > 25$). To improve robustness given these small sample sizes, we folded legal and monitoring instruments together with regulatory instruments, reducing the number of instrument types from eight to six. We then re-estimate our original model (Table R10-column 2) using six policy instrument types (Table R10-column 3). Results remain virtually the same with the coefficient for the instrument type HHI marginally changing from -0.116 to -0.114 and remaining strongly significant. This further confirms that this effect is not driven by individual instrument types.

Table R10. Sensitivity analysis with instrument type taxonomies

VARIABLES	Baseline	Baseline	New baseline	Sensitivity
	(1) Log(CO2/GDP)	(2) LOG(CO2/GDP)	(3) LOG(CO2/GDP)	(4) LOG(CO2/GDP)
Policy density	-0.000554*** (0.000169)	-0.000682*** (0.000165)	-0.000688*** (0.000166)	-0.000681*** (0.000165)
HHI - 8 instrument types		-0.116*** (0.0402)		
HHI - 6 instrument types (legal & monitoring combined with regulatory)			-0.114***	

			(0.0407)	-0.117***
HHI - 7 instrument types (legal removed)				(0.0402)
Constant	-17.94*** (1.318)	-18.10*** (1.380)	-18.08*** (1.378)	-18.09*** (1.380)
Controls	YES	YES	YES	YES
FE - Country	YES	YES	YES	YES
FE - Year	YES	YES	YES	YES
Observations	941	919	919	919
R-squared	0.968	0.969	0.969	0.969

Robust standard errors are reported in parentheses. *** p<0.01, ** p<0.05, * p<0.1.

We further replicate our original table ED13 with the single instrument type models using a three category taxonomy: regulatory instruments, economic (including market-based and direct provision instrument types), and soft (including planning, information, and voluntary instrument types). This is also the more aggregated taxonomy we used in our main model for testing instrument type diversity in policy portfolios.

Results show a significant effect for economic instruments (-0.0015). The coefficient for regulatory instruments (-0.000359) is non-significant and similar in size to soft instruments (-0.00077) and one order of magnitude smaller than economic instruments. Table ED13 has the full results including controls.

Given that our dependent variable is emission intensity (CO₂/GDP), we interpret these results with caution as the independent variable comprises only a subset of all policies (coded as being of a particular instrument type). Nevertheless, the null result for regulations is surprising.

Our interpretation is that regulations as an instrument type span a wide range from binding performance and emission standards to weaker reporting, safety, or compliance type regulations. Such regulations include, for example, safety and performance requirements for heaters, air conditioners or refrigerators in buildings (such as the regulation "GB 4706.13-2014 on Safety on Household and Similar Electrical Appliances" adopted in China in 2016), mandatory reporting requirements for energy efficiency measures (such as a regulation adopted in the Netherlands in 2019) or guidelines for estimating greenhouse gas emissions (such as a regulation adopted in Australia in 2017). This increases the heteroskedasticity (noise) within our coding of regulatory instrument types.

Stechemesser, Koch et al. (2024) also find that regulatory instruments tend to be more effective when combined with economic instruments: *"some of the most widely used regulatory instruments and subsidy schemes may require complementary instruments to enable substantial emission reductions. The effect sizes of policy mixes that combine these non-price-based instruments with taxation or reduced fossil fuel subsidies suggest that in most cases pricing is the complement that enables effective emission reductions."* (Stechemesser, Koch et al. 2024).

Best, Burke et al. (2020), Köppl and Schratzenstaller (2023), and Döbbeling-Hildebrandt, Miersch et al. (2024) provide further evidence supporting the effectiveness of market based instruments, and in particular carbon taxation, in reducing CO₂ emissions.

Summary of Revisions to Manuscript & SI:

We have included additional text in both main manuscript and Methods explaining the six instrument type classification. We also replace our original 8 instrument type HHI diversity measure in all relevant models with the more robust 6 instrument type HHI that avoids single instrument types with few occurrences in our dataset. This includes the main model reported in Table 1 of the manuscript and all relevant Extended Data Tables that report coefficients for instrument type variables.

In SI1's dedicated section on policy density by instrument type, we update the text to remove discussion of the previous 8 instrument type models, and focus on the results presented above on the significance and effect of economic instruments relative to the other two instrument type categories.

In Table ED13 we provide the full results for models that test policy density of only one instrument type at a time.

R3.10 It would be less confusing to present the regression results in Table 1 in a standard format (models in columns, not rows). The caption should explain how to interpret the coefficients (as in line 663ff). Also it could be clarified here what is included in the 3 types and 8 types.

Response:

In our original submission, we presented the regression results in standard format (models in columns, not rows), but another reviewer asked us to change the format as it created too many blank cells. So in our revised submission, we presented the regression results in the current format (models in rows) to address this problem but we agree it is harder to interpret. In this second revision, we revert back to the standard format, but split the table into two, the first on the main effects of our policy portfolio variables (Table 1), the second reporting interaction terms with our dummy variables (Table 2).

We have included a short explanation for interpreting coefficients in the table caption as the reviewer suggests. We've also included a note on the 6 instrument types for ease of interpretation.

R3.11 The link to the replication materials does not work for me so I cannot comment on the availability of the materials.

Response:

We had uploaded and checked the links for all our replication materials worked on our previous submission, so as with the tracked changes version of our manuscript, we're mystified why these did not work and apologise to the reviewer. In this submission we've again uploaded our Stata .do and .dta files and checked accessibility. If these do not work for the reviewer, we'd be grateful if the journal office could be alerted so we can try and work out what the problem is.

Additional Bibliography

Acemoglu, D. and J. A. Robinson (2002). "The Political Economy of the Kuznets Curve." Review of Development Economics 6(2): 183-203.

Adam, C., S. Hurka, C. Knill and Y. Steinebach (2019). Policy accumulation and the democratic responsiveness trap. Cambridge, UK, Cambridge University Press.

Angrist, J. D. and J.-S. Pischke (2010). "The Credibility Revolution in Empirical Economics: How Better Research Design Is Taking the Con out of Econometrics." Journal of Economic Perspectives **24**(2): 3–30.

Babiker, M., P. Bertoldi, C. Bataille, F. Creutzig, N. K. Dubash, M. Grubb, E. Haites, B. Hinder, J. Hoppe, Y. Kim, G. F. Nemet, A. Patt, Y. Saheb and R. Slade (2022). Cross-Chapter Box 10: Policy Attribution. Climate Change 2022: Mitigation of Climate Change. Working Group III contribution to the Sixth Assessment Report of the Intergovernmental Panel on Climate Change (IPCC). P. R. Shukla, J. Skea, R. Slade et al. Geneva, Switzerland, Intergovernmental Panel on Climate Change (IPCC): 1479-1481.

Bertram, C., G. Luderer, R. C. Pietzcker, E. Schmid, E. Kriegler and O. Edenhofer (2015). "Complementing carbon prices with technology policies to keep climate targets within reach." Nature Clim. Change **5**(3): 235-239.

Best, R., P. J. Burke and F. Jotzo (2020). "Carbon Pricing Efficacy: Cross-Country Evidence." Environmental and Resource Economics **77**(1): 69-94.

Celli, V. (2022). "Causal mediation analysis in economics: Objectives, assumptions, models." Journal of Economic Surveys **36**(1): 214-234.

Cinelli, C., A. Forney and J. Pearl (2024). "A Crash Course in Good and Bad Controls." Sociological Methods & Research **53**(3): 1071-1104.

Cinelli, C. and C. Hazlett (2019). "Making Sense of Sensitivity: Extending Omitted Variable Bias." Journal of the Royal Statistical Society Series B: Statistical Methodology **82**(1): 39-67.

Creutzig, F., S. Becker, P. Berrill, C. Bongs, A. Bussler, B. Cave, S. M. Constantino, M. Grant, N. Heeren, E. Heinen, M. J. Hintz, T. Ingen-Housz, E. Johnson, N. Kolleck, C. Liotta, S. Lorek, G. Mattioli, L. Niamir, T. McPhearson, N. Milojevic-Dupont, F. Nachtigall, K. Nagel, H. Närgler, M. Pathak, P. Perrin de Brichambaut, D. Reckien, L. A. Reisch, A. Revi, F. Schuppert, A. Sudmant, F. Wagner, J. Walkenhorst, E. Weber, M. Wilmes, C. Wilson and A. Zekar (2024). "Towards a public policy of cities and human settlements in the 21st century." npj Urban Sustainability **4**(1): 29.

Cunningham, S. (2021). Causal inference: The mixtape, Yale university press.

Döbbling-Hildebrandt, N., K. Miersch, T. M. Khanna, M. Bachelet, S. B. Bruns, M. Callaghan, O. Edenhofer, C. Flachsland, P. M. Forster, M. Kalkuhl, N. Koch, W. F. Lamb, N. Ohlendorf, J. C. Steckel and J. C. Minx (2024). "Systematic review and meta-analysis of ex-post evaluations on the effectiveness of carbon pricing." Nature Communications **15**(1): 4147.

Eggers, A. C., G. Tuñón and A. Dafoe (2024). "Placebo Tests for Causal Inference." American Journal of Political Science **68**(3): 1106-1121.

Eskander, S., S. Fankhauser and J. Setzer (2021). "Global lessons from climate change legislation and litigation." Environmental and Energy Policy and the Economy **2**(1): 44-82.

Eskander, S. M. S. U. and S. Fankhauser (2020). "Reduction in greenhouse gas emissions from national climate legislation." Nature Climate Change **10**(8): 750-756.

Eskander, S. M. S. U. and S. Fankhauser (2023). "The Impact of Climate Legislation on Trade-Related Carbon Emissions 1996–2018." Environmental and Resource Economics **85**(1): 167-194.

Grubb, M., P. Drummond, A. Poncia, W. McDowall, D. Popp, S. Samadi, C. Peñasco, K. Gillingham, S. Smulders, M. Glachant, G. Hassall, E. Mizuno, E. S. Rubin, A. Dechezlepretre and G. Pavan (2021). "Induced innovation in energy technologies and systems: a review of evidence and potential implications for CO₂ mitigation." Environmental Research Letters.

Grubb, M., C. Okereke, J. Arima, V. Bosetti, Y. Chen, J. Edmonds, S. Gupta, A. Köberle, S. Kverndokk, A. Malik and L. Sulistiawati (2022). Chapter 1: Introduction and Framing. Climate Change 2022: Mitigation of Climate Change. Working Group III contribution to the Sixth Assessment Report of the Intergovernmental Panel on Climate Change (IPCC). P. R. Shukla, J. Skea, R. Slade et al. Geneva, Switzerland, Intergovernmental Panel on Climate Change (IPCC).

Hartman, E. and F. D. Hidalgo (2018). "An equivalence approach to balance and placebo tests." American journal of political science **62**(4): 1000-1013.

Herrendorf, B., R. Rogerson and Á. Valentinyi (2014). Chapter 6 - Growth and Structural Transformation. Handbook of Economic Growth. P. Aghion and S. N. Durlauf, Elsevier. **2**: 855-941.

Hoppe, J., B. Hinder, R. Rafaty, A. Patt and M. Grubb (2023). "Three Decades of Climate Mitigation Policy: What Has It Delivered?" Annual Review of Environment and Resources **48**(1): 615-650.

Hünermund, P. and E. Bareinboim (2023). "Causal inference and data fusion in econometrics." The Econometrics Journal **28**(1): 41-82.

Jahn, D. (2024). "The stringency and potential impact of climate laws and policies in the European Union and the 21 OECD countries." npj Climate Action **3**(1): 90.

Köppl, A. and M. Schratzenstaller (2023). "Carbon taxation: A review of the empirical literature." Journal of Economic Surveys **37**(4): 1353-1388.

Linsenmeier, M., A. Mohommad and G. Schwerhoff (2022). "Policy sequencing towards carbon pricing among the world's largest emitters." Nature Climate Change **12**(12): 1107-1110.

Maamoun, N. (2019). "The Kyoto protocol: Empirical evidence of a hidden success." Journal of Environmental Economics and Management **95**: 227-256.

Meckling, J. and J. Nahm (2018). "The power of process: state capacity and climate policy." Governance **31**(4): 741-757.

Mummolo, J. and E. Peterson (2018). "Improving the Interpretation of Fixed Effects Regression Results." Political Science Research and Methods **6**(4): 829-835.

Nachtigall, D., L. Lutz, M. Cárdenas Rodríguez, F. M. D'Arcangelo, I. Hašičič, T. Kruse and R. Pizarro (2024). "The Climate Actions and Policies Measurement Framework: A Database to Monitor and Assess Countries' Mitigation Action." Environmental and Resource Economics **87**(1): 191-217.

Nachtigall, D., L. Lutz, M. C. Rodríguez, I. Hašičič and R. Pizarro (2022). The climate actions and policies measurement framework: A structured and harmonised climate policy database to monitor countries' mitigation action. OECD Environment Working Papers. Paris, France, OECD.

Nascimento, L., M. den Elzen, T. Kuramochi, S. Woollands, I. Dafnomilis, M. Moisiso, M. Roelfsema, N. Forsell and Z. Araujo Gutierrez (2023). "Comparing the Sequence of Climate Change Mitigation Targets and Policies in Major Emitting Economies." Journal of Comparative Policy Analysis: Research and Practice: 1-18.

Pahle, M., D. Burtraw, C. Flachsland, N. Kelsey, E. Biber, J. Meckling, O. Edenhofer and J. Zysman (2018). "Sequencing to ratchet up climate policy stringency." Nature Climate Change **8**(10): 861-867.

Pearl, J. (2009). Causality. Cambridge, Cambridge University Press.

Peñasco, C., L. D. Anadón and E. Verdolini (2021). "Systematic review of the outcomes and trade-offs of ten types of decarbonization policy instruments." Nature Climate Change **11**(3): 257–265.

Rodrik, D. (2000). "Institutions for high-quality growth: What they are and how to acquire them." Studies in Comparative International Development **35**(3): 3-31.

Rodrik, D. (2018). "What Do Trade Agreements Really Do?" Journal of Economic Perspectives **32**(2): 73–90.

Schaffer, L. M., B. Oehl and T. Bernauer (2021). "Are policymakers responsive to public demand in climate politics?" Journal of Public Policy: 1-29.

Schaffrin, A., S. Sewerin and S. Seubert (2015). "Toward a Comparative Measure of Climate Policy Output." Policy Studies Journal **43**(2): 257-282.

Schmidt, N. M. and A. Fleig (2018). "Global patterns of national climate policies: Analyzing 171 country portfolios on climate policy integration." Environmental Science & Policy **84**: 177-185.

Schmidt, T. S. and S. Sewerin (2019). "Measuring the temporal dynamics of policy mixes – An empirical analysis of renewable energy policy mixes' balance and design features in nine countries." Research Policy **48**(10): 103557.

Stechemesser, A., N. Koch, E. Mark, E. Dilger, P. Klösel, L. Menicacci, D. Nachtigall, F. Pretis, N. Ritter, M. Schwarz, H. Vossen and A. Wenzel (2024). "Climate policies that achieved major emission reductions: Global evidence from two decades." Science **385**(6711): 884-892.

Sterner, T. and J. Coria (2013). Policy instruments for environmental and natural resource management, Routledge.

UNFCCC (2020). Subsidiary Body for Implementation, Compilation and Synthesis of fourth biennial reports of Parties included in Annex I to the Convention. Bonn, Germany, UN Framework Convention on Climate Change.

Versteeg, M. and T. Ginsburg (2017). "Measuring the Rule of Law: A Comparison of Indicators." Law & Social Inquiry **42**(1): 100-137.

Wooldridge, J. M. (2021) "Two-Way Fixed Effects, the Two-Way Mundlak Regression, and Difference-in-Differences Estimators." Available at SSRN: <https://ssrn.com/abstract=3906345> DOI: <http://dx.doi.org/10.2139/ssrn.3906345>.

World Bank (2023). State and Trends of Carbon Pricing 2023. Washington, DC, World Bank.